 eLife RESEARCH ARTICLE

# Emergence and propagation of epistasis in metabolic networks

**Sergey Kryazhimskiy\***

Division of Biological Sciences, University of California, San Diego, La Jolla, United States

**Abstract** Epistasis is often used to probe functional relationships between genes, and it plays an important role in evolution. However, we lack theory to understand how functional relationships at the molecular level translate into epistasis at the level of whole-organism phenotypes, such as fitness. Here, I derive two rules for how epistasis between mutations with small effects propagates from lower- to higher-level phenotypes in a hierarchical metabolic network with first-order kinetics and how such epistasis depends on topology. Most importantly, weak epistasis at a lower level may be distorted as it propagates to higher levels. Computational analyses show that epistasis in more realistic models likely follows similar, albeit more complex, patterns. These results suggest that pairwise inter-gene epistasis should be common, and it should generically depend on the genetic background and environment. Furthermore, the epistasis coefficients measured for high-level phenotypes may not be sufficient to fully infer the underlying functional relationships.

## Introduction

Life emerges from an orchestrated performance of complex regulatory and metabolic networks within cells. The blueprint for these networks is encoded in the genome. Mutations alter the genome. Some of them, once decoded by the cell, perturb cellular networks and thereby change the phenotypes important for life. Understanding how mutations affect the function of cellular networks is key to solving many practical and fundamental problems, such as finding mechanistic causes of genetic disorders (*Hu et al., 2011*; *Fang et al., 2019*), deciphering the architecture of complex traits (*Zuk et al., 2012*; *Mackay, 2014*; *Wei et al., 2014*), building artificial cells (*Hutchison et al., 2016*), explaining past, and predicting future evolution (*Blount et al., 2008*; *Wiser et al., 2013*; *de Visser and Krug, 2014*; *Harms and Thornton, 2014*; *Kryazhimskiy et al., 2014*; *Sailer and Harms, 2017a*; *Sohail et al., 2017*). Conversely, mutations can help us learn how cellular networks are organized (*Phillips, 2008*; *van Opijnen and Camilli, 2013*).

To infer the wiring diagram of a cellular network that produces a certain phenotype, one approach in genetics is to measure the pairwise and higher-order genetic interactions (or 'epistasis') between mutations that perturb it (*Phillips, 2008*). Much effort has been devoted in the past 20 years to a systematic collection of such genetic interaction data for several model organisms and cell lines (*Kelley and Ideker, 2005*; *Lehner et al., 2006*; *Jasnos and Korona, 2007*; *Collins et al., 2007*; *St Onge et al., 2007*; *Typas et al., 2008*; *Roguev et al., 2008*; *Costanzo et al., 2010*; *Szappanos et al., 2011*; *Huang et al., 2012*; *Roguev et al., 2013*; *Bassik et al., 2013*; *Babu et al., 2014*; *Costanzo et al., 2016*; *van Leeuwen et al., 2016*; *Skwark et al., 2017*; *Du et al., 2017*; *Heigwer et al., 2018*; *Horlbeck et al., 2018*; *Norman et al., 2019*; *Liu et al., 2019*; *Kuzmin et al., 2018*; *New and Lehner, 2019*; *Celaj et al., 2020*). This approach is particulary powerful when the phenotypic effect of one mutation changes qualitatively depending on the presence or absence of a second mutation in another gene, for example when a mutation has no effect on the phenotype in the wildtype background, but abolishes the phenotype when introduced together with another mutation, such as synthetic lethality (*Tong et al., 2001*). Such qualitative genetic interactions can

**\*For correspondence:**
skryazhimskiy@ucsd.edu

**Competing interests:** The author declares that no competing interests exist.

often be directly interpreted in terms of a functional relationship between gene products (*Tong et al., 2001*; *Davierwala et al., 2005*; *Phillips, 2008*).

Most pairs of mutations do not exhibit qualitative genetic interactions. Instead, the phenotypic effect of a mutation may change measurably but not qualitatively depending on the presence or absence of other mutations in the genome (*Babu et al., 2014*; *Costanzo et al., 2016*). The genetic interactions can in this case be quantified with one of several metrics that are termed 'epistasis coefficients' (*Wagner et al., 1998*; *Hansen and Wagner, 2001*; *Mani et al., 2008*; *Wagner, 2015*; *Poelwijk et al., 2016*). Although some rules have been proposed for interpreting epistasis coefficients, in particular, their sign (*Dixon et al., 2009*; *Lehner, 2011*; *Baryshnikova et al., 2013*), the validity, and robustness of these rules are unknown because there is no theory for how functional relationships translate into measurable epistasis coefficients in any system (*Lehner, 2011*; *Domingo et al., 2019*). To avoid this major difficulty, most large-scale empirical studies focus on correlations between epistasis coefficients rather than on their actual values (but see *Velenich and Gore, 2013*, for a notable exception). Genes with highly correlated epistasis profiles are then interpreted as being functionally related (*Segrè et al., 2005*; *Bellay et al., 2011*; *Babu et al., 2014*; *Costanzo et al., 2016*; *Horlbeck et al., 2018*). Although this approach successfully groups genes into protein complexes and larger functional modules (*Michaut et al., 2011*; *Bellay et al., 2011*), it does not reveal the functional relationships themselves. As a result, many if not most, genetic interactions between genes and modules still await their biological interpretation (*Costanzo et al., 2016*; *Fang et al., 2019*).

While geneticists measure epistasis to learn the architecture of biological networks, evolutionary biologists face the reverse problem: they need to know how the genetic architecture constrains epistasis at the level of fitness. Epistasis determines the structure of fitness landscapes on which populations evolve (*Fragata et al., 2019*). Understanding it would bear on many important evolutionary questions, such as why so many organisms reproduce sexually (*Kondrashov, 2018*), how novel phenotypes evolve (*Blount et al., 2008*; *Bridgham et al., 2009*; *Natarajan et al., 2013*; *Harms and Thornton, 2014*), how predictable evolution is (*Weinreich et al., 2006*; *Tenaillon et al., 2012*; *Wiser et al., 2013*; *Kryazhimskiy et al., 2014*), etc. So far, evolutionary biologists have relied primarily on abstract models of fitness landscapes (see *Orr, 2005*, for a review), rather than those firmly grounded in organismal biochemistry and physiology (e.g. *Dykhuizen et al., 1987*; *Das et al., 2020*). For example, Fisher's geometric model—one of the most widely used fitness landscape models—is explicitly devoid of the physiological and biochemical details (*Fisher, 1930*; *Tenaillon, 2014*; *Martin, 2014*).

A theory of epistasis must address two challenges. First, it must specify how the architecture of a biological network constrains epistasis. Such knowledge is important not only for evolutionary questions, but also for the inference problem in genetics. Consider a biological network module that produces a phenotype of interest but whose internal structure is unknown. By genetically perturbing all genes within the module and measuring the phenotype in all single, double and possibly some higher-order mutants, we can obtain the matrix of epistasis coefficients. In principle, we can then fit a network topology and parameters to these data. However, without knowing what information about the network is contained in the matrix in the first place, we cannot be sure whether the inferred topology and parameters are close to their true values or represent one of many possible solutions consistent with the data.

The second challenge is that epistasis may arise at a different level of biological organization than where it is measured by the experimentalist or by natural selection. For example, geneticists are often interested in understanding the structures of specific regulatory or metabolic network modules (*Collins et al., 2007*; *Costanzo et al., 2010*). However, measuring the peformance of a module directly is often experimentally difficult or impossible. Then epistasis is measured for an experimentally accessible 'high-level' phenotype, such as fitness, which depends on the performance of the focal 'lower-level' module, but also on other unrelated modules. However, if we do not know how epistasis that originally emerged in one module maps onto epistasis that is measured, it is unclear what we can infer about module's internal structure.

Evolutionary biologists encounter a related problem when they wish to learn the evolutionary history of a protein or a larger cellular module. To do so, they would in principle need to know how different mutations in this module affected fitness of the whole organism in its past environment. But such information is rarely available. Instead, it is sometimes possible to reconstruct past mutations

and measure their biochemical effects in the lab (*Lunzer et al., 2005*; *Bridgham et al., 2009*; *Natarajan et al., 2013*; *Sarkisyan et al., 2016*). When interesting patterns of epistasis are identified at the biochemical level, it is usually assumed that the same patterns manifested themselves at the level of fitness and drove module's evolution. However, this is not obvious. If interactions with other modules distort epistasis as it propagates from the biochemical level to the level of fitness (*Snitkin and Segrè, 2011*), our ability to infer past evolutionary history from in vitro biochemical measurements could be diminished. Therefore, the second challenge that a theory of epistasis must address is how epistasis propagates from lower-level phenotypes to higher-level phenotypes.

There is a large body of theoretical and computational literature on epistasis. As early as 1934, Sewall Wright realized that epistasis naturally emerges in molecular networks (*Wright, 1934*). This was later explicitly demonstrated in many mathematical and computational models (e.g. *Kacser and Burns, 1981*; *Keightley, 1989*; *Szathmáry, 1993*; *Gibson, 1996*; *Keightley, 1996*; *Wagner et al., 1998*; *Omholt et al., 2000*; *Peccoud et al., 2004*; *Gjuvsland et al., 2007*; *Gertz et al., 2010*; *Fiévet et al., 2010*; *Pumir and Shraiman, 2011*). Metabolic control analysis became one of the most successful and general frameworks for understanding epistasis between metabolic genes (*Kacser and Burns, 1973*). *Dean et al., 1986*, *Dykhuizen et al., 1987*, *Dean, 1989*, *Lunzer et al., 2005*, *MacLean, 2010* used it to interpret the empirically measured fitness effects of mutations and their interactions in terms of the metabolic relationships between the products of mutated genes. *Kacser and Burns, 1981*, *Hartl et al., 1985*, *Keightley, 1989*, *Clark, 1991*, *Keightley, 1996*, *Bagheri-Chaichian et al., 2003*, *Bagheri and Wagner, 2004*, *Fiévet et al., 2010* explored the implications of epistasis in metabolism for genetic variation in populations, their response to selection, long-term evolutionary dynamics and outcomes, such as the evolution of dominance. However, most studies analyzed only the linear metabolic pathway (but see *Keightley, 1989*) and assumed that fitness equals flux through the pathway (but see *Szathmáry, 1993*), thereby bypassing the problem of epistasis propagation.

There have been few attempts to theoretically relate the molecular architecture of an organism to the types of epistasis that would arise for its high-level phenotypes, such as fitness. *Segrè et al., 2005* and *He et al., 2010* used flux balance analysis (FBA, *Orth et al., 2010*) to compute genome-wide distributions of epistasis coefficients in metabolic models of *Escherichia coli* and *Saccharomyces cerevisiae* and arrived at starkly discordant conclusions. Recently, *Alzoubi et al., 2019* showed that FBA is generally poor in predicting experimentally measured genetic interactions, suggesting that it might be difficult to understand the emergence and propagation of epistasis by relying exclusively on genome-scale computational models. *Sanjuán and Nebot, 2008* and *Macía et al., 2012* modeled various abstract metabolic and regulatory networks and found a possible link between epistasis and network complexity. The work by *Chiu et al., 2012* is a more systematic attempt to develop a general theory of epistasis. They established a fundamental connection between epistasis and the curvature of the function that maps lower-level phenotypes onto a higher-level phenotype. However, further progress has been so far hindered by uncertainty in what types of functions map phenotypes onto one another in real biological systems. Previous studies made various idiosyncratic choices with respect to this mapping, leaving us without a clear guidance as to the conditions or systems where they are expected to hold.

To overcome this problem, here I consider a whole class of hierarchical metabolic networks and obtain the family of all functions that determine how the effective activity of a larger metabolic module can depend on the activities of smaller constituent modules. There are several advantages to this approach. First, it leads to an intuitive understanding of how the structure of the network influences epistasis emergence and propagation. Second, my approach is based on basic biochemical principles, so it should be relevant for many phenotypes. For example, epistasis is often measured at the level of growth rate (*Jasnos and Korona, 2007*; *St Onge et al., 2007*; *Babu et al., 2014*; *Costanzo et al., 2016*), and metabolism fuels growth. Moreover, metabolic genes occupy a large fraction of most genomes (*Orth et al., 2011*) and the general organization of metabolism is conserved throughout life (*Csete and Doyle, 2004*). Thus, by understanding genetic interactions between metabolic genes, we will gain an understanding of a large fraction of all genetic interactions.

In my model, I consider a hierarchical network with first-order kinetics but arbitrary topology, and ask two questions related to the two challenges mentioned above. (1) How does an epistasis coefficient that arose at some level of the metabolic hierarchy propagate to higher levels of the hierarchy?

(2) How does the network topology constrain the value of an epistasis coefficient between two muta-
tions that affect different enzymes in this network? I obtain answers to these questions analytically in
the limiting case when the effects of mutations are vanishingly small. I then computationally probe
the validity of the conclusions outside of the domain where they are expected to hold.

My model is not intended to generate predictions of epistasis for any specific organism. Instead,
its main purpose is to provide a baseline expectation for how epistasis that emerges at lower-level
phenotypes manifests itself at higher-level whole-organism phenotypes, such as fitness, and what
kind of information may be gained from measurements of such higher-level epistasis. One possible
outcome of this analysis is that there may be fundamental limitations to what an epistasis measure-
ment at one level of biological organization can tell us about epistasis at another level. On the other
hand, if it turns out that there is a general correspondence between epistasis coefficients at different
levels in this simple model, then it may be worth developing more sophisticated and general models
on which inference from data can be based.

## Model

### Hierarchical metabolic network

Consider a set of metabolites $A = \{1, 2, \ldots, n\}$ with concentrations $S_1, \ldots, S_n$ which can be intercon-
verted by reversible first-order biochemical reactions. The rate of the reaction converting metabolite
$i$ into metabolite $j$ is $x_{ij}(S_i - S_j/K_{ij})$ where $K_{ij}$ is the equilibrium constant. The rate constants $x_{ij}$, which
satisfy the Haldane relationships $x_{ji} = x_{ij}/K_{ij}$ (***Cornish-Bowden, 2013***), form the matrix $\vec{x} = \|x_{ij}\|_{i,j=1}^{n}$.
The metabolite set $A$ and the rate matrix $\vec{x}$ define a biochemical network $\mathcal{N} = (A, \vec{x})$.

The first-order kinetics assumption makes the model analytically tractable, as discussed below;
biologically, it is equivalent to assuming that all enzymes are far from saturation. The rate constants
$x_{ij}$ depend on the concentrations and the specific activities of enzymes and therefore can be altered
by mutations. $K_{ij}$ characterize the fundamental chemical nature of metabolites $i$ and $j$ and cannot be
altered by mutations (***Savageau, 1976***).

The whole-cell metabolic network is large, and it is often useful to divide it into subnetworks that
carry out certain functions important for the cell. I define subnetworks mathematically as follows. I
say that two metabolites $i$ and $j$ are adjacent (in the graph-theoretic sense) if there exists an enzyme
that catalyzes a biochemical reaction between them, that is, if $x_{ij} > 0$. Now consider a subset of
metabolites $A_\mu \subset A$. For this subset, let $A_\mu^{\mathrm{IO}}$ be the set of all metabolites that do not belong to $A_\mu$
but are adjacent to at least one metabolite from $A_\mu$. Let $\vec{x}_\mu$ be the submatrix of $\vec{x}$ which corresponds
to all reactions where both the product and the substrate belong to either $A_\mu$ or $A_\mu^{\mathrm{IO}}$. The metabolite
subset $A_\mu$ and the rate matrix $\vec{x}_\mu$ form a subnetwork $\mu = (A_\mu, \vec{x}_\mu)$ of network $\mathcal{N}$. I refer to $A_\mu$ and $A_\mu^{\mathrm{IO}}$
as the sets of internal and 'input/output' ('I/O' for short) metabolites for subnetwork μ, respectively.
Thus, all internal metabolites and all reactions that involve only internal and I/O metabolites are part
of the subnetwork. Note that the I/O metabolites do not themselves belong to the subnetwork, but
reactions between them, if they exist, are part of the subnetwork. Metabolites that are neither inter-
nal nor I/O for μ are referred to as external to subnetwork μ. These definitions are illustrated in
*Figure 1A*.

The main objects in this work are biochemical modules, which are a special type of subnetworks.
To define modules, I introduce some auxiliary concepts. I say that two metabolites $i$ and $j$ are con-
nected if there exists a series of enzymes that interconvert $i$ and $j$, possibly through a series of inter-
mediates. Mathematically, $i$ and $j$ are connected if there exists a simple (i.e. non-self-intersecting)
path between them. If all metabolites in this path are internal to the subnetwork μ (possibly exclud-
ing the terminal metabolites $i$ and $j$ themselves) then $i$ and $j$ are connected within the subnetwork μ,
and such path is said to lie within μ. By this definition, metabolites $i$ and $j$ can be connected within μ
only if they are either internal or I/O metabolites for μ.

### Definition 1

A subnetwork μ is called a module if (a) it has two I/O metabolites, and (b) for every internal metabo-
lite $i \in A_\mu$, there exists a simple path between the I/O metabolites that lies within μ and contains $i$.

This definition is illustrated in *Figure 1A*. The assumption that modules only have two I/O metab-
olites is not essential. However, mathematical calculations become unwieldy when the number of I/O

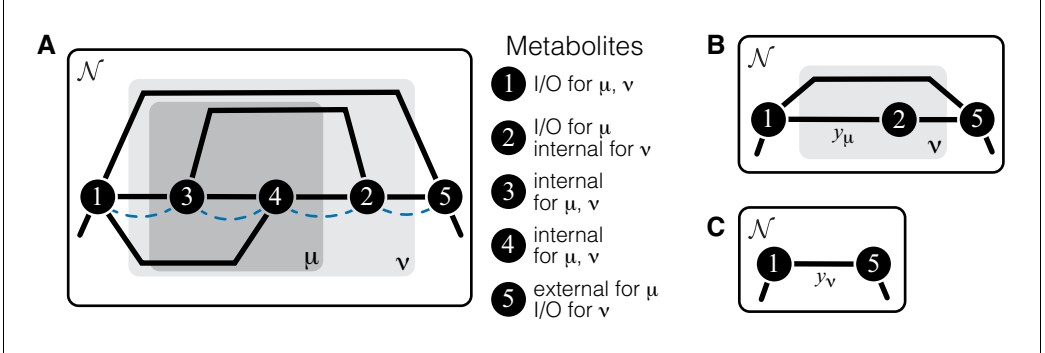

**Figure 1.** Illustration of a hierarchical metabolic network and its coarse-graining. (**A**) White rectangle represents the whole metabolic network $\mathcal{N}$. Example subnetworks μ and ν are represented by the dark and light gray rectangles. Only metabolites and reactions that belong to these subnetworks are shown; other metabolites and reactions in $\mathcal{N}$ are not shown. Metabolites 1 and 5 may be adjacent to other metabolites in $\mathcal{N}$; this fact is represented by short black lines that do not terminate in metabolites. Subnetworks μ and ν are both modules because there exists a simple path connecting their I/O metabolites that lies within μ and ν and contains all their internal metabolites (dashed blue line). (**B**) Network $\mathcal{N}$ can be coarse-grained by replacing module μ at steady state with an effective reaction between its I/O metabolites 1 and 2, with the rate constant is $y_\mu$. (**C**) Network $\mathcal{N}$ can be coarse-grained by replacing module ν at steady state with an effective reaction between its I/O metabolites 1 and 5, with the rate constant is $y_\nu$.

metabolites increases. Moreover, modules with just two I/O metabolites already capture two most salient features of metabolism: its directionality, and its complex branched topology (*Csete and Doyle, 2004*). Such modules are a natural generalization of the linear metabolic pathway which has been extensively studied in the previous literature (*Kacser and Burns, 1973*; *Szathmáry, 1993*; *Bagheri-Chaichian et al., 2003*; *MacLean, 2010*).

Modules have two important properties. First, for any given concentrations of the two I/O metabolites, all internal metabolites in the module can achieve a unique steady state which depends only on concentrations of these I/O metabolites but not on the concentrations of any other metabolites in the network (see Proposition 1 in Materials and methods). Now consider a module μ whose I/O metabolites are (without loss of generality) labeled 1 and 2 (*Figure 1A*). The second property is that, at steady state, the flux through this module is $J_\mu = y_\mu(S_1 - S_2/K_{12})$, where

$$y_\mu = F(\vec{x}_\mu) \tag{1}$$

is the effective reaction rate constant of module μ (*Figure 1B*). Importantly, $y_\mu$ depends only on the rate matrix $\vec{x}_\mu$, but not on any other rate constants (see Corollary 2 in Materials and methods), and it can be recursively computed for any module, as described in Materials and methods. In other words, metabolic network $\mathcal{N}$ can be coarse-grained by replacing module μ at steady state with a single first-order biochemical reaction with rate $y_\mu$. Importantly, such coarse-graining does not alter the dynamics of any metabolites outside of module μ (see Proposition 1 in Materials and methods). This statement is the biochemical analog of the star-mesh transformation (and its generalization, Kron reduction, *Rao et al., 2014*) well known in the theory of electric circuits (*Versfeld, 1970*). The biological interpretation of these properties is that a module is somewhat isolated from the rest of the metabolic network. And vice versa, the larger network (i.e. the cell) 'cares' only about the total rate at which the I/O metabolites are interconverted by the module but 'does not care' about the details of how this conversion is enzymatically implemented. In this sense, the effective rate $y_\mu$ quantifies the function of module μ (a macroscopic parameter) while the rates $\vec{x}_\mu$ describe the specific biochemical implementation of the module (microscopic parameters).

The effective rate constant $y_\mu$ of module μ depends on the entire rate matrix $\vec{x}_\mu$. In general, a single mutation may perturb several rate constants within a module, so that the entire shape of the function $F$ may be important. Here, I focus on a special case when each mutation perturbs one reaction (real or effective) within a module, while all others remain constant. To examine epistasis between mutations, I will also consider two different mutations that perturb two separate reactions

within a module. In these special cases, we do not need to know the entire function $F$. We only need to know how module's effective rate constant $y_\mu$ depends on the one or two rate constants of the perturbed reactions. When $y_\mu$ is considered as a function of the rate constant $\xi$ of one reaction, I write

$$y_\mu = f_1(\xi), \tag{2}$$

and when $y_\mu$ is considered as a function of the rate constants $\xi$ and $\eta$ of two reactions, I write

$$y_\mu = f_2(\xi, \eta). \tag{3}$$

The rate constants of all other reactions within module $\mu$ play a role of parameters in functions $f_1$ and $f_2$.

Consider now a network $\mathcal{N}$ that has a hierarchical structure, such that there is a series of nested modules $\mu \subset \nu \subset \cdots$, in the sense that $A_\mu \subset A_\nu \subset \cdots$ (**Figure 1A**). Since any module at steady state can be replaced with an effective first-order biochemical reaction, there exists a hierarchy of quantitative metabolic phenotypes $y_\mu, y_\nu, \ldots$ (**Figure 1B,C**). These phenotypes are of course functionally related to each other. Specifically, because $\nu$ is a 'higher-level' module (in the sense that it contains a 'lower-level' module $\mu$), the matrix $\vec{x}_\nu$ can be decomposed into two submatrices $\vec{x}_\mu$ and $\vec{x}_{\nu \setminus \mu}$ where the latter is the matrix of rate constants of reactions that belong to module $\nu$ but not to module $\mu$. Since replacing the lower-level module $\mu$ with an effective reaction with rate constant $y_\mu$ does not alter the dynamics of metabolites outside of $\mu$, $y_\nu$ must depend on all elements of $\vec{x}_\mu$ only through $y_\mu$, that is,

$$y_\nu = f_1(y_\mu), \tag{4}$$

where rates $\vec{x}_{\nu \setminus \mu}$ act as parameters of function $f_1$. Thus, in the hierarchy of metabolic phenotypes $y_\mu, y_\nu, \ldots$, a phenotype at each subsequent level depends on the phenotype at the preceding level according to **Equation 4**, and the lowest level phenotype $y_\mu$ depends on the actual rate constants accroding to **Equation 1**. This hierarchy of functionally nested phenotypes is conceptually similar to the hierarchical 'ontotype' representation of genomic data proposed recently by **Yu et al., 2016**.

## Quantification of epistasis

Consider a mutation $A$ that perturbs only one rate constant $x_{ij}$, such that the wildtype value $x_{ij}^0$ changes to $x_{ij}^A$. This mutation can be quantified at the microscopic level by its relative effect $\delta^A x_{ij} = x_{ij}^A / x_{ij}^0 - 1$. If the reaction between metabolites $i$ and $j$ belongs to nested modules $\mu, \nu, \ldots$, then mutation $A$ may impact the functions of these modules, which can be quantified by the relative effects $\delta^A y_\mu, \delta^A y_\nu$, etc. at each level of the hierarchy.

Consider now another mutation $B$ that only perturbs the rate constant $x_{k\ell}$ of another reaction. Since mutations $A$ and $B$ perturb distinct enzymes, they by definition do not genetically interact at the microscopic level. However, if both perturbed reactions belong to the metabolic module $\mu$ (and, as a consequence, to all higher-level modules which contain $\mu$), they may interact at the level of the function of this module, in the sense that the effect of mutation $B$ on the effective rate $y_\mu$ may depend on whether mutation $A$ is present or not. Such epistasis between mutations $A$ and $B$ can be quantified at the level $\mu$ of the metabolic hierarchy by a number of various epistasis coefficients (**Wagner et al., 1998**; **Hansen and Wagner, 2001**; **Mani et al., 2008**). I will quantify it with the epistasis coefficient

$$\varepsilon^{AB} y_\mu = \frac{\delta^{AB} y_\mu - \delta^A y_\mu - \delta^B y_\mu}{2 \delta^A y_\mu \delta^B y_\mu}, \tag{5}$$

where $\delta^{AB} y_\mu$ denotes the effect of the combination of mutations $A$ and $B$ on phenotype $y_\mu$ relative to the wildtype. Since I only consider two mutations $A$ and $B$, I will write $\varepsilon y_\mu$ instead of $\varepsilon^{AB} y_\mu$ to simplify notations. Note that other epistasis coefficients can always be computed from $\varepsilon y_\mu$, $\delta^A y_\mu$ and $\delta^B y_\mu$, if necessary. Expressions for epistasis coefficients at other levels of the metabolic hierarchy are analogous.

# Results

The central goal of this paper is to understand the patterns of epistasis between mutations that affect reaction rates in the hieararchical metabolic network described above. Specifically, I am interested in two questions. (1) Given that two mutations $A$ and $B$ have an epistasis coefficient $\varepsilon y_\mu$ at a lower level $\mu$ of the metabolic hierarchy, what can we say about their epistasis coefficient $\varepsilon y_\nu$ at a higher level $\nu$ of the hierarchy? In other words, how does epistasis propagate through the metabolic hierarchy? (2) If mutation $A$ only perturbs the activity $x_{ij}$ of one enzyme and mutation $B$ only perturbs the activity $x_{k\ell}$ of another enzyme that belongs to the same module $\mu$, then what values of $\varepsilon y_\mu$ can we expect to observe based on the topological relationship between the two perturbed reactions within module $\mu$? In other words, what kinds of epistasis emerge in a metabolic network?

## Propagation of epistasis through the hierarchy of metabolic phenotypes

Assuming that the effects of both individual mutations and their combined effect at the lower-level $\mu$ are small, it follows from *Equation 4* and *Equation 5* that

$$\varepsilon y_\nu = \frac{\varepsilon y_\mu}{C} + \frac{H}{2C^2} + o(1), \tag{6}$$

where $C = f_1' y_\mu / y_\nu$ and $H = f_1'' y_\mu^2 / y_\nu$ are the first- and second-order control coefficients of the lower-level module $\mu$ with respect to the flux through the higher-level module $\nu$ and $o(1)$ denotes all terms that vanish as the effects of mutations tend to zero (see Materials and methods for details). Note that *Equation 6* is a special case of a more general *Equation 49* which describes the case when mutations affect multiple enzymes. *Equation 6* defines a linear map $\phi$ with slope $1/C$ and a fixed point $\bar{\varepsilon} = -H(2C(1-C))^{-1}$, which both depend on the topology of the higher-level module $\nu$ and the rate constants $\vec{x}_{\nu\backslash\mu}$.

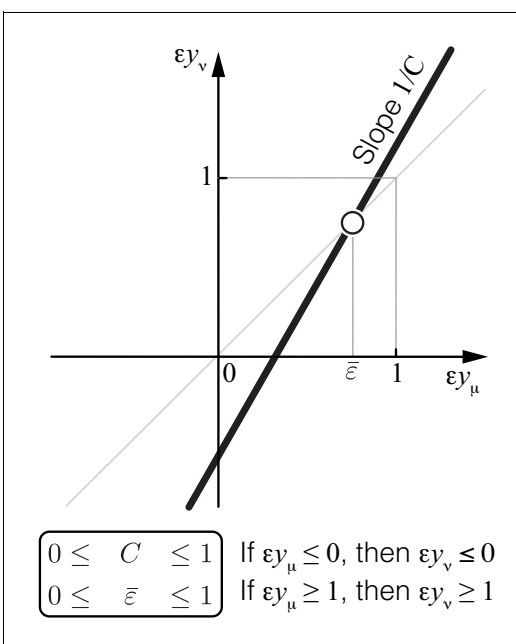

**Figure 2.** Propagation of epistasis. Properties of *Equation 6* that maps lower-level epistasis $\varepsilon y_\mu$ onto higher-level epistasis $\varepsilon y_\nu$. Slope $1/C$ and fixed point $\bar{\varepsilon}$ depend on the topology and the rate constants of the higher-level module $\nu$, but they are bounded, as shown. Thus, the fixed point $\bar{\varepsilon}$ of this map lies between 0 and 1 and is always unstable (open circle).

To gain some intuition for how the map $\phi$ governs the propagation of epistasis from a lower level $\mu$ to a higher level $\nu$, suppose that module $\nu$ is a linear metabolic pathway. In this case, it is intuitively clear that function $f_1$ is monotonically increasing (i.e. the higher $y_\mu$, the more flux can pass through the linear pathway $\nu$) and concave (i.e. as $y_\mu$ grows, other reactions in $\nu$ become increasingly more limiting, such that further gains in $y_\mu$ yield smaller gains in $y_\nu$). Indeed, it is easy to show that $C = (1 + \alpha y_\mu)^{-1} > 0$ and $H = -2\alpha y_\mu (1 + \alpha y_\mu)^{-2} < 0$, where $\alpha$ is a positive constant that depends on other reactions in the pathway (see Materials and methods for details). It then immediately follows that any zero or negative epistasis $\varepsilon y_\mu$ that already arose at the lower level would propagate to negative epistasis $\varepsilon y_\nu$ at the level of the linear pathway $\nu$. Moreover, since $C < 1$, the fixed point of the map in *Equation 6* is unstable. Therefore, if epistasis $\varepsilon y_\mu$ was already sufficiently large at the lower level, it would induce even larger positive epistasis $\varepsilon y_\nu$ at the level of the linear pathway $\nu$. In fact, when module $\nu$ is a linear pathway, $\bar{\varepsilon} = 1$, so that $\varepsilon y_\nu > 1$ whenever $\varepsilon y_\mu > 1$.

The first result of this paper is the following theorem, which shows that the same rules of propagation of epistasis hold not only for a linear pathway but for any module (*Figure 2*).

## Theorem 1

For any module $\nu$,

$$0 \leq C \leq 1$$

(7)

and

$$0 \leq \bar{\varepsilon} \leq 1.$$

(8)

The proof of Theorem 1 is given in Materials and methods. Its main idea is the following. The functional form of $f_1$ in *Equation 4* depends on the topology of module $\nu$. Since the number of topologies of $\nu$ is infinite, we might a priori expect that there is also an infinite number of functional forms of $f_1$. However, this is not the case. In fact, all higher-level modules that contain a lower-level module fall into three topological classes defined by the location of the lower-level module with respect to the I/O metabolites of the higher-level module (see Proposition 2 and Figure 7 in Materials and methods). To each topological class corresponds a parametric family of the function $f_1$, so that there are only three such families. For each family, the values of $C$ and $H$ can be explicitly calculated, yielding the bounds in *Equation 7* and *Equation 8*.

*Equation 6* together with *Equation 7* and *Equation 8* show that the linear map $\phi$ from epistasis at a lower-level to epistasis at the higher-level has an unstable fixed point between 0 and 1 (*Figure 2*). This implies that negative epistasis at a lower level of the metabolic hierarchy necessarily induces negative epistasis of larger magnitude at the next level of the hierarchy, that is, $\varepsilon y_\nu \leq \varepsilon y_\mu < 0$. Therefore, once negative epistasis emerges somewhere along the hierarchy, it will induce negative epistasis at all higher levels of the hierarchy, irrespectively of the topology or the kinetic parameters of the network.

Similarly, if epistasis at the lower level of the metabolic hierarchy is positive and strong, $\varepsilon y_\mu > 1$, it will induce even stronger positive epistasis at the next level of the hierarchy, that is, $\varepsilon y_\nu \geq \varepsilon y_\mu > 1$. Therefore, once strong positive epistasis emerges somewhere in the metabolic hierarchy, it will induce strong positive epistasis of larger magnitude at all higher levels of the hierarchy, irrespectively of the topology or the kinetic parameters of the network. If positive epistasis at a lower level of the hierarchy is weak, $0 < \varepsilon y_\mu < 1$, it could induce either negative, weak positive or strong positive epistasis at the higher level of the hierarchy, depending on the precise value of $\varepsilon y_\mu$, the topology of the higher-level module $\nu$ and the microscopic rate constants $\vec{x}_{\nu \setminus \mu}$.

In summary, there are three regimes of how epistasis propagates through a hierarchical metabolic network. Negative and strong positive epistasis propagate robustly irrespectively of the topology and kinetic parameters of the metabolic network, whereas the propagation of weakly positive epistasis depends on these details. The strongest qualitative prediction that follows from Theorem 1 is that negative epistasis for a lower-level phenotype cannot turn into positive epistasis for a higher-level phenotype, but the converse is possible.

## Emergence of epistasis between mutations affecting different enzymes

Which of the three regimes described above can emerge in metabolic networks and under what circumstances? In other words, if two mutations affect the same module, are there any constraints on epistasis that might arise at the level of the effective rate constant of this module? To address this question, I consider two mutations $A$ and $B$ that affect the rate constants of different single reactions within a given module.

Consider a relatively simple module $\nu$ shown in *Figure 1A* and two mutations $A$ and $B$ that affect the reactions, as shown in *Figure 3A*. I will now show that the epistasis coefficient $\varepsilon y_\nu$ can take values in all three domains described above, depending on the biochemical details of this module. Using the recursive procedure for evaluating $y_\mu$ described in Materials and methods, it is straightforward to obtain an analytical expression for $y_\nu$ as a function of the rate matrix $\vec{x}_\nu$, from which $\varepsilon y_\mu$ can also be obtained (see Materials and methods for details). To demonstrate that $\varepsilon y_\mu$ can take values below 0, between 0 and 1, and above 1, it is convenient to keep all of the rate constants fixed except for the rate constant $z \equiv x_{34}$ of a reaction that is not affected by mutations $A$ or $B$, as shown in *Figure 3A*. *Figure 3B* then shows how the epistasis coefficient $\varepsilon y_\mu$ varies as a function of $z$ for one

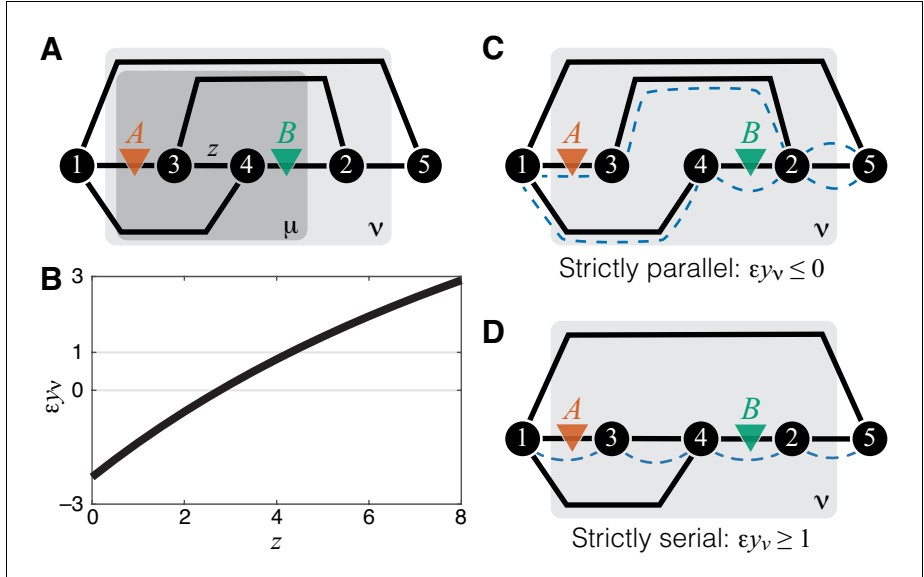

**Figure 3.** Emergence of epistasis and its dependence on the topological relationship between the reactions affected by mutations. (A) An example of a simple module $\nu$ (same as in *Figure 1A*) where negative, weak positive and strong positive epistasis can emerge between two mutations $A$ and $B$. (B) Epistasis between mutations $A$ and $B$ at the level of module $\nu$ depicted in (A) as a function of the rate constant $z$ of a third reaction. The values of other parameters of the network are given in Materials and Methods. (C) An example of a simple module where reactions affected by mutations are strictly parallel. In such cases, epistasis for the effective rate constant $y_\nu$ is non-positive. Dashed blue lines highlight paths that connect the I/O metabolites and each contain only one of the affected reactions. (D) An example of a simple module where reactions affected by mutations are strictly serial. In such cases, epistasis for the effective rate constant $y_\nu$ is equal to or greater than 1 (i.e. strongly positive). Dashed blue line highlights a path that connects the I/O metabolites and contains both affected reactions.

particular choice of all other rate constants. When $z$ is small, $\varepsilon y_\mu < 0$. As $z$ increases, it becomes weakly positive ($0 < \varepsilon y_\mu < 1$) and eventually strongly positive ($\varepsilon y_\mu > 1$). Thus, in my model, there are no fundamental constraints on the types of epistasis that can emerge between mutations.

This simple example also reveals that not only the value but also the sign of epistasis generically depend on the rates of other reactions in the network, such that other mutations or physiological changes in enzyme expression levels can modulate epistasis sign and strength. In other words, 'higher-order' and 'environmental' epistasis are generic features of metabolic networks.

Upon closer examination, the toy example in *Figure 3* also suggests that the sign of $\varepsilon y_\nu$ may depend predictably on the topological relationship between the affected reactions. When $z = 0$, the two reactions affected by mutations are parallel, and epistasis is negative. When $z$ is very large, most of the flux between the I/O metabolites passes through $z$ such that the two reactions affected by mutations become effectively serial, and epistasis is strongly positive. Other toy models show consistent results: epistasis between mutations affecting different reactions in a linear pathway is always positive and epistasis between mutations affecting parallel reactions is negative (see Materials and methods for details). These observation suggest an interesting conjecture. Do mutations affecting parallel reactions always exhibit negative epistasis and do mutations affecting serial reactions always exhibit positive epistasis? In fact, such relationship between sign of epistasis and topology has been previously suggested in the literature (e.g. *Dixon et al., 2009*; *Lehner, 2011*).

To formalize and mathematically prove this hypothesis, I first define two reactions as *parallel* within a given module if there exist at least two distinct simple (i.e. non-self-intersecting) paths that connect the I/O metabolites, such that each path lies within the module and contains only one of the two focal reactions. Analogously, two reactions are *serial* within a given module if there exists at least one simple path that connects the I/O metabolites, lies within the module and contains both focal reactions.

According to these definitions, two reactions can be simultaneously parallel and serial, as, for example, the reactions affected by mutations *A* and *B* in *Figure 3A*. I call such reaction pairs *serial-parallel*. I define two reactions to be *strictly parallel* if they are parallel but not serial (*Figure 3C*) and I define two reactions to be *strictly serial* if they are serial but not parallel (*Figure 3D*). Thus, each pair of reactions within a module can be classified as either strictly parallel, strictly serial or serial-parallel.

The second result of this paper is the following theorem.

## Theorem 2

Let $\xi$ and $\eta$ be the rate constants of two different reactions in module $\mu$. Suppose that mutation *A* perturbs only one of these reactions by $\delta^A \xi$ and mutation *B* perturbs only the other reaction by $\delta^B \eta$. In the limit $\delta^A \xi \to 0$ and $\delta^B \eta \to 0$, the following statements are true. If the affected reactions are strictly parallel then $\varepsilon y_\mu \leq 0$. If the affected reactions are strictly serial, then $\varepsilon y_\mu \geq 1$.

The detailed proof of this theorem is given in Materials and methods. Its key ideas and the logic are the following. It follows from *Equation 3* and *Equation 5* that

$$\varepsilon y_\mu = \frac{H_{\xi\eta}}{2\, C_\xi\, C_\eta} + o(1), \tag{9}$$

where $C_\xi = \frac{\partial f_2}{\partial \xi} \frac{\xi}{y_\mu}$, $C_\eta = \frac{\partial f_2}{\partial \eta} \frac{\eta}{y_\mu}$, $H_{\xi\eta} = \frac{\partial^2 f_2}{\partial \xi \partial \eta} \frac{\xi \eta}{y_\mu}$ are the first- and second-order control coefficients of the affected reactions with respect to the flux through module $\mu$ and $o(1)$ denotes terms that vanish when $\delta^A \xi$ and $\delta^B \eta$ approach zero (see Materials and methods for details). Note that *Equation 9* was previously derived by *Chiu et al., 2012*.

To compute the epistasis coefficient $\varepsilon y_\mu$ for an arbitrary module $\mu$, we need to know the first and second derivatives of function $f_2$. Analogous to function $f_1$, there is a finite number of parametric families to which $f_2$ can belong. Specifically, all modules fall into nine topological classes with respect to the locations of the affected reactions within the module (see Figure 8), and each of these topologies defines a parametric family of function $f_2$ (see Proposition 3 and its Corollary 3 in Materials and methods). Most of these topological classes are broad and contain modules where the affected reactions are strictly parallel, those where they are strictly serial as well as those where they are serial-parallel. And it is easy to show that not all members of each topological class have the same sign of $\varepsilon y_\mu$. However, modules from the same topological class where the affected reactions are strictly parallel or strictly serial fall into a finite number of topological sub-classes (see Figure 10 through Figure 14, Table 2 and Table 3). Overall, there are only 17 distinct topologies where the affected reactions are strictly parallel (Table 2), which define 17 parametric sub-families of function $f_2$. For all members of these sub-families, *Equation 9* yields $\varepsilon y_\mu \leq 0$ (see Proposition 7 in Materials and methods). Similarly, there are only 11 distinct topologies where the affected reactions are strictly serial (Table 3), which define 11 parametric sub-families of function $f_2$. For all members of these sub-families, *Equation 9* yields $\varepsilon y_\mu \geq 1$ (see Proposition 8 in Materials and methods).

The results of Theorem 1 and Theorem 2 together imply that the topological relationship at the microscopic level between two reactions affected by mutations constrains the values of their epistasis coefficient at all higher phenotypic levels. Specifically, if negative epistasis is detected at any phenotypic level, the affected reactions cannot be strictly serial. And conversely, if strong positive epistasis is detected at any phenotypic level, the affected reactions cannot be strictly parallel. In this model, weak positive epistasis in the absence of any additional information does not imply any specific topological relationship between the affected reactions.

## Sensitivity of results with respect to the magnitude of mutational effects

Both Theorem 1 and Theorem 2 strictly hold only when the effects of both mutations are infinitesimal. Next, I investigate how these results might change when the mutational effects are finite.

## Propagation of epistasis between mutations with finite effect sizes

As mentioned above and discussed in detail in Materials and methods, all higher-level modules that contain a lower-level module fall into three topological classes, which I label $\mathcal{M}^{\mathrm{b}}$, $\mathcal{M}^{\mathrm{io}}$ and $\mathcal{M}^{\mathrm{i}}$,

depending on the location of the lower-level module within the higher-level module (see Figure 7). The topological class specifies the parametric family of the function $f_1$ which maps the effective rate constant $y_\mu$ onto the effective rate constant $y_\nu$ (see *Equation 4*). For all modules from the topological class $\mathcal{M}^b$, function $f_1$ is linear (see *Equation 29*), which implies that the results of Theorem 1 hold exactly even when the effects of mutations are finite. For modules from the topological classes $\mathcal{M}^{io}$ and $\mathcal{M}^i$, function $f_1$ is hyperbolic (see *Equation 30* and *Equation 31*), so that the results of Theorem 1 may not hold when the effects of mutations are finite. To test the validity of Theorem 1 in these cases, I calculated the non-linear function $\phi$ that maps the epistasis coefficient $\varepsilon y_\mu$ onto the epistasis coefficient $\varepsilon y_\nu$ for 1000 randomly generated modules from each of the two topological classes and for mutations that increase or decrease the effective rate constant of the lower-level module $y_\mu$ by up to 50% (see Materials and methods for details).

The validity of Theorem 1 depended on the sign of mutational effects. When at least one of the two mutations had a negative effect on $y_\mu$, map $\phi$ had the same properties as described in Theorem 1, even for mutations with large effect, that is, it had a fixed point $\bar{\varepsilon}$ in the interval $[0, 1]$ and this fixed point was unstable. When the effects of both mutations on $y_\mu$ were positive and small, these results also held in about 82% of sampled modules (see *Figure 4A*, *Figure 4—figure supplement 1*, *Figure 4—figure supplement 2*). In the remaining ~18% of sampled modules, the fixed point $\bar{\varepsilon}$ shifted slightly above 1. As the magnitude of mutational effects increased, the fraction of sampled modules with $\bar{\varepsilon} > 1$ grew, reaching 42% when both mutations increased $y_\mu$ by 50%. In most of these cases, $\bar{\varepsilon}$ remained below 2, and I found only one module with $\bar{\varepsilon} > 4$ (*Figure 4A*, *Figure 4—figure supplement 1*, *Figure 4—figure supplement 2*). Whenever the fixed point existed, it was unstable, with the exception of 12 modules for which $\phi$ was very close to the identity map. For 289 modules (14.5%), the fixed point disappeared when both mutations increased $y_\mu$ by 50%. In all these cases, $\varepsilon y_\nu < \varepsilon y_\mu$, indicating that even large positive epistasis may decline as it propagates through the metabolic hierarchy when the effects of mutations are finite.

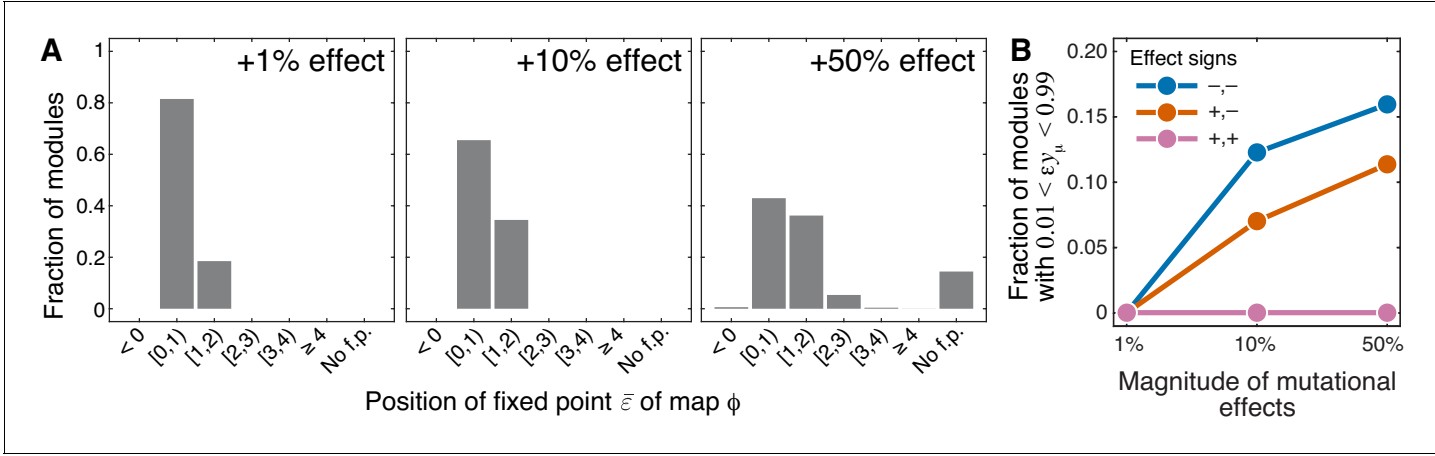

**Figure 4.** Sensitivity of results of Theorem 1 and Theorem 2 with respect to the magnitude of mutational effects. (**A**) Distribution of the position of the fixed point $\bar{\varepsilon}$ of the function $\phi$ that maps lower-level epistasis $\varepsilon y_\mu$ onto higher-level epistasis $\varepsilon y_\nu$ in modules with random parameters and for mutations with positive effects on $y_\mu$ (see text and Materials and methods for details). All cases are shown in *Figure 4—figure supplement 1* and *Figure 4—figure supplement 2*. The effect size of both mutations is indicated on each panel. 'No f.p'. indicates that no fixed point exists. (**B**) Fraction of sampled modules (averaged across generating topologies) where mutations affect strictly serial reactions but the epistasis coefficient is less than 1, contrary to the statement of Theorem 2 (see text and Materials and methods for details). All cases stratified by generating topology are shown in *Figure 4—figure supplement 3*.

The online version of this article includes the following figure supplement(s) for figure 4:

**Figure supplement 1.** Distribution of the position of the fixed point in 1000 modules from the topological class $\mathcal{M}^{io}$ with random parameters.

**Figure supplement 2.** Distribution of the position of the fixed point in 1000 modules from the topological class $\mathcal{M}^i$ with random parameters.

**Figure supplement 3.** Fraction of sampled modules with different strictly serial generating topologies where the epistasis coefficient falls between 0.01 and 0.99.

## Emergence of epistasis between mutations with finite effect sizes

As mentioned above and discussed in detail in Materials and methods, modules where the reactions affected by mutations are strictly parallel fall into 17 topological classes (see Table 2) and modules where the reactions affected by mutations are strictly serial fall into 11 topological classes (see Table 3). The topological class specifies the parametric family of the function $f_2$ which maps the rate constants $\xi$ and $\eta$ of the affected reactions onto the effective rate constant $y_\mu$. To test how well Theorem 2 holds when the effects of mutations are finite, I calculated $\varepsilon y_\mu$ for randomly generated modules from these topological classes and for mutations increasing or decreasing $\xi$ and $\eta$ by up to 50% (see Materials and methods for details).

The validity of Theorem 2 depended most strongly on the topological relationship between the reaction affected by mutations. Whenever the affected reactions were strictly parallel, the epistasis coefficient at the level of module $\mu$ was always less than or equal to zero, even when mutations perturbed the rate constants by as much as 50%, consistent with Theorem 2. This was also true for strictly serial reactions, as long as both mutations had positive effects. When the affected reaction were strictly serial and at least one of the mutations had a negative effect, the epistasis coefficient was always positive, but in some cases it was less than 1 (see *Figure 4B*, *Figure 4—figure supplement 3*), in disagreement with Theorem 2. This indicates that when the effects of mutations are not infinitesimal, even mutations that affect strictly serial reactions can potentially produce negative epistasis for higher-level phenotypes.

Taken together, these results suggest that both Theorem 1 and Theorem 2 extend reasonably well, but not perfectly, to mutations with finite effect sizes. The domains of validity of both theorems appear to depend on the sign of mutational effects. The way in which the theorems break down as their assumptions are violated appears to be stereotypical: when the mutational effects increase, more types of mutations produce weak epistasis, and the bias toward negative epistasis increases during propagation from lower to higher levels of the metabolic hierarchy.

## Beyond first-order kinetics: epistasis in a kinetic model of glycolysis

The results of previous sections revealed a relationship between network topology and the ensuing epistasis coefficients in an analytically tractable model. However, the assumptions of this model are most certainly violated in many realistic situations. It is therefore important to know whether the same or similar rules of epistasis emergence and propagation hold beyond the scope of this model. I address this question here by analyzing a computational kinetic model of glycolysis developed by *Chassagnole et al., 2002*. This model keeps track of the concentrations of 17 metabolites, reactions between which are catalyzed by 18 enzymes (*Figure 5A* and *Figure 5—figure supplement 1*; see Materials and methods for details). This model falls far outside of the analytical framework introduced in this paper: some reactions are second-order, reaction kinetics are non-linear, and in several cases the reaction rates are modulated by other metabolites (*Chassagnole et al., 2002*).

Testing the predictions of the analytical theory in this computational model faces two complications. First, in a non-linear model, modules are no longer fully characterized by their effective rate constants, even at steady state. Instead, each module is described by the flux between its I/O metabolites which non-linearly depends on the concentrations of these metabolites. Consequently, the effects of mutations and epistasis coefficients also become functions of the I/O metabolite concentrations. An epistasis coefficient at the level of module $\nu$ can still be evaluated according to *Equation 5*, with $y_\nu$ now representing the flux through module $\nu$ evaluated at a particular concentration of the I/O metabolites. For simplicity, I computationally find the steady state of the full glycolysis network and evaluate the epistasis coefficients only at this steady state, that is, for each module, I keep the concentrations of the I/O metabolites fixed at their steady-state values for the full network (see Materials and methods for details).

The second complication is that some control coefficients are so small that they fall below the threshold of numerical precision. Perturbing such reactions has no detectable effect on flux (*Figure 5—figure supplement 2*). In the analysis that follows, I ignore such reactions because the epistasis coefficient defined by *Equation 5* can only be computed for mutations with non-zero effects on flux. In addition, the control coefficients of some reactions are negative, which implies that an increase in the rate of such reaction decreases the flux through the module (*Figure 5—figure supplement 2*). I also ignore such reactions because there is no analog for them in the analytical theory

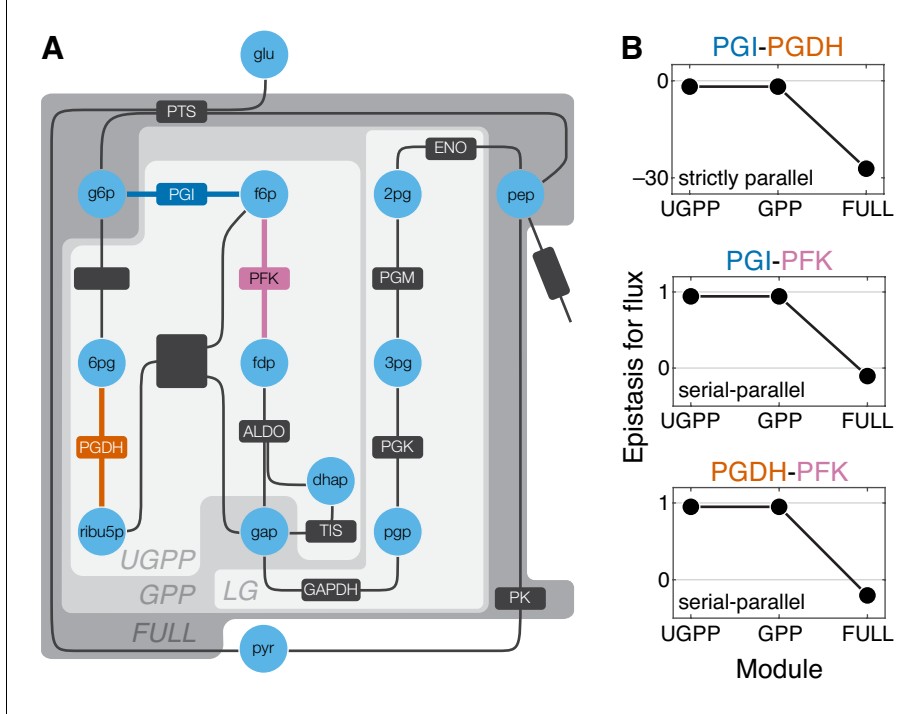

**Figure 5.** Epistasis in a kinetic model of *Escherichia coli* glycolysis. (**A**) Simplified schematic of the model (see *Figure 5—figure supplement 1* for details). Different shades of gray in the background highlight four modules as indicated (see text). Light blue circles represent metabolites. Reactions are shown as lines with dark gray boxes. The enzymes catalyzing reactions whose control coefficients with respect to the flux through the module are positive are named; other enzyme names are ommitted for clarity (see *Table 5* and *Table 6* for abbreviations). Three reactions, catalyzed by PGI, PFK, PGDH, for which the epistasis coefficients are shown in panel B are highlighted in dark blue, red, and orange, respectively. (**B**) Epistasis coefficients for flux through each module between mutations perturbing the respective reactions, computed at steady state (see text and Materials and methods for details). Reactions catalyzed by PGI and PGDH are strictly parallel (path g6p-f6p-fdp-gap contains only PGI, path g6p-6pg-ribu5p-gap contains only PGDH and there is no simple path in UGPP between g6p and gap that contains both PGI and PGDH). Reactions catalyzed by PGI and PFK are serial-parallel (path g6p-f6p-fdp-gap contains both reactions, path g6p-f6p-gap contains only PGI, path g6p-6pg-ribu5p-f6p-fdp-gap contains only PFK). Reactions catalyzed by PFK and PGDH are also serial-parallel (path g6p-6pg-ribu5p-f6p-fdp-gap contains both reactions, path g6p-f6p-fdp-gap contains only PFK, path g6p-6pg-ribu5p-gap contains only PGDH).

The online version of this article includes the following figure supplement(s) for figure 5:

**Figure supplement 1.** Detailed schematic of the kinetic model of glycolysis.

**Figure supplement 2.** Control coefficients for the output flux in the FULL module.

**Figure supplement 3.** Control and epistasis coefficients for the fluxes through multiple sub-modules within glycolysis.

presented above. After excluding seven reactions for these reasons, I examine epistasis in 55 pairs of mutations that affect the remaining 11 reactions.

The glycolysis network shown in *Figure 5A* (see also *Figure 5—figure supplement 1*) can be naturally partitioned into four modules which I name 'LG' (lower glycolysis), 'UGPP' (upper glycolysis and pentose phosphate), 'GPP' (glycolysis and pentose phosophate), and 'FULL'. Modules LG and UGPP are non-overlappng and both of them are nested in module GPP which in turn is nested in the FULL module. Thus, at least for some reaction pairs it is possible to calculate epistasis coefficients at three levels of metabolic hierarchy. There are three such pairs, and the results for them are shown in *Figure 5B*. Epistasis for the remaining pairs of reactions can be evaluated only at one or two levels of the hierarchy because these reactions belong to different modules at the lowest levels or because their individual effects are too small. The results for all reaction pairs are shown in *Figure 5—figure supplement 3*.

The strongest qualitative prediction of the analytical theory described above is that negative epistasis for a lower-level phenotype cannot turn into positive epistasis for a higher-level phenotype, while the converse is possible. *Figure 5B* and *Figure 5—figure supplement 3* show that the data are consistent with this prediction. Another prediction is that epistasis between strictly parallel reactions should be negative. There is only one pair of reactions that are strictly parallel, those catalyzed by glucose-6-phosphate isomerase (PGI) and 6-phosphogluconate dehydrogenase (PGDH), and indeed the epistasis coefficients between mutations affecting these reactions are negative at all levels of the hierarchy (*Figure 5B*). Finally, the analytical theory predicts that mutations affecting strictly serial reactions should exhibit strong positive epistasis. There are 36 reaction pairs that are strictly serial. Epistasis is positive between mutations in 33 of them, and it is strongly positive in 17 of them (*Figure 5—figure supplement 3*). Three pairs of strictly serial reactions (those where one reaction is catalyzed by PK and the other is catalyzed by PGI, PGDH, or PFK) exhibit negative epistasis (*Figure 5—figure supplement 3*). These results suggest that, although one may not be able to naively extrapolate the rules of emergence and propagation of epistasis derived in the simple analytical model to more complex networks, some generalized versions of these rules may nevertheless hold more broadly.

## Discussion

Genetic interactions are a powerful tool in genetics, and they play an important role in evolution. Yet, how epistasis emerges from the molecular architecture of the cell and how it propagates to higher-level phenotypes, such as fitness, remains largely unknown. Several recent studies made a statistical argument that the structure of the fitness landscape (and, as a consequence, the epistatic interactions between mutations at the level of fitness) may be largely independent of the underlying molecular architecture of the organism (*Martin, 2014*; *Lyons et al., 2020*; *Reddy and Desai, 2020*). If mutations are typically highly pleiotropic (i.e. affect many independent phenotypes relevant for fitness) or are engaged in a large number of idiosyncratic epistatic interactions with other mutations in the genome, the resulting fitness landscapes converge to certain limiting shapes, such as the Fisher's geometric model (*Martin, 2014*; *Tenaillon, 2014*). To what extent these arguments indeed apply in practice is unclear. But if they do, most genetic interactions detected at the fitness level may be uninformative about the architecture of the underlying biological networks.

In this paper, I took a 'mechanistic' approach, which is in a sense orthogonal to the statistical one. In my model of a hierarchical metabolic network, mutations are highly pleiotropic (a mutation in any enzyme affects all the fluxes in the module) and highly epistatic (a mutation in any enzyme interacts with mutations in any other enzyme). Yet, these pleiotropic and epistatic effects appear to be sufficiently structured that some information about the topology of the network is preserved through all levels of the hierarchy. Indeed, the emergence and propagation of epistasis follow two simple rules in my model. First, once epistasis emerges at some level of the hierarchy, its propagation through the higher levels of the hierarchy depends weakly on the details of the network. Specifically, negative epistasis at a lower level induces negative epistasis at all higher levels and strong positive epistasis induces strong positive epistasis at all higher levels, irrespectively of the topology or the kinetic parameters of the network. Second, what type of epistasis emerges in the first place depends on the topological relationship between the reactions affected by mutations. In particular, negative epistasis emerges between mutations that affect strictly parallel reactions and positive epistasis emerges between mutations that affect strictly serial reactions. Insofar as my model is relevant to nature, the key conclusion from it is that epistasis at high-level phenotypes carries some, albeit incomplete, information about the underlying topological relationship between the affected reactions.

These results have implications for the interpretation of empirically measured epistasis coefficients. It is often assumed that a positive epistasis coefficient between mutations that affect distinct genes signals that their gene products act in some sense serially, whereas a negative epistasis coefficient is a signal of genetic redundancy, that is, a parallel relationship between gene products (*Dixon et al., 2009*). My results suggest that this reasoning is generally correct, but that the relationship between epistasis and topology is more nuanced. In particular, the sign of the epistasis coefficient in my model constrains but does not uniquely specify the topological relationship, such that a negative epistasis coefficient implies that the affected reactions are not strictly serial (but may or may not be strictly parallel) and an epistasis coefficient exceeding unity excludes a strictly parallel

relationship (but does not necessarily imply a strictly serial relationship). My model suggests that one should also be careful with inferences going in the other direction, that is, extrapolating the patterns of epistasis measured at the biochemical level to those at the level of fitness. For example, if one wishes to infer the past evolutionary trajectory of an enzyme and finds two amino acid changes that exhibit a positive interaction at the level of enzymatic activity, it does not automatically imply that these mutations will exhibit a positive interaction at the level of fitness.

The strongest results presented here rely on several assumptions. I proved Theorem 1 and Theorem 2 in the limit of vanishingly small mutational effects. Some results of the metabolic control analysis, notably the summation theorem, are sensitive to this assumption (*Bagheri-Chaichian et al., 2003*; *Bagheri and Wagner, 2004*). To test the sensitivity of my analytical results with respect to this assumption, I used numerical simulations of networks with randomly sampled kinetic parameters and found that the results hold reasonably well when the effects of mutations are not infinitesimal.

The most restrictive assumption in the present work is that of first-order kinetics. Networks with only first-order kinetics clearly fail to capture some biologically important phenomena, such as sign epistasis (*Weinreich et al., 2005*; *Chou et al., 2014*; *Ewald et al., 2017*; *Kemble et al., 2020*). I discuss possible ways to relax this assumption below. But at present, a major question remains whether the rules of epistasis and propagation described here hold for realistic biological networks and whether they can be directly used to interpret empirical epistasis coefficients. My analysis of a fairly realistic computational model of glycolysis cautions against overinterpreting empirical epistasis coefficients using the rules derived here. But it also suggests that more general rules of propagation and emergence of epistasis may be found for more realistic networks. Thus, the simple rules derived here should probably be thought of as null expectations.

Relaxing the first-order kinetics assumption is analytically challenging because it is critical for replacing a module with a single effective reaction without altering the dynamics of the rest of the network. Although such lossless replacement is almost certainly not possible in networks with more complex kinetics, advanced network coarse-graining techniques may offer a promising way forward (*Rao et al., 2014*). Flux balance analysis (FBA) is an alternative approach (*Orth et al., 2010*). FBA is appealing because it entirely removes the dependence of the model on reaction kinetics. However, this comes at a substantial cost. In FBA models, fitness and other high-level phenotypes become independent of the internal kinetic parameters, which is clearly unrealistic. Nevertheless, FBA is often very good at capturing the effects of mutations that change the topology of metabolic networks, such as reaction additions and deletions (reviewed in *Gu et al., 2019*). At the same time, there is no natural way within FBA to model mutations that perturb reaction kinetics (*He et al., 2010*; *Alzoubi et al., 2019*). In short, FBA and my approach are complementary (see Appendix 5 for a more detailed discussion).

## Generic properties of epistasis in biological systems

Simple models help us identify generic phenomena—those that are shared by a large class of systems—which should inform our 'null' expectations in empirical studies. Deviations from such null in a given system under examination inform us about potentially interesting peculiarities of this system. The model presented here suggests several generic features of epistasis between genome-wide mutations.

### Epistasis has two contributions

My analysis shows that the value of an epistasis coefficient measured for a higher level phenotype is a result of two contributions (*Domingo et al., 2019*), propagation and emergence, which correspond to two terms in *Equation 6* (or the more general *Equation 49*). The first term, propagation, shows that if two mutations exhibit epistasis for a lower-level phenotype they also generally exhibit epistasis for a higher-level phenotype. The second contribution comes from the fact that lower-level phenotypes map onto higher-level phenotypes via non-linear functions. This is true even in a simple model with linear kinetics considered here. As a result, even if two mutations exhibit no epistasis at the lower-level phenotype, epistasis must emerge for the higher-level phenotype, as previously pointed out by multiple authors (e.g. *Kacser and Burns, 1981*; *DePristo et al., 2005*; *Martin et al., 2007*; *Chiu et al., 2012*; *Otwinowski et al., 2018*; *Domingo et al., 2019*; *Husain and Murugan, 2020*).

## Epistasis depends on the genetic background and environment

My analysis shows that the value of an epistasis coefficient for a particular pair of mutations is in large part determined by the topological relationship between reactions affected by them. Since the topology of the metabolic network itself depends on the genotype (which genes are present in the genome) and on the environment (which enzymes are active or not), the topological relationship between two specific reactions might change if, for example, a third mutation knocks out another enzyme or if an enzyme is up- or down-regulated due to an environmental change (see *Figure 3*). Thus, we should generically expect epistasis between mutations to depend on the environment and on the presence or absence of other mutations in the genome. In other words, $G \times G \times G$ interactions (higher-oder epistasis) and $G \times G \times E$ interactions (environmental epistasis) should be common (*Snitkin and Segrè, 2011*; *Flynn et al., 2013*; *Lindsey et al., 2013*; *Taylor and Ehrenreich, 2015*; *Sailer and Harms, 2017a*). This fact complicates the interpretation of inter-gene epistasis since mutations in the same pair of genes can exhibit qualitatively different genetic interactions in different strains, organisms and environments, as has been observed (*St Onge et al., 2007*; *Musso et al., 2008*; *Tischler et al., 2008*; *Dowell et al., 2010*; *Heigwer et al., 2018*; *Li et al., 2019*). However, the situation may not be hopeless because the topological relationship between two reactions cannot change arbitrarily after addition or removal of a single reaction. For example, if two reactions are strictly parallel, removing a third reaction does not alter their relationship (see Proposition 5). Thus, comparing matrices of epistasis coefficients measured in different environments or genetic backgrounds could inform us about how the organism rewires its metabolic network in response to these perturbations (*St Onge et al., 2007*; *Musso et al., 2008*; *Heigwer et al., 2018*; *Li et al., 2019*).

## Skew in the distribution of epistasis coefficients

Studies that measure epistasis for fitness-related phenotypes among genome-wide mutations usually find both positive and negative epistases, but the preponderance of positive and negative epistasis varies. Some authors reported a skew toward positive interactions among deleterious mutations (*Jasnos and Korona, 2007*; *He et al., 2010*; *Johnson et al., 2019*), whereas others reported a skew toward negative interactions (*Szappanos et al., 2011*; *Costanzo et al., 2016*). Beneficial mutations appear to predominantly exhibit negative epistasis, also known as 'diminishing returns' epistasis (e.g. *Martin et al., 2007*; *Khan et al., 2011*; *Chou et al., 2011*; *Kryazhimskiy et al., 2014*; *Schoustra et al., 2016*). The reasons for these patterns are currently unclear. Several recent theoretical papers offer possible statistical explanations for them (*Martin, 2014*; *Lyons et al., 2020*; *Reddy and Desai, 2020*). On the other hand, mechanistic predictions for the distribution of epistasis coefficients are not yet available (but see *Sanjuán and Nebot, 2008*; *Macía et al., 2012*; *Chiu et al., 2012*). The present work does not directly address this problem either, but it provides some additional clues.

First, my model shows that the sign of epistasis at least to some extent reflects the topology of the network. Thus, the distribution of epistasis coefficients at high-level phenotypes in real organisms should ultimately depend on the preponderance of different topological relationships between the edges in biological networks. It then seems a priori unlikely that positive and negative interactions would be exactly balanced. Thus, we should expect the distribution of epistasis coefficients to be skewed in one or another direction.

The second observation is that in the metabolic model considered here a positive epistasis coefficient at one level of the hierarchy can turn into a negative one at a higher level, but the reverse is not possible. This bias toward negative epistasis at higher-level phenotypes appears to be even stronger in networks with saturating kinetics (*Figure 5* and *Figure 5—figure supplement 3*).

The third observation is that epistasis among beneficial and deleterious mutations affecting metabolic genes should be identical at the level where they arise, provided that beneficial and deleterious mutations are identically distributed among metabolic reactions. Thus, a stronger skew toward negative epistasis among beneficial mutations at the level of fitness could arise in my model for two mutually non-exclusive reasons. One possibility is that beneficial mutations tend to affect certain special subsets of genes, those that predominantly give rise to negative epistasis. For example, beneficial mutations may for some reason predominantly arise in enzymes that catalyze strictly parallel reactions. Another possibility is that when epistasis between beneficial mutations propagates

through the metabolic hierarchy it tends to exhibit a stronger negative bias compared to epistasis between deleterious mutations. Indeed, this phenomenon arises in my model among mutations with large effect (see *Figure 4A*, *Figure 4—figure supplement 1* and *Figure 4—figure supplement 2*).

## Epistasis is generic

Perhaps the most important—and also the most intuitive—conclusion of this work is that we should expect epistasis for high-level phenotypes, such as fitness, to be extremely common. Consider first a unicellular organism growing exponentially. Its fitness is fully determined by its growth rate, which can be thought of as the rate constant of an effective biochemical reaction that converts external nutrients into cells (see Appendix 6 for a simple mathematical model of this statement). In other words, growth rate is the most coarse-grained description of a metabolic network and, as such, it depends on the rate constants of all underlying biochemical reactions. Many previous studies have shown that within-protein epistasis is extremely common (e.g. *Lunzer et al., 2005*; *DePristo et al., 2005*; *Sailer and Harms, 2017b*; *Husain and Murugan, 2020*). Present work shows that, once epistasis arises at the level of protein activity, it will propagate all the way up the metabolic hierarchy and will manifest itself as epistasis for growth rate. It also suggests that growth rate is a generically non-linear function of the rate constants of the underlying biochemical reactions, such that all mutations that affect growth rate individually would also exhibit pairwise epistasis for growth rate with each other (*Kacser and Burns, 1981*; *DePristo et al., 2005*; *Martin et al., 2007*; *Chiu et al., 2012*; *Otwinowski et al., 2018*; *Husain and Murugan, 2020*).

In more complex organisms and/or in certain variable environments, it may be possible to decompose fitness into multiplicative or additive components, for example, plant's fitness may be equal to the product of the number of seeds it produces and their germination probability, as pointed out by *Chou et al., 2011*. Then, mutations that affect different components of fitness would exhibit no epistasis. However, such situations should be considered exceptional, as they require fitness to be decomposable and mutations to be non-pleiotropic.

If epistasis is in fact generic for high-level phenotypes, why do we not observe it more frequently? For example, a recent study that tested almost all pairs of gene knock-out mutations in yeast found genetic interactions for fitness for only about 4% of them (*Costanzo et al., 2016*). One possibility is that many pairs of mutations exhibit epistasis that is simply too small to detect with current methods. As the precision of fitness measurements improves, we would then expect the fraction of interacting gene pairs to grow. Another possibility is that systematic shifts in the distribution of estimated epistasis coefficients away from zero are taken by researchers as systematic errors rather than real phenomena, and are normalized out. Thus, some epistasis that would otherwise be detectable may be lost during data processing.

If epistasis is indeed as ubiquitous as the present analysis suggests, it would call into question how observations of inter-gene epistasis are interpreted. In particular, contrary to a common belief, a non-zero epistasis coefficient does not necessarily imply any specific functional relationship between the components affected by mutations beyond the fact that both components somehow contribute to the measured phenotype (*Boyle et al., 2017*). The focus of future research should then be not merely on documenting epistasis but on developing theory and methods for a robust inference of biological relationships from measured epistasis coefficients.

# Materials and methods

## Key ideas and logic of proofs of Theorems 1 and 2

Before proceeding to the detailed proofs of Theorem 1 and Theorem 2, I informally outline some key ideas and the basic logic.

The central object of the theory is a metabolic module. Modules have two key properties. First, a module is somewhat isolated from the rest of the metabolic network, in the sense that all metabolites inside it interact with only two metabolites outside, the I/O metabolites. The second property is that the metabolites within the module are sufficiently connected that each of them individually as well as any subset of them collectively can achieve a quasi-steady state (QSS), given the concentrations of the remaining metabolites. This property is proven in Proposition 1.

When some metabolites are at QSS, they can be effectively removed from the network and replaced with effective reactions among the remaining metabolites. In other words, one can 'coarse-grain' the network by removing metabolites. This approach is a standard biochemical-network reduction technique (*Segel, 1988*); for example, the Briggs-Haldane derivation of the Michaelis-Menten formula is based on this idea. In general, the resulting effective reactions have more complex (non-linear) kinetics than the original reactions. However, when the original reactions are first-order, the effective reactions are also first-order, that is, there is no increase in complexity. In Network coarse graining and an algorithm for evaluating the effective rate constant for an arbitrary module, I formally define the coarse-graining procedure (CGP) that eliminates one or multiple metabolites and replaces them with effective reactions.

CGP is an essential concept in my theory. I use it to compute the QSS concentrations for internal metabolites within a module (Corollary 1) and thereby prove Proposition 1, mentioned above. Since any module μ can achieve a QSS at any concentrations of its I/O metabolites and since any module has only two I/O metabolites, it can be replaced with a single effective reaction (Corollary 2). CGP provides a way to calculate the rate constant $y_\mu$ of this reaction. In other words, the CGP is an algorithm for evaluating function $F(\vec{x}_\mu)$ in *Equation 1* for any module μ.

CGP has an important property: its result does not depend on the order in which metabolites are eliminated. Therefore, in computing the effective rate constant of a module, we can choose any convinient way to eliminate its metabolites. Suppose that one module μ is nested within another module ν as in *Figure 1A*. A convenient way to compute the effective rate $y_\nu$ of the larger module is to first coarse-grain the smaller module μ, replacing it with an effective rate $y_\mu$, and then eliminate all the remaining metabolites in ν. Since effective rates after coarse-graining do not depend on the order of metabolite elimination, $y_\nu$ must depend on the rate constants $\vec{x}_\mu$ only indirectly, through $y_\mu$. In other words, all the information about the smaller module μ that is relevant for the performance of the larger module ν is contained in $y_\mu$. Therefore, if a mutation or mutations perturb only reactions inside of the smaller module μ, we only need to know their effects on the effective rate constant $y_\mu$ to completely understand how they will perturb the performance of the larger module ν. Specifically, if we have two such mutations A and B, all the information about them is contained in three numbers, $\delta^A y_\mu$, $\delta^B y_\mu$ and $\varepsilon y_\mu$. Theorem 1 then describes how epistasis at the level of module μ propagates to epistasis at the level of module ν.

The proof of Theorem 1 proceeds as follows. Let $a$ be the effective reaction with rate constant $y_\mu$ that represents module μ within the larger module ν, and consider $y_\nu$ as a function of $y_\mu$, as in *Equation 4*. To obtain $f_1(y_\mu)$, it is convenient to first eliminate all metabolites that do not participate in reaction $a$. No matter what the initial structure of module ν is, such coarse-graining will produce only one of three topologically distinct 'minimal' modules, which differ by the location of reaction $a$ with respect to the I/O metabolites of module ν (Figure 7). This implies that the function $f_1$ can belong to three parameteric families, where the parameters are the effective rate constants of reactions other than $a$ in each of the minimal modules. This is the essence of Proposition 2. Then Theorem 1 can be easily proven by explicitly evaluating the first- and second-order control coefficients for each of the three functions and showing that the statements of the theorem hold for all of them, irrespectively of the function's parameters.

Now consider two reactions $a$ and $b$ with rate constants ξ and η, and imagine the two mutations A and B that affect these reactions. To understand what value of $\varepsilon y_\mu$ will occur, we need to obtain $y_\mu$ as a function of ξ and η (*Equation 3*). To do so, it is convenient to first eliminate all metabolites that do not participate in reactions $a$ or $b$. No matter what the initial structure of module μ is, such coarse-graining will produce only one of nine topologically distinct minimal modules, which differ by the location of reactions $a$ and $b$ with respect to the I/O metabolites of module μ and each other (Figure 8). This implies that the function $f_2$ can belong to nine parameteric families. This is the essence of Proposition 3 and Corollary 3.

How does the topological relationship between reactions $a$ and $b$ translate into epistasis? First, there are only three types of relationships between any pair of reactions in a module: strictly serial, strictly parallel, or serial-parallel (see *Figure 3*). Second, Proposition 4 and Corollary 4 show that coarse-graining does not alter the strict relationships, that is, if reactions $a$ and $b$ are strictly serial or strictly parallel before coarse-graining they will remain so after coarse-graining. This is important because it implies that to prove Theorem 2 we do not need to consider an infinitely large space of

all modules but only a much smaller space of all minimal modules, that is, those that have only those metabolites that participate in the affected reactions $a$ and $b$. Although the space of all minimal modules is still infinite, the space of their topologies is finite (see Figure 8). For some minimal topologies, the connection between the strictly serial or strictly parallel relationship and epistasis can be established very easily. For example, if reaction $a$ and reaction $b$ both share an I/O metabolite as a substrate (see topological class $\mathcal{M}^{\text{io,io,IO}}$ in Figure 8), then they are always strictly parallel, no matter what the rest of the module looks like. Evaluating the first- and second-order control coefficients for the function $f_2$ that corresponds to this topological class reveals that $\varepsilon y_\mu \leq 0$ for any parameter values of $f_2$.

Unfortunately, most topological classes are too broad and include modules where reactions $a$ and $b$ are strictly serial as well as modules where they are strictly parallel or serial-parallel (e.g., class $\mathcal{M}^{\text{io,io},\emptyset}$). Consequently, the sign of $\varepsilon y_\mu$ for such modules can change depending on the values of the rate constants. However, since the number of distinct minimal topologies is finite, it is possible to identify all minimal topologies where the reactions $a$ and $b$ are strictly serial or strictly parallel. These topological sub-classes define parametric sub-families of function $f_2$, and we can explicitly calculate $\varepsilon y_\mu$ for all such functions. However, such brute-force approach is extremely cumbersome because the number of distinct minimal topologies is very large.

Fortunately, the following simple and intuitive fact greatly simplifies this problem. If two reactions are strictly serial or strictly parallel, this relationship does not change if a third reaction is removed from the module. This statement is the essence of Proposition 5. However, if the two reactions are serial-parallel, removal of a third reaction can change the relationship to a strictly serial or a strictly parallel one. As a consequence, there exist certain most connected 'generating' topologies where the relationship between the focal reactions is strictly parallel or strictly serial, and any other strictly serial minimal topology can be produced from at least one of the generating topologies by removal of reactions. This is the essence of Proposition 6. All generating topologies can be discovered by a simple algorithm provided in Appendix 3. They are listed in Table 2 and Table 3 and shown in Figure 10 through Figure 14. Each generating topology defines a parametric sub-family of function $f_2$, and I explicitly evaluate the first- and second-order control coefficients for all these sub-families (see Proposition 7 and Proposition 8) which essentially completes the proof of Theorem 2.

## Network coarse-graining

### Notations and definitions

Here, I give a more precise definition of the model and introduce additional notations and definitions. As mentioned above, all reactions are first order and reversible. Thus, each reaction $i \leftrightarrow j$ has one substrate $i \in A$ and one product $j \in A$, and it is fully described by its rate constant $x_{ij}$. By definition, $x_{ii} = 0$. I denote the set of all reactions by $R = \{i \leftrightarrow j : i, j \in A, \ x_{ij} > 0\}$. The dynamics of metabolite concentrations $S_1, \ldots, S_n$ in the network $\mathcal{N}$ are governed by equations

$$\dot{S}_i = \sum_{j=1}^{n} x_{ji} S_j - D_i S_i, \quad i \in A \tag{10}$$

where

$$D_i = \sum_{j=1}^{n} x_{ij}, \quad i \in A. \tag{11}$$

In what follows, it will be important to distinguish three types of reactions within a module, based on their topological relationship to the I/O metabolites of that module. The topology of the module $\mu$ is determined by its set of reactions $R_\mu = \left\{i \leftrightarrow j \in R : i, j \in A_\mu \cup A_\mu^{\text{IO}}\right\}$. I call all reactions where both the substrate and the product are internal to module $\mu$ as reactions internal to $\mu$, and I denote the set of all such reactions by $R_\mu^{\text{i}} \subset R_\mu$. For example, module $\mu$ in *Figure 1A* has one internal reaction $3 \leftrightarrow 4$. I call all reactions where one of the metabolites is internal to $\mu$ and the other is an I/O metabolite as the i/o reactions for $\mu$, and I denote the set of all such reactions by $R_\mu^{\text{io}} \subset R_\mu$. (I reserve upper-case 'I/O' for metabolites and use lower-case 'i/o' for reactions.) For example, module $\mu$ in *Figure 1A* has three i/o reactions $1 \leftrightarrow 3$, $1 \leftrightarrow 4$ and $2 \leftrightarrow 4$. Finally, I refer to reactions between any

two I/O metabolites for module μ as bypass reactions for module μ, and I denote the set of all such reactions by $R_\mu^b \subset R_\mu$. For example, module μ in *Figure 1A* has no bypass reactions but reaction $1 \leftrightarrow 5$ is a bypass reaction for module ν. By definition, all these three sets of reactions $R_\mu^i$, $R_\mu^{io}$ and $R_\mu^b$ are non-overlapping, and $R_\mu = R_\mu^i \cup R_\mu^{io} \cup R_\mu^b$.

Another important concept are the simple paths that lie within a module. For any two metabolites $i, j \in A \cup A_\mu^{IO}$, I denote a simple path between them that lies within μ as $p_{ij}^\mu$ or, equivalently as $i \leftrightarrow k_1 \leftrightarrow \ldots \leftrightarrow k_m \leftrightarrow j$ (where all $k_\ell \in A_\mu$). I say that each of the metabolites $k_\ell$ belongs to (or is contained in) path $p_{ij}^\mu$ (denoted by $k_\ell \in p_{ij}^\mu$). Similarly, I say that each of the reactions $k_\ell \leftrightarrow k_{\ell+1}$ belong to (or are contained in) path $p_{ij}^\mu$ (denoted by $k_\ell \leftrightarrow k_{\ell+1} \in p_{ij}^\mu$). I will drop superindex μ from $p_{ij}^\mu$ if it is clear what module is being referred to.

## Network coarse graining and an algorithm for evaluating the effective rate constant for an arbitrary module

In this section, I formally introduce and characterize the coarse-graining procedure (CGP). First, I introduce the main idea, which is to eliminate a metabolite that is at QSS and to replace it with a set of new reactions between metabolites adjacent to the eliminated one. This is exactly analogous to the star-mesh transformation in the theory of electric circuits (*Versfeld, 1970*). The resulting network is a coarse-grained version of the original network in the sense that it has one less metabolite. Next, I define the CGP, which is simply multiple metabolite eliminations applied successively. I prove Proposition 1, which justifies applying the CGP to a whole module and replacing it with a single effective reaction (Corollary 2). Finally, I show how to apply the CGP in practice to compute function $F$ from *Equation 1* for modules with some simple topologies.

### Elimination of a single metabolite

I begin by outlining the main idea behind the CGP, which is to replace one metabolite internal to a module with a series of effective reactions between metabolites adjacent to it. If the effective rate constants are defined appropriately, the dynamics of all metabolites in the resulting coarse-grained network are the same as in the original network, provided that the eliminated metabolite is at QSS in the original network.

To formalize this idea, suppose that module $\mu = (A_\mu, \vec{x}_\mu)$ contains $m$ internal metabolites. Let $k \in A_\mu$ be the internal metabolites that will be eliminated. Let $A^{\{k\}} = A \setminus \{k\}$ be the reduced metabolite set and let $\vec{x}^{\{k\}}$ be the reduced $(n-1) \times (n-1)$ matrix of rate constants defined by

$$x_{ij}^{\{k\}} = x_{ij} + \frac{x_{ik} x_{kj}}{D_k}, \quad i, j \in A^{\{k\}}, \quad i \neq j, \tag{12}$$

$$x_{ii}^{\{k\}} = 0, \quad i \in A^{\{k\}}, \tag{13}$$

where $D_k$ is given by *Equation 11*.

Such metabolite elimination has three properties that follow immediately from *Equation 12* and *Equation 13*. First, the rate constants of reactions do not change as long as the eliminated metabolite does not participate in them. Mathematically, $x_{ij}^{\{k\}} = x_{ij}$ for all metabolites $i$ and $j$ that are not adjacent to the eliminated metabolite $k$. In particular, this is true for all metabolites external to module μ. Second, because equilibrium constants have the property $K_{ij} = K_{i\ell} K_{\ell j}$ for any metabolites $i, j, \ell$, the rate constants $x_{ij}^{\{k\}}$ obey Haldane's relationships. Therefore, the reduced metabolite set $A^{\{k\}}$ and the reduced rate matrix $\vec{x}^{\{k\}}$ define a new 'coarse-grained' metabolic network $\mathcal{N}^{\{k\}} = (A^{\{k\}}, \vec{x}^{\{k\}})$. It is easy to show that subnetwork μ after the elimination of metabolite $k$ is still a module. Third, the reaction set of module μ (i.e., its topology) in the coarse-grained network $\mathcal{N}^{\{k\}}$ depends only on the reaction set of this module in the original network $\mathcal{N}$, but not on the particular values in the rate matrix $\vec{x}_\mu$.

Next, I will show that the dynamics of metabolites in the coarse-grained network $\mathcal{N}^{\{k\}}$ are identical to the dynamics of metabolites in the original network $\mathcal{N}$ where metabolite $k$ is at QSS. Note that if metabolite $k$ is at QSS in the network $\mathcal{N}$, its concentration is given by

$$S_k = \sum_{j \in A_\mu^{IO} \cup A_\mu \setminus \{k\}} \frac{x_{jk} S_j}{D_k}, \tag{14}$$

which follows from *Equation 10*. Now, the dynamics of metabolites in the network $\mathcal{N}^{\{k\}}$ are governed by equations

$$\dot{S}_i = \sum_{j \in A^{\{k\}}} x_{ji}^{\{k\}} S_j - D_i^{\{k\}} S_i, \quad for\ i \in A^{\{k\}}, \tag{15}$$

where $D_i^{\{k\}} = \sum_{j \in A^{\{k\}}} x_{ij}^{\{k\}}$. As mentioned above, $x_{ij}^{\{k\}} = x_{ij}$ for all pairs of metabolites where at least one metabolite is external to module μ. Therefore, *Equation 15* for the external metabolites are identical to *Equation 10* that govern the dynamics of these metabolites in the original network $\mathcal{N}$. Next, consider the dynamics of the I/O and internal metabolites for module μ in the coarse-grained network $\mathcal{N}^{\{k\}}$, that is, those in the set $A_\mu^{IO} \cup A_\mu \setminus \{k\}$. For any such metabolite $i$, the sum in the right-hand side of *Equation 15* can be re-written as

$$\sum_{j \in A_\mu^{IO} \cup A_\mu \setminus \{k\}} x_{ji}^{\{k\}} S_j = \sum_{j \in A_\mu^{IO} \cup A_\mu \setminus \{k\}} \left( x_{ji} + \frac{x_{jk} x_{ki}}{D_k} \right) S_j - \frac{x_{ik} x_{ki}}{D_k} S_i$$

$$= \sum_{j \in A_\mu^{IO} \cup A_\mu \setminus \{k\}} x_{ji} S_j + x_{ki} \sum_{j \in A_\mu^{IO} \cup A_\mu \setminus \{k\}} \frac{x_{jk} S_j}{D_k} - \frac{x_{ik} x_{ki}}{D_k} S_i. \tag{16}$$

According to *Equation 14*, the second term in *Equation 16* equals $x_{ki} S_k$, so that *Equation 16* becomes

$$\sum_{j \in A_\mu^{IO} \cup A_\mu \setminus \{k\}} x_{ji}^{\{k\}} S_j = \sum_{j \in A_\mu^{IO} \cup A_\mu} x_{ji} S_j - \frac{x_{ik} x_{ki}}{D_k} S_i. \tag{17}$$

For any metabolite $i \in A_\mu^{IO} \cup A_\mu \setminus \{k\}$, the second term in the righthand side of *Equation 15* can be re-written as

$$D_i^{\{k\}} = \sum_{j \in A_\mu^{IO} \cup A_\mu \setminus \{k\}} \left( x_{ij} + \frac{x_{ik} x_{kj}}{D_k} \right) - \frac{x_{ik} x_{ki}}{D_k} = D_i - \frac{x_{ik} x_{ki}}{D_k}. \tag{18}$$

Substituting *Equation 17* and *Equation 18* into *Equation 15*, we see that *Equation 15* is in fact equivalent to *Equation 10* for all $i \in A \setminus \{k\}$ with $S_k$ given by *Equation 14*.

## The coarse-graining procedure (CGP)

Here, I define the CGP for an arbitrary set of internal metabolites by applying metabolite elimination recursively.

Let $E \subseteq A_\mu$ be an arbitrary subset of metabolites internal to module μ in the metabolic network $\mathcal{N}$ and let $n_E$ be the number of elements in $E$. Let $A^E = A \setminus E$ be the reduced metabolite set after the metabolites have been eliminated. I define the reduced $(n - n_E) \times (n - n_E)$ matrix of rate constants $\vec{x}^E$ as follows. If $n_E = 1$, the matrix $\vec{x}^E$ is defined by *Equation 12* and *Equation 13*. If $n_E > 1$, then I define it recursively. Suppose that all metabolites in $E$ other than some metabolite $k \in E$ have been previously eliminated, such that the coarse-grained network $\mathcal{N}^{E'} = (A^{E'}, \vec{x}^{E'})$ is defined, with the set of eliminated metabolites $E' = E \setminus \{k\}$, $A^{E'} = A \setminus E'$, and the known matrix $\vec{x}^{E'}$. Then, I define the matrix $\vec{x}^E$ through the elimination of metabolite $k$ from $\mathcal{N}^{E'}$, that is,

$$x_{ij}^E = x_{ij}^{E'} + \frac{x_{ik}^{E'} x_{kj}^{E'}}{D_k^{E'}}, \quad i, j \in A^E, i \neq j, \tag{19}$$

$$x_{ii}^E = 0, \quad i \in A^E, \tag{20}$$

with

## Box 1. Properties of the coarse-graining procedure.

The coarse-graining procedure $\mathrm{CG}^E$ eliminating metabolites in set $E \subseteq A_\mu$ has the following useful properties which follow from *Equation 24*.

1. The effective rate constants $x_{ij}^E$ do not depend on the order in which metabolites are eliminated. Therefore, the composition of coarse-graining procedures is commutative, that is, if $E = E_1 \cup E_2$, where $E_1$ and $E_2$ are two non-overlapping subsets of $A_\mu$, then

$$\mathrm{CG}^{E_1} \circ \mathrm{CG}^{E_2} = \mathrm{CG}^{E_2} \circ \mathrm{CG}^{E_1} = \mathrm{CG}^E.$$

2. If at least one of the metabolites $i$ or $j$ is not adjacent to any of the eliminated metabolites, then $x_{ij}^E = x_{ij}$. In particular, $x_{ij}^E = x_{ij}$ if either $i$ and/or $j$ are external to µ.

3. The topology of module µ after the application of $\mathrm{CG}^E$ depends only on its original topology but not on the values of its rate constants $\vec{x}_\mu$.

4. If both metabolites $i$ and $j$ are adjacent to at least one eliminated metabolite, then $x_{ij}^E = x_{ij} + \alpha$, where $\alpha \geq 0$ depends only on the rate constants of reactions that involve at least one eliminated metabolite. In particular, if both $k$ and $\ell$ are from $A \setminus E$, then $x_{ij}^E$ is independent of $x_{k\ell}$.

5. If $E = A_\mu$, then the effective rate constant $y_\mu$ of module µ depends on the rate matrix $\vec{x}_\mu$ but does not depend on any other reaction rates, that is, *Equation 1* holds. Furthermore, the functional form of $F(\vec{x}_\mu)$ depends only on the topology of module µ, that is, all modules with the same topology are mapped onto $y_\mu$ by the same function $F$.

6. Suppose that module µ is nested in a larger module $\nu = (A_\nu, \vec{x}_\nu)$ (see *Figure 1A*). It follows from Property #1 that $\mathrm{CG}^\nu = \mathrm{CG}^\mu \circ \mathrm{CG}^{A_\nu \setminus A_\mu}$, that is, $y_\nu$ can be obtained by carrying out the CGP in two stages, by first eliminating module µ and replacing it with the effective reaction with the rate constant $y_\mu$ and then eliminating the remaining metabolites in $A_\nu$. In the network $\mathcal{N}^\mu$ after the first stage of coarse-graining, all rate constants $\vec{x}_{\nu \setminus \mu}$ are identical to those in the original network, that is, they are independent of $\vec{x}_\mu$, by virtue of Property #2. Therefore $y_\nu$ depends on the rate constants $\vec{x}_\mu$ of reactions within module µ only through the effective rate constant $y_\mu$ of module µ. In other words, *Equation 4* holds.

7. If, in the original network $\mathcal{N}$, metabolites $i$ and $j$ are adjacent or connected by a simple path that contains only the eliminated metabolites, then metabolites $i$ and $j$ are adjacent in the coarse-grained network $\mathcal{N}^E$.

8. If, in the metabolic network $\mathcal{N}$, metabolites $i$ and $j$ are not adjacent (i.e. $x_{ij} = 0$) and no simple path exists within the set $E$ (i.e. such that all non-terminal metabolites in this path are from $E$) that connects metabolites $i$ and $j$, then metabolites $i$ and $j$ are also not adjacent in the coarse-grained network $\mathcal{N}^E$ (i.e., $x_{ij}^E = 0$).

9. It follows from properties #7 and #8 that for a simple path $p_{\ell_1 \ell_m} = \ell_1 \leftrightarrow \ell_2 \leftrightarrow \cdots \leftrightarrow \ell_m$ to exist in module µ after the application of $\mathrm{CG}^\mu$, it is neccessary and sufficient that for each $i = 1, \ldots, m-1$, either $\ell_i$ and $\ell_{i+1}$ are adjacent in the original module µ or there exists a simple path $\ell_i \leftrightarrow \cdots \leftrightarrow \ell_{i+1}$ within the original module µ all of whose non-terminal metabolites are from $E$.

$$D_k^{E'} = \sum_{j \in A^{E'}} x_{kj}^{E'}. \tag{21}$$

I define the *the coarse-graining procedure that eliminates the metabolite set E* as a map

$$\mathrm{CG}^E : \mathcal{N} \mapsto \mathcal{N}^E = \left( A^E, \vec{x}^E \right).$$

As with the elimination of a single metabolite, it is straightforward to show that the rate constants $x_{ij}^E$ obey Haldane's relationships, so that $\mathcal{N}^E$ is indeed a metabolic network. $\mathrm{CG}^E$ maps module µ within the metabolic network $\mathcal{N}$ onto a subnetwork $\mu'$ within the metabolic network $\mathcal{N}^E$. It is straightforward to show that $\mu'$ is a module. Whenever there is no ambiguity, I will label both the original and the coarse-grained versions of the module by µ. To simplify notations, if the CGP

eliminates the entire module μ (i.e., if $E = A_\mu$), I label it $\mathrm{CG}^\mu$. I label the coarse-grained network that restults from the application of $\mathrm{CG}^\mu$ by $\mathcal{N}^\mu$ and I label the effective rate of the reaction substituting module μ in network $\mathcal{N}^\mu$ as $y_\mu$.

Intuitively, the result of coarse-graining should not depend on the order in which the metabolites are eliminated. To see this, let us obtain explicit (i.e. not recursive) expressions for $x_{ij}^E$. First, by applying the recursion *Equation 19*, it is easy to show that elimination of two metabolites $E = \{k, \ell\}$ yields effective rate constants

$$
\begin{aligned}
x_{ij}^{\{k,\ell\}} &= x_{ij}^{\{k\}} + \frac{x_{i\ell}^{\{k\}} x_{\ell j}^{\{k\}}}{D_\ell^{\{k\}}} \\
&= x_{ij} + \frac{D_k\, x_{i\ell}\, x_{\ell j} + D_\ell\, x_{ik}\, x_{kj} + x_{ik}\, x_{k\ell}\, x_{\ell j} + x_{i\ell}\, x_{\ell k}\, x_{kj}}{D_k\, D_\ell - x_{k\ell}\, x_{\ell k}} \quad (22) \\
&= x_{ij} + \frac{x_{i\ell}\, x_{\ell j}}{D_\ell^{\{k\}}} + \frac{x_{ik}\, x_{kj}}{D_k^{\{\ell\}}} + \frac{x_{ik}\, x_{k\ell}\, x_{\ell j}}{D_k^{\{\ell\}} D_\ell} + \frac{x_{i\ell}\, x_{\ell k}\, x_{kj}}{D_\ell^{\{k\}} D_k}, \quad i,j \in A \setminus \{k,\ell\},\, i \neq j. \quad (23)
\end{aligned}
$$

As expected, *Equation 22* and *Equation 23* are symmetric with respect to the eliminated metabolites $k$ and $\ell$. Extrapolating from *Equation 23*, it is possible to show that for an arbitrary metabolite subset $E \subseteq A_\mu$ that contains $n_E$ metabolites,

$$
x_{ij}^E = x_{ij} + \sum_{L=1}^{n_E} \sum_{(k_1, \ldots, k_L)} \frac{x_{ik_1}}{1} \frac{x_{k_1 k_2}}{D_{k_1}^{E \setminus \{k_1\}}} \cdots \frac{x_{k_{L-1} j}}{D_{k_L}^{E \setminus \{k_1, k_2, \ldots, k_L\}}}, \quad i,j \in A \setminus E,\, i \neq j. \quad (24)
$$

Here, the second sum is taken over all $n_E! / (n_E - L)!$ ordered lists of metabolites $(k_1, \ldots, k_L)$ from $E$. Each list can be thought of as a simple path within $E$ that connects metabolites $i$ and $j$. The proof of *Equation 24* can be found in Appendix 1. As expected, *Equation 24* shows that the effective reation rate $x_{ij}^E$ does not depend on the order in which metabolites are eliminated. This and other properties of the CGP are listed in *Box 1*.

One of key building blocks of the proofs of Theorem 1 and Theorem 2 is the fact that modules can be classified into a finite number of topological classes (see below). To arrive at this classification, it will be convenient to define a composition of coarse-graining procedures, as follows. Suppose that $\mathrm{CG}^{E_1}$ and $\mathrm{CG}^{E_2}$ are two coarse-graining procedures of network $\mathcal{N}$ for two subsets of metabolites $E_1 \subset A_\mu$ and $E_2 \subset A_\mu$. If the sets $E_1$ and $E_2$ are non-overlapping, $\mathrm{CG}^{E_2}$ is also defined for the coarse-grained network $\mathcal{N}^{E_1}$ which is the result of applying $\mathrm{CG}^{E_1}$ to the original network $\mathcal{N}$. The result of applying $\mathrm{CG}^{E_2}$ to the $\mathcal{N}^{E_1}$ is called *the composition of coarse-graining procedures* $\mathrm{CG}^{E_1}$ and $\mathrm{CG}^{E_2}$ of the original network $\mathcal{N}$ and is denoted as $\mathrm{CG}^{E_1} \circ \mathrm{CG}^{E_2}$.

As defined above, coarse-graining is a formal procedure, and there is no a priori guarantee that (a) it can in fact be carried out for every set of metabolites and (for example, because a metabolite set does not have a steady-state solution); and (b) it will not distort the dynamics in the rest of the network. The following proposition alleviates both of these concerns and thereby justifies the use of the CGP for any subset of internal metabolites within a module (including the entire module μ). It is straightforward to prove it by induction, using the same logic as in the elimination of a single metabolite.

## Proposition 1

Let $E$ be any subset of metabolites internal to module μ. Then,

1. There exists a joint QSS solution $\overline{S}_i$ for all metabolites $i \in E$, given the concentrations of the remaining internal and I/O metabolites for module μ.
2. The dynamics of all remaining metabolites in $A \setminus E$ in the coarse-grained metabolic network $\mathcal{N}^E$ are the same as in $\mathcal{N}$ where all metabolites in $E$ are at QSS.

## Corollary 1

Without loss of generality, suppose that the I/O metabolites for module μ are labeled 1 and 2 and its internal metabolites are labeled $A_\mu = \{3, 4, \ldots, m\}$. There exists a unique QSS $\overline{S}_i$ for all $i \in A_\mu$. The QSS concentrations can be obtained by recursively applying equation.

$$\overline{S}_k = \frac{1}{D_k^{\{k+1,\dots,m\}}} \left( x_{1k}^{\{k+1,\dots,m\}} S_1 + x_{2k}^{\{k+1,\dots,m\}} S_2 + \sum_{j \in \{3,\dots,k-1\}} x_{jk}^{\{k+1,\dots,m\}} \overline{S}_j \right) \tag{25}$$

for $k = 3, 4, \dots, m$.

*Equation 25* follows from *Equation 10* for the coarse-grained network obtained by eliminating metabolites $k+1, \dots, m$.

## Corollary 2

Without loss of generality, suppose that the I/O metabolites for module µ are labeled 1 and 2. Module µ can be replaced with a single effective reaction between its I/O metabolites, whose rate constant $y_\mu$ can be calculated using *Equation 19* and *Equation 20* or *Equation 24*. The dynamics of all metabolites in the resulting coarse-grained metabolic network are identical to their dynamics in the original network $\mathcal{N}$ where all metabolites internal to module µ are at the QSS determined by *Equation 25*.

## Computation of effective rate constants for simple modules

Corollary 2 provides a method for replacing any module µ at QSS with an effective rate $y_\mu = F(\vec{x}_\mu)$, which can be calculated using *Equation 19* and *Equation 20* or *Equation 24*. Here, I show how to implement this calculation for three simple metabolic modules.

### Linear pathway

Consider a linear pathway with I/O metabolites 1 and $m$ and internal metabolites $2, 3, \dots, m-1$ (*Figure 6A*). This labeling of metabolites is more convenient for the linear pathway. To calculate $y_\mu$, I will apply recursion *Equation 19* and *Equation 20*. I start by eliminating metabolite 2. After this initial coarse-graining step, the resulting module is still a linear pathway, where two reactions $1 \leftrightarrow 2 \leftrightarrow 3$ were replaced with a single reaction $1 \leftrightarrow 3$ with the effective rate constant.

$$x_{13}^{\{2\}} = \frac{x_{12} x_{23}}{x_{21} + x_{23}} = \left( \frac{1}{K_{12} x_{23}} + \frac{1}{x_{12}} \right)^{-1}.$$

All other rate constants remain unchanged. Next, I eliminate metabolite 3. The resulting module is still a linear pathway, where now three reactions $1 \leftrightarrow 2 \leftrightarrow 3 \leftrightarrow 4$ were replaced with a single reaction $1 \leftrightarrow 4$ with the effective rate constant

$$x_{14}^{\{2,3\}} = \frac{x_{13}^{\{2\}} x_{34}}{x_{31}^{\{2\}} + x_{34}} = \left( \frac{1}{K_{13} x_{34}} + \frac{1}{K_{12} x_{23}} + \frac{1}{x_{12}} \right)^{-1}.$$

All other rate constants remain unchanged. Continuing this process until all internal metabolites are eliminated, I obtain

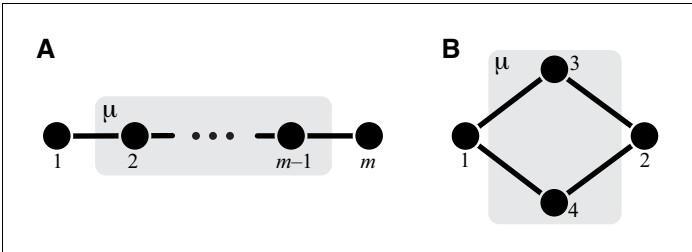

**Figure 6.** Simple modules. (A) Linear pathway. (B) Two parallel pathways.

$$y_\mu = \left( \sum_{i=1}^{m-1} \frac{1}{K_{1i} x_{ii+1}} \right)^{-1}, \tag{26}$$

which is identical to the expression originally obtained by *Kacser and Burns, 1973*.

## Two parallel pathways

Consider two parallel pathways with I/O metabolites 1 and 2 and internal metabolites 3 and 4 (*Figure 6B*). I obtain the effective rate constant using *Equation 22* with $i = 1$, $j = 2$, $k = 3$, $\ell = 4$. Since $x_{12} = x_{34} = 0$, *Equation 22* simplifies to

$$y_\mu = \frac{D_3 x_{14} x_{42} + D_4 x_{13} x_{32}}{D_3 D_4} = \frac{x_{14} x_{42}}{x_{41} + x_{42}} + \frac{x_{13} x_{32}}{x_{31} + x_{32}}. \tag{27}$$

Thus, the contributions of parallel pathways are simply added.

## Module μ in *Figure 1*

To obtain the effective rate constant for module μ shown in *Figure 1*, I again use *Equation 22* with $i = 1, j = 2, k = 3, \ell = 4$.

$$y_\mu = \frac{D_3 x_{14} x_{42} + D_4 x_{13} x_{32} + x_{13} x_{34} x_{42} + x_{14} x_{43} x_{32}}{D_3 D_4 - x_{34} x_{43}}, \tag{28}$$

with $D_3 = x_{31} + x_{32} + x_{34}$ and $D_4 = x_{41} + x_{42} + x_{43}$.

## Classification of modules with respect to 'marked' reactions, and the parametric families of functions $f_1$ and $f_2$

The CGP described above allows us to calculate the function $F$ that maps the rate matrix $\vec{x}_\mu$ for an arbitrary module μ onto the module's effective rate constant $y_\mu$. $F$ is a multivariate function of the entire matrix $\vec{x}_\mu$. However, in many applications, only one or two reactions are varied at a time while all others remain constant, and we want to know how module's effective parameter $y_\mu$ depends on these one or two perturbed reactions. I refer to such singled-out reactions as 'marked'. When $y_\mu$ is considered as a function of the rate constant $\xi$ of one marked reaction, I write $y_\mu = f_1(\xi)$, as in *Equation 2*. When $y_\mu$ is considered as a function of the rate constants $\xi$ and $\eta$ of two marked reactions, I write $y_\mu = f_2(\xi, \eta)$ as in *Equation 3*.

The functional form of $F$ and, as a consequence, the functional forms of $f_1$ and $f_2$ depend only on the topology of module μ (see Property #5 of the CGP in *Box 1*). In other words, modules with identical topologies have the same functional forms of $f_1$ and $f_2$, such that each topology of module μ defines a parametric family of functions $f_1$ and $f_2$, where all rate constants within module μ other than $\xi$, or $\xi$ and $\eta$, play a role of parameters.

Since the number of possible topologies is infinite, there is an infinite number of functional forms of $F$. However, the number of parametric families of functions $f_1$ and $f_2$ is finite, and it turns out to be small. To see this, notice that for any module with a single marked reaction, the CGP can be carried out in two stages. In the first stage, we can eliminate all metabolites that do not participate in the marked reaction. The resulting coarse-grained module is minimal in the sense that it can have at most two internal metabolites. Such minimal modules (and, as a consequence, all modules with one marked reaction) fall into three distinct topological classes, which are specified by the location of the marked reaction with respect to the I/O metabolites, as shown in *Figure 7*. This implies that there are only three parametric families of the function $f_1$. The topologies of the three minimal modules are sufficiently simple that the three corresponding parametric functional forms of $f_1$ can be easily computed by applying the coarse-graining *Equation 19* or *Equation 22*. This result is formulated in Proposition 2.

The same logic applies to modules with two marked reactions. CGP that eliminates all metabolites that do not participate in the marked reactions maps all such modules onto respective minimal modules, which can have at most four internal metabolites (see *Figure 8*). This result is formulated in Proposition 3. Minimal modules (and, as a consequence, all modules with two marked reactions) fall into nine distinct topological classes, which are specified by the locations of the marked reactions.

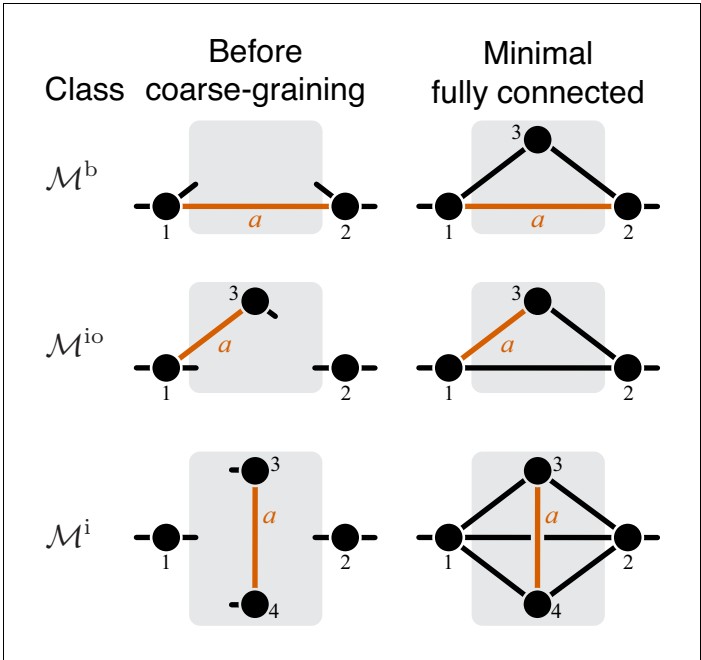

**Figure 7.** Classification of single-marked modules. Left column shows a general module from each topological class. The right column shows a minimal fully connected module in each topological class (see text for details). Circles represent metabolites and lines represent reactions. Only the I/O metabolites and the metabolites that participate in the marked reaction are shown, all other metabolites are suppressed. Short lines that have only one terminal metabolite represent all remaining reactions in which this metabolite participates, reactions between all other metabolites are suppressed. Metabolites are labeled according to the conventions listed in the text. The marked reaction is colored orange and labeled $a$. The module is represented by a gray rectangle, and the rest of the network is not shown.

All modules from the same topological class are described by functions $f_2$ from the same parametric family. These families are characterized in Corollary 3.

These topological classifications are extremely useful for the following reason. If we can show that all functions from the same parameteric family (corresponding to a given topological class) have some common property irrespectively of the values of the parameters, it would imply that this property holds for all modules from the corresponding topological class. This logic is a key part of the proofs of both Theorem 1 and Theorem 2.

To formalize this reasoning, consider module $\mu = (A_\mu, \vec{x}_\mu)$ and let $a = i_a \leftrightarrow j_a$ and $b = i_b \leftrightarrow j_b$ be two reactions from its set of reactions $R_\mu$. I call a pair $(\mu, a)$ a *single-marked module* and I call a triplet $(\mu, a, b)$ a *double-marked module*, and I refer to reactions $a$ and $b$ as *marked* within module μ. The topology of a single-marked module $(\mu, a)$ is determined not only by the reaction matrix $R_\mu$, but also by the position of the marked reaction, so I refer to the pair $(R_\mu, a)$ as the *topology of the single-marked module* $(\mu, a)$. Similarly, I refer to the triplet $(R_\mu, a, b)$ as the *topology of the double-marked module* $(\mu, a, b)$. I denote by $\vec{x}_{\mu\backslash a}$ the matrix of rate constants of all reactions in module μ other than reaction $a$ and I denote by $\vec{x}_{\mu\backslash\{a,b\}}$ the matrix of all rate constants in module μ other than reactions $a$ and $b$. I denote the sets of all single- and double-marked modules by $\mathcal{M}_1$ and $\mathcal{M}_2$, respectively. To avoid metabolite labeling ambiguities, I adopt the following conventions:

i.   The I/O metabolites are labeled $1$ and $2$ and the internal metabolites are labeled $3, 4, \ldots$;
ii.  $i_a, j_a \in \{1, 2, 3, 4\}$; $i_b, j_b \in \{1, 2, 3, 4, 5, 6\}$;
iii. $i_a < j_a$, $i_b < j_b$, $i_a \leq i_b$, $j_a \leq j_b$.

It is easy to see that the set of all single-marked modules $\mathcal{M}_1$ can be partitioned into three non-overlapping topological classes depending on the type of the marked reaction $a$. I denote the classes of all single-marked modules where the marked reaction is bypass, i/o or internal (see Notations and definitions) by $\mathcal{M}^b$, $\mathcal{M}^{io}$ and $\mathcal{M}^i$, respectively (*Figure 7*). Similarly, the set $\mathcal{M}_2$ can be partitioned

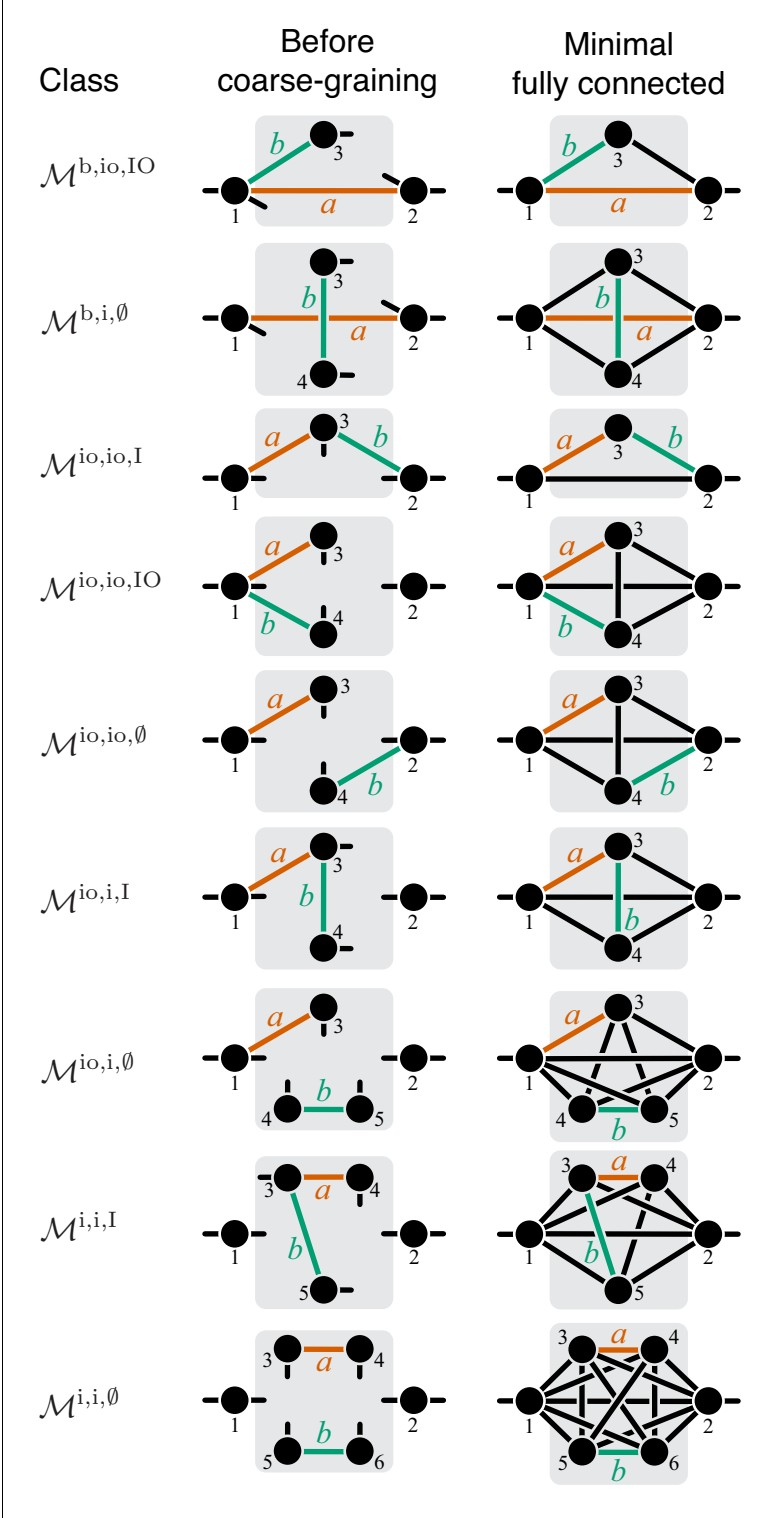

**Figure 8.** Classification of double-marked modules. Notations as in *Figure 7*.

into nine non-overlapping topological classes according to the types of marked reactions and the type of metabolite that is shared by both of these reactions (I/O, internal, or none). These classes are listed in *Table 1* and illustrated in *Figure 8*.

**Table 1.** Classification of double-marked modules.

Metabolites are labeled according to conventions described in the text. $m_{\mathcal{M}}$ is the minimum number of internal metabolites in a module from class $\mathcal{M}$. $A_{\mathcal{M}}$ is the set of internal and I/O metabolites in all minimal modules in class $\mathcal{M}$.

| Class | $a$ | $b$ | Shared metab. | Verbal description | $m_{\mathcal{M}}$ | $A_{\mathcal{M}}$ | Equation for $f_2$ |
|---|---|---|---|---|---|---|---|
| $\mathcal{M}^{b,io,IO}$ | $(1,2)$ | $(1,3)$ | 1 | Bypass and i/o reactions, shared I/O metabolite | 2 | $\{1,2,3\}$ | *Equation (34)* |
| $\mathcal{M}^{b,i,\emptyset}$ | $(1,2)$ | $(3,4)$ | – | Bypass and internal reactions, no shared metabolies | 2 | $\{1,2,3,4\}$ | *Equation (35)* |
| $\mathcal{M}^{io,io,I}$ | $(1,3)$ | $(2,3)$ | 3 | i/o reactions, shared internal metabolite | 1 | $\{1,2,3\}$ | *Equation (36)* |
| $\mathcal{M}^{io,io,IO}$ | $(1,3)$ | $(1,4)$ | 1 | i/o reactions, shared I/O metabolite | 2 | $\{1,2,3,4\}$ | *Equation (37)* |
| $\mathcal{M}^{io,io,\emptyset}$ | $(1,3)$ | $(2,4)$ | – | i/o reactions, no shared metabolites | 2 | $\{1,2,3,4\}$ | *Equation (38)* |
| $\mathcal{M}^{io,i,I}$ | $(1,3)$ | $(3,4)$ | 3 | i/o and internal reactions, shared internal metabolite | 2 | $\{1,2,3,4\}$ | *Equation (39)* |
| $\mathcal{M}^{io,i,\emptyset}$ | $(1,3)$ | $(4,5)$ | – | i/o and internal reactions, no shared metabolites | 3 | $\{1,2,3,4,5\}$ | *Equation (40)* |
| $\mathcal{M}^{i,i,I}$ | $(3,4)$ | $(3,5)$ | 3 | Internal reactions, shared internal metabolite | 3 | $\{1,2,3,4,5\}$ | *Equation (41)* |
| $\mathcal{M}^{i,i,\emptyset}$ | $(3,4)$ | $(5,6)$ | – | Internal reactions, no shared metabolites | 4 | $\{1,2,3,4,5,6\}$ | *Equation (42)* |

Each topological class contains infinitely many modules, with various numbers of metabolites and various topologies. However, for each topological class $\mathcal{M}$, there is a minimum number of internal metabolites $m_{\mathcal{M}}$, such that all modules within $\mathcal{M}$ must have at least $m_{\mathcal{M}}$ internal metabolites. I denote the set of metabolites minimal in the topological class $\mathcal{M}$ by $A_{\mathcal{M}}$. It is clear that for the single-marked module classes $\mathcal{M}^b$ and for $\mathcal{M}^{io}$, $m_{\mathcal{M}^b} = m_{\mathcal{M}^{io}} = 1$ and $A_{\mathcal{M}^b} = A_{\mathcal{M}^{io}} = \{3\}$, and for $\mathcal{M}^i$, $m_{\mathcal{M}^i} = 2$ and $A_{\mathcal{M}^i} = \{3,4\}$ (see second column in *Figure 7*). For the double-marked modules, $m_{\mathcal{M}}$ and $A_{\mathcal{M}}$ are given in *Table 1* and illustrated in *Figure 8*.

If a single-marked module from the topological class $\mathcal{M}$ has the minimum number of metabolites $n_{\mathcal{M}}$ in that class, I call such module and its topology *minimal* in $\mathcal{M}$. There may be several minimal topologies in a topological class, but there is only one minimal topology that is fully connected. A topology $(R_\mu, a)$ is called *fully connected* if the reaction set $R_\mu$ is *complete* in the sense that it contains reactions between all pairs of metabolites in the minimal metabolite set $A_{\mathcal{M}}$. I denote such complete reaction set for the class $\mathcal{M}$ by $R_{\mathcal{M}}$, and I denote the respective fully connected topology by $(R_{\mathcal{M}}, a)$. I employ the same terminology and analogous notations for double-marked modules. The minimal fully connected topologies are shown in the second column in *Figure 7* and *Figure 8*.

Next, I prove Proposition 2, which is the key step toward the proof of Theorem 1. This proposition states that there are only three functional forms for the function $f_1$ and characterizes them. The idea of the proof is the following. According to Property 5 (*Box 1*), all single-marked modules that are mapped by the CGP onto a minimal module with the same topology $(R_\mu, a)$ have the same functional form of $f_1$. In other words, each minimal topology $(R_\mu, a)$ specifies a parameteric family of the function $f_1$. Since the number of possible minimal topologies is finite, the claim of Theorem 1 can be tested for each corresponding functional form of $f_1$. However, the number of minimal topologies is rather large. Fortunately, another simplification is possible. Since the reaction set $R_\mu$ of any minimal single-marked module is a subset of the complete reaction set $R_{\mathcal{M}}$, the fully connected topology $(R_{\mathcal{M}}, a)$ specifies the largest parametric family of the function $f_1$ for the class $\mathcal{M}$, such that all other families can be obtained from it by setting some parameters to zero, which is equivalent to removing reactions from the fully connected topology. In other words, *all* single-marked modules that belong to the topological class $\mathcal{M}$ are described by functions $f_1$ that belong to one parameteric family corresponding to the fully connected topology minimal in $\mathcal{M}$. The three parameteric families of $f_1$ are characterized by Proposition 2.

One important consequence of Proposition 2 is that it is not necessary to test the claim of Theorem 1 for each family of $f_1$ that corresponds to each minimal topology. Instead, it is sufficient to test it for the three families that correspond to the fully connected minimal topologies in each class.

## Proposition 2

Let $(\mu, a)$ be a single-marked module, and let $\xi$ be the rate constant of reaction $a$. Then $y_\mu = f_1(u)$, where $u = \xi + \alpha$ for some $\alpha \geq 0$, and the function $f_1$ is given by one of the following expressions.

$$f_1(u) = u, \quad \text{if } (\mu, a) \in \mathcal{M}^{\mathrm{b}}, \tag{29}$$

$$f_1(u) = w_{12} + \frac{u\, w_{32}}{u/K_{13} + w_{32}}, \quad \text{if } (\mu, a) \in \mathcal{M}^{\mathrm{io}}, \tag{30}$$

$$f_1(u) = w_{12} + \frac{D_3\, w_{14}\, w_{42} + D_4\, w_{13}\, w_{32} + w_{13}\, w_{42}\, u + w_{14}\, w_{32}\, u/K_{34}}{D_3\, D_4 - u^2/K_{34}}, \quad \text{if } (\mu, a) \in \mathcal{M}^{\mathrm{i}}. \tag{31}$$

Here $D_3 = w_{31} + w_{32} + u$, $D_4 = w_{41} + w_{42} + u/K_{34}$, and all $w_{ij} \geq 0$ are independent of $\xi$.

### Proof

Since any single-marked module $(\mu, a)$ belongs to exactly one of three classes $\mathcal{M}^{\mathrm{b}}$, $\mathcal{M}^{\mathrm{io}}$ and $\mathcal{M}^{\mathrm{io}}$, I consider these three cases one by one.

Case $(\mu, a) \in \mathcal{M}^{\mathrm{b}}$. According to the labeling conventions outlined above, $a = 1 \leftrightarrow 2$ (see **Figure 7**). **Equation 29** follows directly from Property #4 of the CGP (**Box 1**).

Case $(\mu, a) \in \mathcal{M}^{\mathrm{io}}$. According to the labeling conventions, $a = 1 \leftrightarrow 3$ (see **Figure 7**). According to Property #1 of the CGP, module $\mu$ can be coarse-grained in two stages, by first applying $\mathrm{CG}^{A_\mu \backslash \{3\}}$ which eliminates metabolites $4, \ldots, m$ (those that do not participate in the marked reaction) and then applying $\mathrm{CG}^{\{3\}}$ which eliminates the remaining metabolite 3. Mathematically, $\mathrm{CG}^\mu = \mathrm{CG}^{A_\mu \backslash \{3\}} \circ \mathrm{CG}^{\{3\}}$. After applying $\mathrm{CG}^{A_\mu \backslash \{3\}}$, the resulting coarse-grained module $\mu'$ has a single internal metabolite 3 and at most three effective reactions $1 \leftrightarrow 2$, $1 \leftrightarrow 3$ and $2 \leftrightarrow 3$ (**Figure 7**), that is, it is minimal in $\mathcal{M}^{\mathrm{io}}$. By virtue of Properties #2 and #4 of the CGP, the effective rate constants $w_{12}$, $w_{23}$ are independent of $\xi$ and $u \equiv w_{13} = \xi + \alpha$. Note that $w_{12}$ may equal zero, but $w_{23} \neq 0$ because $\mu'$ is a module. Regardless, the reaction set $R_{\mu'}$ of module $\mu'$ is always a subset of the complete reaction set $R_{\mathcal{M}^{\mathrm{io}}}$. Thus, to obtain the effective rate constant $y_\mu$, I consider the most general case when $\mu'$ is fully connected and eliminate the remaining internal metabolite 3, which leads to **Equation 30**.

Case $(\mu, a) \in \mathcal{M}^{\mathrm{i}}$. According to the labeling conventions, $a = 3 \leftrightarrow 4$ (see **Figure 7**). Otherwise, the logic of the proof is exactly the same as for the case $(\mu, a) \in \mathcal{M}^{\mathrm{io}}$.

Next I prove Proposition 3 which states that, for any double-marked module that belongs to a given topological class, there exists a double-marked module that is minimal in the same class, such that both modules have the same function $f_2$. The corresponding minimal module is obtained from the original module by applying the CGP. This proposition is important because it implies that all functions $f_2$ can be completely characterized by only examing minimal modules. Then, analogously to single-marked modules, Corollary 3 states that function $f_2$ can belong to one of nine parameteric families which are defined by the fully connected minimal topologies in each topological class.

## Proposition 3

Let $(\mu, a, b)$ be a double-marked module that belongs to the topological class $\mathcal{M}$, and let $\xi$ and $\eta$ be the rate constants of reactions $a$ and $b$, respectively. Then there exist non-negative constants $\alpha$ and $\beta$ and a double-marked module $(\mu', a, b)$ minimal in $\mathcal{M}$ such that $y_\mu = y_{\mu'} = f_2(u, v)$, where

$$u = \xi + \alpha, \tag{32}$$

$$v = \eta + \beta \tag{33}$$

are the rate constants of the marked reactions $a$ and $b$ in $\mu'$, respectively, and all other rate constants in $\mu'$ are independent of $\xi$ or $\eta$. Module $\mu'$ is obtained from $\mu$ by the coarse-graining procedure $\mathrm{CG}^{\mu \backslash \{a,b\}}$ that eliminates all metabolites internal to module $\mu$ that do not participate in reactions $a$ or $b$.

## Proof

To prove this proposition, I will construct the double-marked module $(\mu', a, b)$ minimal in $\mathcal{M}$ by applying $\mathrm{CG}^{\mu\setminus\{a,b\}}$. Let $m_{\mathcal{M}}$ be the mimimal number of internal metabolites in class $\mathcal{M}$ (see **Table 1**). According to the metabolite labeling conventions, metabolites $n_{\mathcal{M}}+3, n_{\mathcal{M}}+4, \ldots$ are neither I/O nor do they participate in the marked reactions. $\mathrm{CG}^{\mu\setminus\{a,b\}}$ eliminates all these metabolites and maps module μ onto module $\mu'$, all of whose internal metabolites participate in reactions $a$ and/or $b$. Therefore, $(\mu', a, b)$ is minimal in class $\mathcal{M}$ (**Figure 8**). According to Properties #2 and #4 of the CGP (**Box 1**), the effective rate constants $u$ and $v$ of reactions $a$ and $b$ in module $\mu'$ are given by linear relationships in **Equation 32** and **Equation 33**, and the remaining effective rate constants are independent of ξ and η. The fact that $y_\mu = y_{\mu'}$ follows from Property #1 of the CGP, $\mathrm{CG}^\mu = \mathrm{CG}^{\mu\setminus\{a,b\}} \circ \mathrm{CG}^A_{\mu'}$.

## Corollary 3

Let $(\mu, a, b)$ be a double-marked module, and let ξ and η be the rate constants of reactions $a$ and $b$, respectively. The function $f_2$ that maps ξ and η onto module's effective rate constant $y_\mu$ belongs to one of nine parametric families. If $(\mu, a, b) \in \mathcal{M}^{\mathrm{b,io,IO}}$, then

$$f_2(u,v) = u + \frac{v\, w_{32}}{v/K_{13} + w_{32}}. \tag{34}$$

If $(\mu, a, b) \in \mathcal{M}^{\mathrm{b,i,\emptyset}}$, then

$$\begin{aligned}
f_2(u,v) &= u + \frac{D_3\, w_{14}\, w_{42} + D_4\, w_{13}\, w_{32} + w_{13}\, w_{42}\, v + w_{14}\, w_{32}\, v/K_{34}}{D_3 D_4 - v^2/K_{34}}, \\
D_3 &= w_{31} + w_{32} + v, \\
D_4 &= w_{41} + w_{42} + v/K_{34}.
\end{aligned} \tag{35}$$

If $(\mu, a, b) \in \mathcal{M}^{\mathrm{io,io,I}}$, then

$$f_2(u,v) = w_{12} + \frac{u\, v}{u/K_{13} + v}, \tag{36}$$

If $(\mu, a, b) \in \mathcal{M}^{\mathrm{io,io,IO}}$, then

$$\begin{aligned}
f_2(u,v) &= w_{12} + \frac{D_3\, v\, w_{42} + D_4\, u\, w_{32} + u\, w_{34}\, w_{42} + v\, w_{43}\, w_{32}}{D_3 D_4 - w_{34}\, w_{43}}, \\
D_3 &= u/K_{13} + w_{32} + w_{34}, \\
D_4 &= v/K_{14} + w_{42} + w_{43}.
\end{aligned} \tag{37}$$

If $(\mu, a, b) \in \mathcal{M}^{\mathrm{io,io,\emptyset}}$, then

$$\begin{aligned}
f_2(u,v) &= w_{12} + \frac{D_3\, w_{14}\, v/K_{24} + D_4\, u\, w_{32} + u\, w_{34}\, v/K_{24} + w_{14}\, w_{43}\, w_{32}}{D_3 D_4 - w_{34}\, w_{43}}, \\
D_3 &= u/K_{13} + w_{32} + w_{34}, \\
D_4 &= w_{41} + v/K_{24} + w_{43}.
\end{aligned} \tag{38}$$

If $(\mu, a, b) \in \mathcal{M}^{\mathrm{io,i,I}}$, then

$$\begin{aligned}
f_2(u,v) &= w_{12} + \frac{D_3\, w_{14}\, w_{42} + D_4\, u\, w_{32} + u\, v\, w_{42} + w_{14}\, w_{32}\, v/K_{34}}{D_3 D_4 - v^2/K_{34}}, \\
D_3 &= u/K_{13} + w_{32} + v, \\
D_4 &= w_{41} + w_{42} + v/K_{43}.
\end{aligned} \tag{39}$$

If $(\mu, a, b) \in \mathcal{M}^{\mathrm{io,i,\emptyset}}$, then

$$f_2(u,v) = W_{12} + \frac{W_{13}\,W_{32}}{W_{31} + W_{32}},$$

$$W_{ij} = w_{ij} + \frac{D_4\,w_{i5}\,w_{5j} + D_5\,w_{i4}\,w_{4j} + w_{i4}\,v\,w_{5j} + w_{i5}\,w_{4j}\,v/K_{45}}{D_4\,D_5 - v^2/K_{45}},$$

$$D_4 = w_{41} + w_{42} + w_{43} + v,$$

$$D_5 = w_{51} + w_{52} + w_{53} + v/K_{45},$$

$$w_{13} \equiv u.$$

(40)

If $(\mu,a,b) \in \mathcal{M}^{\mathrm{i,i,I}}$, then

$$f_2(u,v) = W_{12} + \frac{D_3\,W_{14}\,W_{42} + D_4\,W_{13}\,W_{32} + W_{13}\,W_{34}\,W_{42} + W_{14}\,W_{43}\,W_{32}}{D_3\,D_4 - W_{34}\,W_{43}},$$

$$W_{ij} = w_{ij} + \frac{w_{i5}\,w_{5j}}{D_5},$$

$$D_3 = W_{31} + W_{32} + W_{34},$$

$$D_4 = W_{41} + W_{42} + W_{43},$$

$$D_5 = w_{51} + w_{52} + w_{53} + w_{54},$$

$$w_{34} \equiv u,$$

$$w_{35} \equiv v.$$

(41)

If $(\mu,a,b) \in \mathcal{M}^{\mathrm{i,i,\emptyset}}$, then

$$f_2(u,v) = W_{12} + \frac{D_3\,W_{14}\,W_{42} + D_4\,W_{13}\,W_{32} + W_{13}\,W_{34}\,W_{42} + W_{14}\,W_{43}\,W_{32}}{D_3\,D_4 - W_{34}\,W_{43}},$$

(42)

$$W_{ij} = w_{ij} + \frac{D_5\,w_{i6}\,w_{6j} + D_6\,w_{i5}\,w_{5j} + w_{i5}\,w_{6j}\,v + w_{i6}\,w_{5j}\,v/K_{56}}{D_5\,D_6 - v^2/K_{56}},$$

(43)

$$D_3 = W_{31} + W_{32} + W_{34},$$

$$D_4 = W_{41} + W_{42} + W_{43},$$

$$D_5 = w_{51} + w_{52} + w_{53} + w_{54} + v,$$

$$D_6 = w_{61} + w_{62} + w_{63} + w_{64} + v/K_{56},$$

$$w_{34} \equiv u.$$

In *Equation 34* through *Equation 35*, $u$ and $v$ are given by *Equation 32* and *Equation 33*. All effective activities $w_{ij} \geq 0$ are constants (other than cases where they stand for $u$ or $v$) that depend on the topology of module $\mu$ and on the parameters $\vec{x}_{\mu \backslash \{a,b\}}$ but do not depend on $\xi$ and $\eta$.

## Proof

This statement and *Equation 34* through *Equation 35* follow directly from Proposition 3 and the fact that the reaction set of any double-marked module in any given topological class is a subset of the complete reaction set in that topological class.

### Derivation of *Equation 6* and *Equation 9*

Consider a higher-level phenotype $y$, such as the effective activity of a module, which is function of a multivariate lower-level phenotype $\vec{x} = (x_1, x_2, \ldots, x_n)$, such as the rates of individual reactions within the module, $y = F(\vec{x})$. Denote the wildtype values of the phenotypes as $\vec{x}^0 = (x_1^0, x_2^0, \ldots, x_n^0)$ and $y^0 = F(\vec{x}^0)$. Consider a mutation that perturbes these values, so that the mutant has lower-level phenotypic values $\vec{x}' = (x_1', x_2', \ldots, x_n')$. The relative effect of the mutation on phenotype $x_i$ is $\delta x_i = x_i'/x_i^0 - 1$. If all $\|\delta x_i\| \ll 1$ where $\|\vec{x}\|$ denotes the length of vector $\vec{x}$, then the value of the higher-level phenotype $y'$ in the mutant is given by

$$y' = y^0 \left(1 + \sum_{i=1}^{n} C_i\,\delta x_i + \frac{1}{2}\sum_{i,j=1}^{n} H_{ij}\,\delta x_i\,\delta x_j\right) + o\left(\|\delta \vec{x}\|^2\right).$$

(44)

where

$$C_i = \frac{x_i^0}{y^0} \frac{\partial F}{\partial x_i}\bigg|_{\vec{x}=\vec{x}^0}, \quad i = 1, \ldots, n, \tag{45}$$

$$H_{ij} = \frac{x_i^0 x_j^0}{y^0} \frac{\partial^2 F}{\partial x_i \partial x_j}\bigg|_{\vec{x}=\vec{x}^0}, \quad i,j = 1, \ldots, n, \tag{46}$$

which I refer to as first- and second-order control coefficients of the lower-level phenotypes $x_i$ and $x_j$ with respect to the higher-level phenotype $y$.

Now consider two single mutants, $A$ and $B$, and the double-mutant $AB$. Each mutation $A$ and $B$ and their combination may perturb all $x_i$ phenotypes such that $x_i^A = x_i^0(1 + \delta^A x_i)$, $x_i^B = x_i^0(1 + \delta^B x_i)$, and $x_i^{AB} = x_i^0(1 + \delta^{AB} x_i) = x_i^0(1 + \delta^A x_i + \delta^B x_i + 2\,\delta^A x_i \delta^B x_i \varepsilon x_i)$.

Assuming that $\|\delta^A \vec{x}\| \ll 1$, $\|\delta^B \vec{x}\| \ll 1$ and $\|\delta^{AB} \vec{x}\| \ll 1$, using the approximation in *Equation 44* and the definition of $\varepsilon x_i$ (analogous to *Equation 5*), I obtain

$$\delta^A y = \sum_{i=1}^n C_i \delta^A x_i + o(\|\delta^A \vec{x}\|), \tag{47}$$

$$\delta^B y = \sum_{i=1}^n C_i \delta^B x_i + o(\|\delta^B \vec{x}\|), \tag{48}$$

$$\varepsilon y = \frac{\sum_{i=1}^n C_i \varepsilon x_i \delta^A x_i \delta^B x_i + \frac{1}{2} \sum_{i,j=1}^n H_{ij} \delta^A x_i \delta^B x_j}{\sum_{i=1}^n \sum_{j=1}^n C_i C_j \delta^A x_i \delta^B x_j} + o(1), \tag{49}$$

where $o(1)$ refers to terms that are vanishingly small as $\delta^A x_i \to 0$, $\delta^B x_i \to 0$, $i = 1, \ldots n$.

I examine two special cases of *Equation 49*. The first special case is when both mutations affect a single phenotype $x_k$, that is, when all $\delta^A x_i = 0$ and all $\delta^B x_i = 0$ except for $i = k$. Then *Equation 47*, *Equation 48*, *Equation 49* simplify to

$$\delta^A y = C_k \delta^A x_k + o(|\delta^A x_k|), \tag{50}$$

$$\delta^B y = C_k \delta^B x_k + o(|\delta^B x_k|), \tag{51}$$

$$\varepsilon y = \frac{\varepsilon x_k}{C_k} + \frac{H_{kk}}{2 C_k^2} + o(1). \tag{52}$$

*Equation 52* is equivalent to *Equation 6*.

The second special case is when mutation $A$ affects a single phenotypes $x_k$ and mutation $B$ affects a single phenotype $x_\ell$ ($k \neq \ell$), i.e., all $\delta^A x_i = 0$ except for $i = k$, all $\delta^B x_i = 0$ except for $i = \ell$, and all $\varepsilon x_i = 0$. Then *Equation 47*, *Equation 48*, *Equation 49* simplify to

$$\delta^A y = C_k \delta^A x_k + o(|\delta^A x_k|), \tag{53}$$

$$\delta^B y = C_\ell \delta^B x_\ell + o(|\delta^B x_\ell|), \tag{54}$$

$$\varepsilon y = \frac{H_{k\ell}}{2 C_k C_\ell} + o(1). \tag{55}$$

*Equation 55* is equivalent to *Equation 9*.

## Calculation of epistasis in simple modules

*Equation 52* and *Equation 55* allow me to compute how epistasis propagates and emerges in arbitrary metabolic networks. In this section, I show how to implement these calculations for three simple metabolic modules considered above and in module ν shown in *Figure 3*.

## Linear pathway

First, consider how epistasis propagates through a linear pathway (*Figure 6A*). For simplicity, assume that both mutations $A$ and $B$ affect the same reaction $1 \leftrightarrow 2$. It follows from *Equation 26* that

$$y_\mu = f_1(\xi) = \left( \frac{1}{\xi} + \alpha \right)^{-1},$$

where $\alpha$ is a positive constant. Therefore, the first- and second-order control coefficients of reaction $1 \leftrightarrow 2$ with respect to the flux through the linear pathway $\mu$ are given by

$$C = \frac{y_\mu}{\xi} = \frac{1}{1 + \alpha \xi},$$

$$H = -\frac{2\alpha \xi}{(1 + \alpha \xi)^2}.$$

Substituting these expressions into the expression for the fixed point $\bar{\varepsilon} = -H(2C(1-C))^{-1}$, I find that $\bar{\varepsilon} = 1$, irrespectively of the rates of other reactions in the linear pathway. This implies that epistasis $\varepsilon\xi < 1$ at the level of reaction $1 \leftrightarrow 2$ would induce a lower value of epistasis $\varepsilon y_\mu < \varepsilon\xi < 1$ at the level of the entire linear pathway, any value $\varepsilon\xi > 1$ would induce a higher value of epistasis $\varepsilon y_\mu > \varepsilon\xi > 1$, and $\varepsilon\xi = 1$ would induce $\varepsilon y_\mu = 1$.

Now consider emergence of epistasis in a linear pathway. Suppose that mutation $A$ affects reaction $1 \leftrightarrow 2$ and mutation $B$ affects reaction $2 \leftrightarrow 3$. Denote the rate constant of reactions $1 \leftrightarrow 2$ and $2 \leftrightarrow 3$ by $\xi \equiv x_{12}$ and $\eta \equiv x_{23}$, respectively. It follows from *Equation 26* that

$$y_\mu = f_2(\xi, \eta) = \left( \frac{1}{\xi} + \frac{1}{K_{12}\eta} + \beta \right)^{-1},$$

where $\beta$ is a positive constant. Therefore,

$$\begin{aligned} C_\xi &= \frac{y_\mu}{\xi}, \\ C_\eta &= \frac{y_\mu}{K_{12}\,\eta}, \\ H_{\xi\eta} &= \frac{2y_\mu^2}{K_{12}\,\xi\,\eta}, \end{aligned}$$

which, after substituting into *Equation 9*, yield $\varepsilon y_\mu = 1$. Together with the fact that $\bar{\varepsilon} = 1$ (see above), this result implies that epistasis coefficient between any two mutations that affect different reactions in a linear pathway equals 1.

## Two parallel pathways

Suppose that mutation $A$ affects reaction $1 \leftrightarrow 3$ and mutation $B$ affects reaction $1 \leftrightarrow 4$ in the linear metabolic pathway shown in *Figure 6B*. Denote the rate constants of reaction $1 \leftrightarrow 3$ and $1 \leftrightarrow 4$ by $\xi \equiv x_{13}$ and $\eta \equiv x_{14}$. It follows from *Equation 27* that

$$y_\mu = f_2(\xi, \eta) = \left( \frac{1}{\xi} + \alpha \right)^{-1} + \left( \frac{1}{\eta} + \beta \right)^{-1},$$

where $\alpha = 1/(K_{13}x_{32})$ and $\beta = 1/(K_{14}x_{42})$. Thus, we have $H_{\xi\eta} = 0$, and there is no epistasis between such mutations.

## Module $\nu$ in *Figure 3A*

I denote the rate of the reactions affected by mutations $A$ and $B$ by $\xi = x_{13}$ and $\eta = x_{42}$, and I also denote $z = x_{34}$. I will calculate the epistasis coefficient $\varepsilon y_\nu$ in two stages, by first calculating the epistasis coefficient $\varepsilon y_\mu$ and then propagating it to $\varepsilon y_\nu$ using *Equation 6*. Here I am specifically interested in how $\varepsilon y_\nu$ depends on the rate constant $z$.

To compute epistasis between mutations $A$ and $B$ at the level of module $\mu$, I rewrite *Equation 28* as

$$y_\mu = \frac{a\,\xi\,\eta + b_\xi\,\xi + b_\eta\,\eta + c}{d\,\xi\,\eta + e_\xi\,\xi + e_\eta\,\eta + f},$$

where $a = x_{14}/K_{13} + x_{32} + z$, $b_\xi = x_{32}(x_{41} + z/K_{34})$, $b_\eta = x_{14}(x_{32} + z)$, $c = x_{14}\,x_{32}\,z/K_{34}$, $d = 1/K_{13}$, $e_\xi = (x_{41} + z/K_{34})/K_{13}$, $e_\eta = x_{32} + z$, $f = x_{32}\,z/K_{34} + x_{41}\,z + x_{32}\,x_{41}$. I obtain the following expressions for the first- and second-order control coefficients.

$$C_\xi = \frac{\xi}{y_\mu}\left(\frac{\tilde{c}_1\,z + \tilde{d}_1}{D}\right)^2, \tag{56}$$

$$C_\eta = \frac{\eta}{y_\mu}\frac{1}{K_{14}}\left(\frac{\tilde{c}_2\,z + \tilde{d}_2}{D}\right)^2, \tag{57}$$

$$H_{\xi\eta} = \frac{\xi\,\eta}{y_\mu}\frac{2z}{K_{14}}\frac{(\tilde{c}_1\,z + \tilde{d}_1)(\tilde{c}_2\,z + \tilde{d}_2)}{D^3}, \tag{58}$$

where $D = d\,\xi\,\eta + e_\xi\,\xi + e_\eta\,\eta + f$, $\tilde{c}_1 = x_{23}/K_{24} + \eta$, $\tilde{d}_1 = x_{32}\,(x_{41} + \eta)$, $\tilde{c}_2 = \xi + x_{14}$, $\tilde{d}_2 = x_{14}\,(\xi/K_{13} + x_{32})$. Substituting *Equation 56* through *Equation 58* into *Equation 53* through *Equation 55*, I obtain

$$\delta^A y_\mu = \frac{\xi}{y_\mu}\left(\frac{\tilde{c}_1\,z + \tilde{d}_1}{D}\right)^2 \delta^A \xi, \tag{59}$$

$$\delta^B y_\mu = \frac{\eta}{y_\mu}\frac{1}{K_{14}}\left(\frac{\tilde{c}_2\,z + \tilde{d}_2}{D}\right)^2 \delta^B \eta. \tag{60}$$

$$\varepsilon y_\mu = \frac{z\,(\tilde{a}\,z + \tilde{b})}{(\tilde{c}_1\,z + \tilde{d}_1)(\tilde{c}_2\,z + \tilde{d}_2)}, \tag{61}$$

where $\tilde{a} = \tilde{c}_1\,\tilde{c}_2$ and $\tilde{b} = (\xi/K_{13} + x_{32})\,x_{14}\,\eta + (x_{41} + \eta)\,x_{32}\,\xi$.

To obtain the expression for $\varepsilon y_\nu$, I coarse-grain module $\nu$ by eliminating the only remaining internal metabolite 2 and obtain

$$y_\nu = x_{15} + \frac{y_\mu\,x_{25}}{y_\mu/K_{12} + x_{25}}.$$

I then apply equation *Equation 6* with

$$C = \frac{y_\mu}{y_\nu}\frac{x_{25}^2}{(y_\mu/K_{12} + x_{25})^2}, \tag{62}$$

$$H = -\frac{2y_\mu^2}{y_\nu\,K_{12}}\frac{x_{25}^2}{(u/K_{12} + x_{25})^3}. \tag{63}$$

*Figure 3B* illustrates how $\varepsilon y_\nu$ changes as a function of $z$. It was generated using the following matrix of rate constants:

$$\vec{x} = \begin{pmatrix} 0 & 0.378 & 0.514 & 0.237 & 0 \\ 1.810 & 0 & 0 & 0 & 1.001 \\ 42.232 & 0 & 0 & z & 2.446 \\ 7.957 & 0 & z/2.44 & 0 & 0.259 \\ 0 & 6.982 & 0.994 & 0.257 & 0 \end{pmatrix}.$$

The Matlab code is available at https://github.com/skryazhi/epistasis_theory.

Next, I consider thress special cases of the toy network depicted in *Figure 3A* that relate this network to those in *Figure 3C and D*.

1. $z = 0$. According to *Equation 61*, $\varepsilon y_\mu = 0$ and hence $\varepsilon y_\nu = H/2C^2 \leq 0$, with $C$ and $H$ given by *Equation 62* and *Equation 63*. It is easy to see that in this case the reactions $1 \leftrightarrow 3$ and $2 \leftrightarrow 4$ that are affected by mutations are strictly parallel because there is a simple path $1 \leftrightarrow 3 \leftrightarrow 2 \leftrightarrow 5$ that contains only reaction $1 \leftrightarrow 3$ and there is a simple path $1 \leftrightarrow 4 \leftrightarrow 2 \leftrightarrow 5$ that contains only reaction $2 \leftrightarrow 4$ (*Figure 3C*).
2. $x_{15} = x_{32} = x_{14} = 0$. In this case, module $\nu$ is a linear pathway. Therefore, $\varepsilon y_\nu = 1$ independently of $z$, as shown above. This fact can also be obtained directly from *Equation 61*, *Equation 62*, *Equation 63* and *Equation 6*.
3. $z \to \infty$. In this case, module $\mu$ becomes an effectively linear pathway because most of the metabolic flux between the I/O metabolites 1 and 2 passes through reaction $3 \leftrightarrow 4$. Thus, it follows from *Equation 61* that $\varepsilon y_\mu \to \tilde{a}/(\tilde{c}_1 \tilde{c}_2) = 1$, as expected. Then, according to Theorem 1, $\varepsilon y_\nu \geq 1$.

## Proof of Theorem 1

As discussed above, the key step toward the proof is Proposition 2, which states that the function $f_1$ belongs to one of three parameteric families, given by *Equation 29*, *Equation 30*, *Equation 31*. To complete the proof, I now explicitly evalute the control coefficient $C$ and the $H$ in *Equation 6* for each of these functions and show that the inequalities in *Equation 7* and *Equation 8* hold for all parameter values.

### Proof of Theorem 1

Let $a$ be the effective reaction within higher-level module $\nu$ that represents the lower-level module $\mu$. To simplify notations, I denote $y_\mu \equiv \xi$. According to Proposition 2, the functional from of $f_1$ depends only on the topological class of the single-marked module $(\nu, a)$. So, I consider the three classes one by one.

Case $(\nu, a) \in \mathcal{M}^{\mathrm{b}}$. From *Equation 29*, $C = \xi/y_\nu$ and $H = 0$. Therefore, inequalities in *Equation 7* and *Equation 8* hold.

Case $(\nu, a) \in \mathcal{M}^{\mathrm{io}}$. From *Equation 30*,

$$C = \frac{\xi}{y_\nu}\left(\frac{w_{32}}{D}\right)^2, \tag{64}$$

$$H = -2\frac{\xi^2}{y_\nu}\frac{w_{32}^2}{D^3}\frac{1}{K_{13}} = -2\,C\,\frac{\xi/K_{13}}{D}, \tag{65}$$

where $D = (\xi + \alpha)/K_{13} + w_{32}$. From *Equation 64*, it is clear that $C \geq 0$. Re-writing *Equation 64* as

$$C = \left(\frac{\xi w_{32}/D}{y_\nu}\right)\left(\frac{w_{32}}{D}\right)$$

it is also clear that $C \leq 1$ since both ratios in this expression do not exceed 1. From *Equation 65* and the fact that $0 \leq C \leq 1$, it follows that $\bar{\varepsilon} \geq 0$. To show that $\bar{\varepsilon} \leq 1$, note that

$$D(1 - C) = \frac{\xi + \alpha}{K_{13}} + w_{32}\left(1 - \frac{\xi w_{32}/D}{y_\nu}\right) \geq \frac{\xi}{K_{13}}.$$

Therefore,

$$\bar{\varepsilon} = \frac{\xi/K_{13}}{D(1 - C)} \leq 1.$$

Therefore, inequalities in *Equation 7* and *Equation 8* hold.

Case $(\nu, a) \in \mathcal{M}^{\mathrm{i}}$. I re-write *Equation 31* as

$$y_\nu = w_{12} + \frac{\tilde{A}u + \tilde{B}}{D},$$

with $\quad u = \xi + \alpha, \quad\quad D = \tilde{C}u + \tilde{D}, \quad\quad \tilde{A} = (w_{13} + w_{14})(w_{42} + w_{32}/K_{34})$, $\tilde{B} = (w_{31} + w_{32})w_{14}w_{42} + (w_{41} + w_{42})w_{13}w_{32}$, $\tilde{C} = (w_{31} + w_{32})/K_{34} + (w_{41} + w_{42})$, $\tilde{D} = (w_{31} + w_{32})(w_{41} + w_{42})$, which yields

$$C = \frac{\xi}{y_\nu}\frac{\tilde{A}\tilde{D} - \tilde{B}\tilde{C}}{D^2}, \tag{66}$$

$$H = -2\frac{\xi^2}{y_\nu}\frac{(\tilde{A}\tilde{D} - \tilde{B}\tilde{C})\tilde{C}}{D^3} = -2C\frac{\tilde{C}\xi}{D}. \tag{67}$$

Next, it is straightforward to show that $\tilde{A}\tilde{D} - \tilde{B}\tilde{C} = (w_{41}w_{32} - w_{31}w_{42})^2/K_{31} \geq 0$, which implies that $C \geq 0$. To show that $C \leq 1$, I expand the denominator in *Equation 66* and obtain

$$y_\nu D^2 \geq (\tilde{A}u + \tilde{B})(\tilde{C}u + \tilde{D}) \geq u(\tilde{A}\tilde{D} + \tilde{B}\tilde{C}) \geq \xi(\tilde{A}\tilde{D} - \tilde{B}\tilde{C}).$$

Therefore, numerator in *Equation 66* cannot exceed the denominator. The fact that $\bar{\varepsilon} \geq 0$ follows directly from *Equation 67* together with $C \leq 1$. To show that $\bar{\varepsilon} \leq 1$, first note that

$$y_\nu = w_{12} + \frac{\tilde{A}\xi}{D} + \frac{\tilde{A}\alpha + \tilde{B}}{D} \geq \frac{\tilde{A}\xi}{D}.$$

Therefore,

$$D(1 - C) = \tilde{C}\xi + \tilde{C}\alpha + \tilde{D}\left(1 - \frac{\tilde{A}\xi/D}{y_\nu}\right) + \frac{\xi}{D}\frac{\tilde{B}\tilde{C}}{y_\nu} \geq \tilde{C}\xi.$$

Hence,

$$\bar{\varepsilon} = \frac{\tilde{C}\xi}{D(1 - C)} \leq 1.$$

Therefore, inequalities in *Equation 7* and *Equation 8* hold in this case as well, which completes the proof.

## Proof of Theorem 2

Proving Theorem 2 involves several auxiliary steps. First, I note that any two reactions $a$ and $b$ within any module $\mu$ can be either strictly serial, strictly parallel or serial-parallel. Then, Proposition 4 and its Corollary 4 establish that strictly parallel (serial) reactions in $(\mu, a, b)$ are also strictly parallel (serial) in a minimal module $(\mu', a, b)$, which is obtained from $\mu$ by eliminating all metabolites that do not participate in the marked reactions. Next, recall that in both modules $(\mu, a, b)$ and $(\mu', a, b)$ the same function $f_2$ maps the rate constants of two marked reactions onto module's effective rate constant (Proposition 3). Since the epistasis coefficient depends only on the shape of this function, we only need to consider all minimal modules in order to understand what kinds of epistasis may arise between mutations affecting strictly serial and strictly parallel reactions in any module. This is a big simplification because the number of different minimal topologies is finite and the parameteric families of function $f_2$ are known for all of them (see Corollary 3). As a consequence, to prove Theorem 2, we could in principle list all of the minimal topologies, identify those where the marked reactions are strictly serial or strictly parallel and evaluate the epistasis coefficient using *Equation 9* for every respective function $f_2$.

Unfortunately, the number of minimal topologies is very large, so that such brute-force approach would be quite cumbersome. I take a less cumbersome approach which is based on the realization that a strictly serial or strictly parallel relationship between two reactions cannot be altered by removing a third reaction from the module (Proposition 5). This implies that every minimal topology where the two reactions are strictly serial can be produced from another, more connected, 'generating' topology by removing some reactions; and similarly for minimal modules where the reactions are strictly parallel (Proposition 6). All generating topologies can be identified by a simple algorithm given in Appendix 3.

Finally, I prove Theorem 2 in three steps. First, Proposition 7 shows that $\varepsilon y_\mu \leq 0$ for any minimal module μ with any strictly parallel generating topology. Second, Proposition 8 shows that that $\varepsilon y_\mu \geq 1$ for any minimal module μ with any strictly serial generating topology. Third, the proof of Theorem 2 formally extends this argument to all modules with strictly serial and strictly parallel reactions.

## Topological relationships between reactions within a module

Consider module μ with the I/O metabolites 1 and 2. As described above, a simple path connecting two metabolites $i$ and $j$ within module μ is denoted by $p_{ij}^\mu = i \leftrightarrow k \leftrightarrow \ldots \leftrightarrow \ell \leftrightarrow j$. If such path contains reactions $a_1, a_2, \ldots$ and does not contain reactions $b_1, b_2, \ldots$, I denote it as $p_{ij}^\mu(a_1, a_2, \ldots, \bar{b}_1, \bar{b}_2, \ldots)$. I denote the set of all paths $p_{ij}^\mu(a_1, a_2, \ldots, \bar{b}_1, \bar{b}_2, \ldots)$ by $\mathcal{P}_{ij}^\mu(a_1, a_2, \ldots, \bar{b}_1, \bar{b}_2, \ldots)$.

According to Lemma 1 proven in Appendix 2, any reaction in the module belongs to at least one simple path within module μ that connects the I/O metabolites. Mathematically, $\mathcal{P}_{12}^\mu(a) \neq \emptyset$ for any reaction $a \in R_\mu$. Thus, we can define different topological relationships between any two reactions within a module based on the existence of various paths that do or do not contain them. Consequently, we can classify any double-marked module $(\mu, a, b)$ based on the toplogocial relationship between its marked reactions. This classification is orthogonal to that given in *Table 1*.

Two reactions $a \in R_\mu$ and $b \in R_\mu$ are called *parallel* within module μ if there exists a simple path $p_{12}^\mu(a, \bar{b})$ and a simple path $p_{12}^\mu(b, \bar{a})$. Two reactions $a \in R_\mu$ and $b \in R_\mu$ are called *serial* within module μ if there exist at least one simple path $p_{12}^\mu(a, b)$. Two reactions are called *strictly parallel* within module μ if they are parallel but not serial, they are called *strictly serial* within module μ if they are serial but not parallel, and they are called *serial-parallel* within module μ if they are both serial and parallel. It is straightforward to show that there are no other logical possibilities for any two reactions to be anything other than strictly serial, strictly parallel or serial-parallel. This implies that two reactions are strictly parallel if they are not serial, and they are strictly serial if they are not parallel. If reactions $a$ and $b$ are serial, parallel, strictly serial, strictly parallel or serial-parallel within module μ, I also refer to the double-marked module $(\mu, a, b)$ as serial, parallel, etc. Since the relationship between reactions depends only on the topology of the module, but not on its rate constants, I also refer to the topology $(R_\mu, a, b)$ as serial, parallel, etc.

Recall that coarse-graining procedure $\mathrm{CG}^{\mu \backslash \{a, b\}}$ eliminates all metabolites internal to module μ other than those participating in reactions $a$ and $b$. If the double-marked module $(\mu, a, b)$ belongs to the topological class $\mathcal{M}$, then, according to Proposition 3, $\mathrm{CG}^{\mu \backslash \{a, b\}}$ maps $(\mu, a, b)$ onto a minimal double-marked module $(\mu', a, b)$ from the same class $\mathcal{M}$. The following proposition, which is easy to prove using Property #9 of the CGP (see *Box 1*), establishes how this procedure alters the topological relationship between reactions $a$ and $b$.

## Proposition 4

Let $(\mu, a, b)$ be a double-marked module from the topological class $\mathcal{M}$ (*Table 1*) and let $(\mu', a, b)$ be the minimal double-marked module in $\mathcal{M}$ onto which $(\mu, a, b)$ is mapped by $\mathrm{CG}^{\mu \backslash \{a, b\}}$.

1. Reactions $a$ and $b$ are serial in $(\mu', a, b)$ if and only if they are serial in $(\mu, a, b)$.
2. If reactions $a$ and $b$ are parallel in $(\mu', a, b)$, then they are also parallel in $(\mu, a, b)$.

Note that the converse of the second claim in Proposition 4 is not true. In other words, if two reactions $a$ and $b$ are parallel in $(\mu, a, b)$, they may not be parallel in $(\mu', a, b)$. *Figure 9* shows a counter-example illustrating this.

## Corollary 4

1. If reactions $a$ and $b$ are strictly serial in $(\mu, a, b)$, they are also strictly serial in $(\mu', a, b)$.
2. If reactions $a$ and $b$ are strictly parallel in $(\mu, a, b)$, they are also strictly parallel in $(\mu', a, b)$.
3. If reactions $a$ and $b$ are serial-parallel in $(\mu, a, b)$, they are either strictly serial or serial-parallel in $(\mu', a, b)$.

**Figure 9.** A counter example illustrating that the converse to claim 2 in Proposition 4 may not be true. Reactions $a$ and $b$ are parallel in $(\mu, a, b)$. CGP maps the double-marked module $(\mu, a, b)$ onto the minimal double-marked module $(\mu', a, b)$ where reactions $a$ and $b$ are not parallel.

Corollary 4 is an important step toward proving Theorem 2. According to Proposition 3, the function that maps the rate constants $\xi$ and $\eta$ of the reactions $a$ and $b$ in module $\mu$ onto the effective rate constant $y_\mu$ is the same function that maps the rate constants $u$ and $v$ of these reactions in the minimal module $\mu'$ onto the effective rate constant $y_{\mu'}$. It then immediately follows from *Equation 9* that the epistasis coefficient $\varepsilon y_\mu$ between mutations affecting reactions $a$ and $b$ in the original module $\mu$ is the same as the epistasis coefficient $\varepsilon y_{\mu'}$ in the minimal module $\mu'$. Now, if the reactions $a$ and $b$ are strictly parallel in $(\mu, a, b)$, then, according to Corollary 4, these reactions are also strictly parallel in $(\mu', a, b)$. Hence, to demonstrate that $\varepsilon y_\mu \leq 0$ for any such double-marked module $(\mu, a, b)$, it is sufficient to show that $\varepsilon y_{\mu'} \leq 0$ for all double-marked modules $(\mu', a, b)$ that are minimal in $\mathcal{M}$ and where the reactions $a$ and $b$ are strictly parallel. And similarly for the strictly serial reactions.

According to this logic, Theorem 2 can be proven by identifying all double-marked modules that are minimal in each of the topological classes listed in *Table 1* and where the marked reactions are strictly parallel, evaluating epistasis for all of them, and showing that it is non-positive, irrespectively of the rate constants of any reactions, and similarly for the strictly serial reactions.

## Generating topologies

Since the number of topologically distinct minimal double-marked modules is finite, the approach outlined above is in principle feasible. Unfortunately, the number of topologies to be considered is very large, so in practice it is very cumbersome. To avoid this complication, I take an alternative approach that is based on the same key idea as the proof of Theorem 1. Rather than considering one by one, each minimal topology where the marked reactions are strictly serial or strictly parallel (and the corresponding parametric families of $f_2$), the idea is to identify the most connected minimal topologies (and the corresponding largest parametric families of $f_2$) such that all the other minimal topologies with the strictly serial or strictly parallel reactions (and the corresponding parametric families) can be obtained from them by removing reactions (i.e. setting some parameters to zero).

This idea can be implemented using the following observations. If the two marked reactions are strictly parallel or strictly serial in a minimal module, then removing a third reaction from it does not change this relationship. This statement is proven in Proposition 5. As a consequence, all minimal modules in the topological classes $\mathcal{M}^{b,io,IO}$, $\mathcal{M}^{b,i,\emptyset}$ and $\mathcal{M}^{io,io,IO}$ must be strictly parallel because the fully connected minimal topologies are strictly parallel (*Figure 8*). Similarly, all minimal modules in the topological class $\mathcal{M}^{io,io,I}$ must be strictly serial because the fully connected minimal topology is strictly serial (*Figure 8*). The fully connected minimal topologies in all other topological classes are serial-parallel. If the two reactions are serial-parallel, removing a third reaction can change their relationship into a strictly serial or strictly parallel one, depending on which reaction is removed, as shown for example in *Figure 3A,C and D*. In fact, by removing reactions from the fully connected minimal modules shown in *Figure 8*, it is easy to show that the topological classes $\mathcal{M}^{io,io,\emptyset}$, $\mathcal{M}^{io,i,I}$, $\mathcal{M}^{io,i,\emptyset}$, $\mathcal{M}^{i,i,I}$, $\mathcal{M}^{i,i,\emptyset}$ contain both strictly serial and strictly parallel modules.

These observations suggests that adding reactions to a minimal module where the marked reactions are strictly serial or strictly parallel will either change their relationship into serial-parallel or will preserve their relationship until the minimal module is fully connected. Therefore, among all minimal modules in a topological class, there must exist the most connected modules where the marked reactions are strictly parallel or strictly serial, such that all other less connected strictly serial or

strictly parallel modules can be produced from the most connected ones by removal of reactions. This statement is proven in Proposition 6. Such most connected strictly parallel and strictly serial minimal topologies, which I refer to as 'generating', are listed in *Table 2* and *Table 3*. They define the largest parametric families of functions $f_2$ which I then examine for the value of $\varepsilon y_\mu$.

## Proposition 5

Let $(\mu, a, b)$ and $(\mu', a, b)$ be two minimal double-marked modules from the same topological class whose sets of reactions are $R_\mu$ and $R_{\mu'}$, respectively, and $R_{\mu'} = R_\mu \setminus \{c\}$ where $c \in R_\mu \setminus \{a, b\}$.

1. If reactions $a$ and $b$ are strictly parallel in $(\mu, a, b)$, they are also strictly parallel in $(\mu', a, b)$.
2. If reactions $a$ and $b$ are strictly serial in $(\mu, a, b)$, they are also strictly serial in $(\mu', a, b)$.

### Proof

Denote the I/O metabolites in both modules μ and $\mu'$ by 1 and 2. Since $\mu'$ and μ are topologically identical except for $\mu'$ lacking one reaction $c$, it must be true that $\mathcal{P}_{12}^{\mu'}(a_1, a_2, \ldots, \bar{b}_1, \bar{b}_2, \ldots) \subseteq \mathcal{P}_{12}^{\mu}(a_1, a_2, \ldots, \bar{b}_1, \bar{b}_2, \ldots)$ for any reactions $a_1, a_2, \ldots, b_1, b_2, \ldots$ from $R_{\mu'}$. In other words, there could only be fewer paths connecting the I/O metabolites within module $\mu'$ compared to module μ. The rest of the proof follows immediately from this fact and the definitions of strictly serial and strictly parallel relatioships.

Next, I define a minimal topology as generating either if it is a fully connected topology (as in topological classes $\mathcal{M}^{b,io,IO}$, $\mathcal{M}^{b,i,\emptyset}$ and $\mathcal{M}^{io,io,IO}$, $\mathcal{M}^{io,io,I}$) or if adding any reaction to it would make the marked reactions serial-parallel.

Denote the sets of all double-marked topologies minimal in class $\mathcal{M}$ where the marked reactions are strictly serial, strictly parallel and serial-parallel by by $\mathcal{R}_{\mathcal{M}}^{ser}$, $\mathcal{R}_{\mathcal{M}}^{par}$ and $\mathcal{R}_{\mathcal{M}}^{sp}$, respectively.

**Table 2.** Strictly parallel generating topologies.

| Class | Marked reactions | | Generating topology | | Figure |
|---|---|---|---|---|---|
| | $a$ | $b$ | ID | Excluded reactions | |
| $\mathcal{M}^{b,io,IO}$ | $1 \leftrightarrow 2$ | $1 \leftrightarrow 3$ | b, io, IO, F | $\emptyset$ | *Figure 7* |
| $\mathcal{M}^{b,i,\emptyset}$ | $1 \leftrightarrow 2$ | $3 \leftrightarrow 4$ | b, i, $\emptyset$, F | $\emptyset$ | *Figure 7* |
| $\mathcal{M}^{io,io,IO}$ | $1 \leftrightarrow 3$ | $1 \leftrightarrow 4$ | io, io, IO, F | $\emptyset$ | *Figure 7* |
| $\mathcal{M}^{io,io,\emptyset}$ | $1 \leftrightarrow 3$ | $2 \leftrightarrow 4$ | io, io, $\emptyset$, P | $\{3 \leftrightarrow 4\}$ | *Figure 9* |
| $\mathcal{M}^{io,i,I}$ | $1 \leftrightarrow 3$ | $3 \leftrightarrow 4$ | io, i, I, P | $\{2 \leftrightarrow 4\}$ | *Figure 10* |
| $\mathcal{M}^{io,i,\emptyset}$ | $1 \leftrightarrow 3$ | $4 \leftrightarrow 5$ | io, i, $\emptyset$, $P_1$ | $\{3 \leftrightarrow 4, 3 \leftrightarrow 5\}$ | *Figure 11* |
| | | | io, i, $\emptyset$, $P_2$ | $\{2 \leftrightarrow 5, 3 \leftrightarrow 5\}$ | |
| | | | io, i, $\emptyset$, $P_3$ | $\{2 \leftrightarrow 4, 2 \leftrightarrow 5\}$ | |
| $\mathcal{M}^{i,i,I}$ | $3 \leftrightarrow 4$ | $3 \leftrightarrow 5$ | i, i, I, $P_1$ | $\{2 \leftrightarrow 4, 2 \leftrightarrow 5\}$ | *Figure 12* |
| | | | i, i, I, $P_2$ | $\{1 \leftrightarrow 5, 2 \leftrightarrow 5\}$ | |
| $\mathcal{M}^{i,i,\emptyset}$ | $3 \leftrightarrow 4$ | $5 \leftrightarrow 6$ | i, i, $\emptyset$, $P_1$ | $\{3 \leftrightarrow 5, 3 \leftrightarrow 6, 4 \leftrightarrow 5, 4 \leftrightarrow 6\}$ | *Figure 13* |
| | | | i, i, $\emptyset$, $P_2$ | $\{1 \leftrightarrow 5, 1 \leftrightarrow 6, 2 \leftrightarrow 5, 2 \leftrightarrow 6\}$ | |
| | | | i, i, $\emptyset$, $P_3$ | $\{2 \leftrightarrow 4, 2 \leftrightarrow 6, 3 \leftrightarrow 6, 4 \leftrightarrow 5, 4 \leftrightarrow 6\}$ | |
| | | | i, i, $\emptyset$, $P_4$ | $\{2 \leftrightarrow 4, 2 \leftrightarrow 5, 2 \leftrightarrow 6, 4 \leftrightarrow 5, 4 \leftrightarrow 6\}$ | |
| | | | i, i, $\emptyset$, $P_5$ | $\{1 \leftrightarrow 6, 2 \leftrightarrow 4, 2 \leftrightarrow 5, 2 \leftrightarrow 6, 4 \leftrightarrow 6\}$ | |
| | | | i, i, $\emptyset$, $P_6$ | $\{1 \leftrightarrow 4, 1 \leftrightarrow 6, 2 \leftrightarrow 4, 2 \leftrightarrow 6, 4 \leftrightarrow 6\}$ | |
| | | | i, i, $\emptyset$, $P_7$ | $\{1 \leftrightarrow 4, 1 \leftrightarrow 5, 2 \leftrightarrow 3, 2 \leftrightarrow 6, 3 \leftrightarrow 5, 4 \leftrightarrow 6\}$ | |

**Table 3.** Strictly serial generating topologies.

| Class | Marked reactions | | Generating topologies | | Figure |
|---|---|---|---|---|---|
| | $a$ | $b$ | ID | Excluded reactions | |
| $\mathcal{M}^{\mathrm{io,io,I}}$ | $1 \leftrightarrow 3$ | $2 \leftrightarrow 3$ | io, io, I, F | $\emptyset$ | *Figure 7* |
| $\mathcal{M}^{\mathrm{io,io,\emptyset}}$ | $1 \leftrightarrow 3$ | $2 \leftrightarrow 4$ | io, io, $\emptyset$, S | $\{2 \leftrightarrow 3\}$ | *Figure 9* |
| $\mathcal{M}^{\mathrm{io,i,I}}$ | $1 \leftrightarrow 3$ | $3 \leftrightarrow 4$ | io, i, I, $S_1$ | $\{2 \leftrightarrow 3\}$ | *Figure 10* |
| | | | io, i, I, $S_2$ | $\{1 \leftrightarrow 4\}$ | |
| $\mathcal{M}^{\mathrm{io,i,\emptyset}}$ | $1 \leftrightarrow 3$ | $4 \leftrightarrow 5$ | io, i, $\emptyset$, $S_1$ | $\{1 \leftrightarrow 4, 1 \leftrightarrow 5\}$ | *Figure 11* |
| | | | io, i, $\emptyset$, $S_2$ | $\{2 \leftrightarrow 3, 2 \leftrightarrow 4, 3 \leftrightarrow 5\}$ | |
| | | | io, i, $\emptyset$, $S_3$ | $\{1 \leftrightarrow 5, 2 \leftrightarrow 3, 2 \leftrightarrow 5\}$ | |
| $\mathcal{M}^{\mathrm{i,i,I}}$ | $3 \leftrightarrow 4$ | $3 \leftrightarrow 5$ | i, i, I, $S_1$ | $\{1 \leftrightarrow 3, 2 \leftrightarrow 3\}$ | *Figure 12* |
| | | | i, i, I, $S_2$ | $\{2 \leftrightarrow 3, 2 \leftrightarrow 5, 4 \leftrightarrow 5\}$ | |
| $\mathcal{M}^{\mathrm{i,i,\emptyset}}$ | $3 \leftrightarrow 4$ | $5 \leftrightarrow 6$ | i, i, $\emptyset$, $S_1$ | $\{2 \leftrightarrow 3, 2 \leftrightarrow 5, 2 \leftrightarrow 6, 4 \leftrightarrow 5, 4 \leftrightarrow 6\}$ | *Figure 13* |
| | | | i, i, $\emptyset$, $S_2$ | $\{1 \leftrightarrow 3, 1 \leftrightarrow 6, 2 \leftrightarrow 3, 2 \leftrightarrow 6, 4 \leftrightarrow 6\}$ | |

### Definition 2
Topology $(R, a, b)$ minimal in $\mathcal{M}$ is called a strictly serial generating topology in $\mathcal{M}$ if it is strictly serial (i.e. $(R, a, b) \in \mathcal{R}_{\mathcal{M}}^{\mathrm{ser}}$) and either it is fully connected (i.e. $R = R_{\mathcal{M}}$) or $(R \cup \{c\}, a, b) \in \mathcal{R}_{\mathcal{M}}^{\mathrm{sp}}$ for any reaction $c \in R_{\mathcal{M}} \setminus R$.

### Definition 3
Topology $(R, a, b)$ minimal in $\mathcal{M}$ is called a strictly parallel generating topology in $\mathcal{M}$ if it is strictly parallel (i.e. $(R, a, b) \in \mathcal{R}_{\mathcal{M}}^{\mathrm{par}}$) and either it is fully connected (i.e. $R = R_{\mathcal{M}}$) or $(R \cup \{c\}, a, b) \in \mathcal{R}_{\mathcal{M}}^{\mathrm{sp}}$ for any reaction $c \in R_{\mathcal{M}} \setminus R$.

Clearly, a topological class $\mathcal{M}$ may have multiple generating topologies, and it is easy to show that every topological class has at least one generating topology. I denote the set of all strictly serial generating topologies for the class $\mathcal{M}$ by $\mathcal{G}_{\mathcal{M}}^{\mathrm{ser}}$ and I denote the set of all strictly parallel generating topologies for class $\mathcal{M}$ by $\mathcal{G}_{\mathcal{M}}^{\mathrm{par}}$. The following proposition justifies the name 'generating topology'. It states that any strictly serial minimal topology can be produced from some strictly serial generating topology by removing one or multiple reactions, and similarly for any strictly parallel minimal topology.

### Proposition 6
If $(R, a, b)$ is a strictly parallel topology minimal in the topological class $\mathcal{M}$, then there exists a strictly parallel generating topology $(R_g, a, b)$ in $\mathcal{M}$, such that $R \subseteq R_g$. If $(R, a, b)$ is a strictly serial topology minimal in the topological class $\mathcal{M}$, then there exists a strictly serial generating topology $(R_g, a, b)$ in $\mathcal{M}$, such that $R \subseteq R_g$.

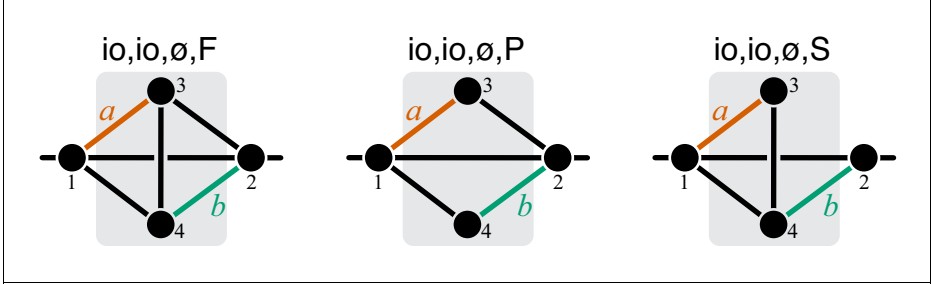

**Figure 10.** Graphical representation of strictly serial and strictly parallel generating topologies in the class $\mathcal{M}^{\mathrm{io,io,\emptyset}}$. Fully connected topology io, io, $\emptyset$, F is shown for reference (same as in *Figure 8*).

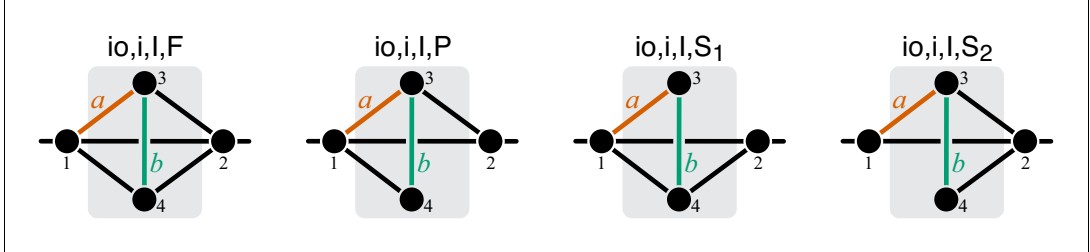

**Figure 11.** Graphical representation of strictly serial and strictly parallel generating topologies in class $\mathcal{M}^{\mathrm{io,i,I}}$. Fully connected topology $\mathrm{io,i,I,F}$ is shown for reference (same as in **Figure 8**).

## Proof

Suppose that $\mathcal{M}$ is one of the topological classes $\mathcal{M}^{\mathrm{b,io,IO}}$, $\mathcal{M}^{\mathrm{b,i,\emptyset}}$, or $\mathcal{M}^{\mathrm{io,io,IO}}$. Since the fully connected minimal topology $(R_{\mathcal{M}}, a, b)$ is strictly parallel, it is a generating topology in $\mathcal{M}$. Then, according to Proposition 5, any topology $(R, a, b)$ minimal in $\mathcal{M}$ is strictly parallel, and Proposition 6 holds. By the same logic, Proposition 6 holds for the topological class $\mathcal{M}^{\mathrm{io,io,I}}$.

Suppose that $\mathcal{M}$ is one of the remaining topological classes $\mathcal{M}^{\mathrm{io,io,\emptyset}}$, $\mathcal{M}^{\mathrm{io,i,I}}$, $\mathcal{M}^{\mathrm{io,i,\emptyset}}$, $\mathcal{M}^{\mathrm{i,i,I}}$ or $\mathcal{M}^{\mathrm{i,i,\emptyset}}$. Then the fully connected minimal topology $(R_{\mathcal{M}}, a, b)$ is serial-parallel. Suppose that $(R, a, b)$ is strictly parallel. Then $R$ must be a strict subset of $R_{\mathcal{M}}$, so that the set $C = R_{\mathcal{M}} \setminus R$ is not empty. Then, either $(R, a, b) \in \mathcal{G}_{\mathcal{M}}^{\mathrm{par}}$ or $(R, a, b) \notin \mathcal{G}_{\mathcal{M}}^{\mathrm{par}}$. If $(R, a, b) \in \mathcal{G}_{\mathcal{M}}^{\mathrm{par}}$, the Proposition 6 holds. Suppose that $(R, a, b) \notin \mathcal{G}_{\mathcal{M}}^{\mathrm{par}}$. This implies that there exists a reaction $c_1 \in C$, such that $R_1 = R \cup \{c_1\}$ and $(R_1, a, b) \in \mathcal{R}_{\mathcal{M}}^{\mathrm{par}}$ ($(R_1, a, b)$ cannot be in $\mathcal{R}_{\mathcal{M}}^{\mathrm{ser}}$ due to Proposition 5). There are three possibilities.

1. $R_1 = R_{\mathcal{M}}$.
2. $R_1 \subset R_{\mathcal{M}}$ and $(R_1, a, b) \in \mathcal{G}_{\mathcal{M}}^{\mathrm{par}}$.
3. $R_1 \subset R_{\mathcal{M}}$ and $(R_1, a, b) \notin \mathcal{G}_{\mathcal{M}}^{\mathrm{par}}$.

Option (a) is not possible since $(R_1, a, b) \in \mathcal{R}_{\mathcal{M}}^{\mathrm{par}}$ while $(R_{\mathcal{M}}, a, b) \in \mathcal{R}_{\mathcal{M}}^{\mathrm{sp}}$. Option (b) implies that the Proposition 6 holds. Option (c) implies that there exists a reaction $c_2 \in C \setminus \{c_1\}$, such that $R_2 = R_1 \cup \{c_2\}$ and $(R_2, a, b) \in \mathcal{R}_{\mathcal{M}}^{\mathrm{par}}$, and we have the same three possibilities for $R_2$ as above. Thus, by induction, Proposition 6 must hold. The proof is analogous if $(R, a, b)$ is strictly serial.

Discovering all strictly serial and strictly parallel generating topologies in any given topological class $\mathcal{M}$ is straightforward because all minimal topologies within $\mathcal{M}$ can be produced by removing reactions from the unique fully connected topology minimal in $\mathcal{M}$ shown in **Figure 8**. In Appendix 3,

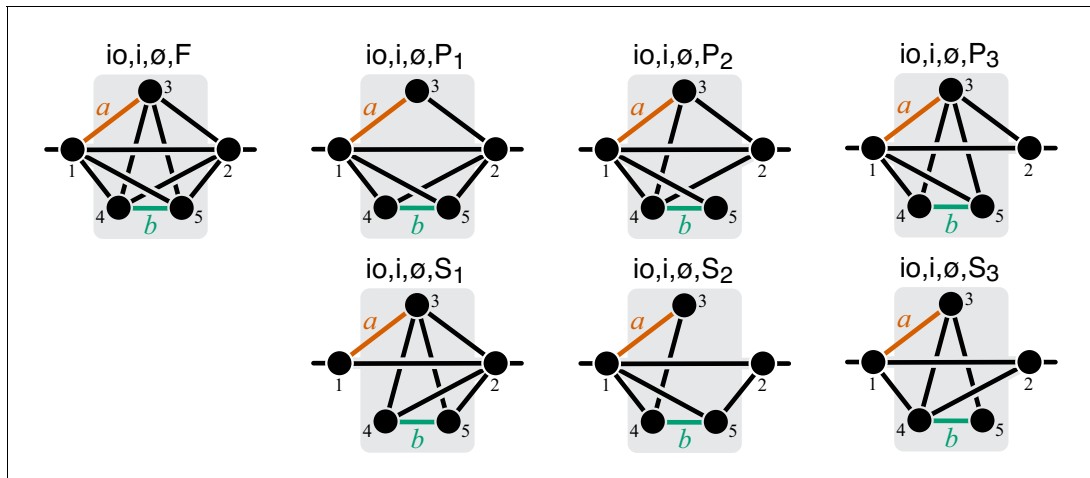

**Figure 12.** Graphical representation of strictly serial and strictly parallel generating topologies in class $\mathcal{M}^{\mathrm{io,i,\emptyset}}$. Fully connected topology $\mathrm{io,i,\emptyset,F}$ is shown for reference (same as in **Figure 8**).

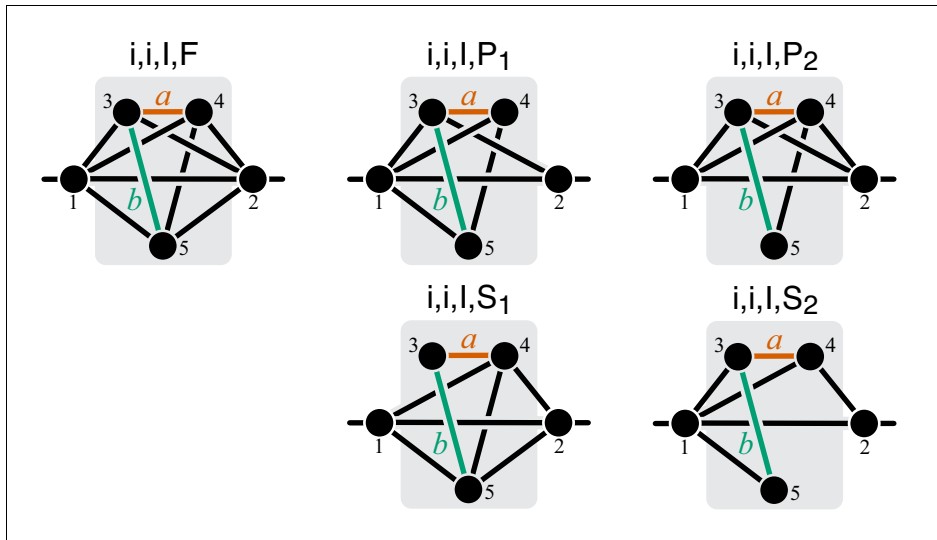

**Figure 13.** Graphical representation of strictly serial and strictly parallel generating topologies in class $\mathcal{M}^{\mathrm{i,i,I}}$. Fully connected topology $\mathrm{i, i, I, F}$ is shown for reference (same as in *Figure 8*).

I provide an algorithm that discovers all strictly serial and strictly parallel generating topologies by sequentially removing reactions from the fully connected minimal topology in each topological class. The code implementing this algorithm in Matlab is available at https://github.com/skryazhi/epistasis_theory. All strictly parallel generating topologies are listed in *Table 2* and all strictly serial generating topologies are listed in *Table 3*. They are also illustrated in *Figure 8* and *Figure 10* through Figure 14. I label each generating topology by a four-letter combination (see column 4 in *Table 2* and *Table 3*): the first three letters denote the topological class and the last letter (F, P, or S) denotes the specific generating topology within that class. Letter 'F' (stands for 'full') denotes the fact that the reaction set in the generating topology is complete. Letters 'P' (for 'parallel') and 'S' (stands for 'serial') denote strictly parallel and strictly serial generating topologies, respectively; if there are a multiple generating topologies within the same class, they are distinguished by subindices, for example, $\mathrm{io, i, \emptyset, P_1}$; $\mathrm{io, i, \emptyset, P_2}$, etc.

### Topological relationship between reactions and epistasis

Each strictly serial and strictly parallel generating topology in a given class $\mathcal{M}$ (listed in *Table 2* and *Table 3*) is produced by removing reactions from the fully connected topology minimal in $\mathcal{M}$ (shown in *Figure 8*). This implies that the parametric family of function $f_2$ that corresponds to any generating topology is obtained from *Equation 34* through *Equation 35* by setting some parameters $w_{ij}$ to zero. In other words, these parametric families are known. Next, I prove Proposition 7 that shows that $\varepsilon y_\mu \leq 0$ for every member of every parameteric family of $f_2$ that corresponds to a strictly parallel generating topology and the analogous Proposition 8 for strictly serial topologies.

Now, any minimal strictly parallel topology can in turn be produced by removing reactions from some strictly parallel generating topology, and any minimal strictly serial topology can be produced by removing reactions from some strictly serial generating topology. This implies that any function $f_2$ that corresponds to any strictly parallel minimal topology belongs to the parametric family that corresponds to some strictly parallel generating topology; and any function $f_2$ that corresponds to any strictly serial minimal topology belongs to the parametric family that corresponds to some strictly serial generating topology. Therefore, Propositions 7 and 8 imply that $\varepsilon y_\mu \leq 0$ for *any* minimal strictly parallel topology and that $\varepsilon y_\mu \geq 1$ for *any* minimal strictly serial topology. The proof of Theorem 2 is then concluded by recalling that every strictly parallel module is mapped onto its effective rate constant via function $f_2$ that corresponds to some minimal strictly parallel module, and similarly for strictly serial modules.

## Proposition 7

Let $(\mu, a, b)$ be a minimal double-marked module in the topological class $\mathcal{M}$, with $u$ and $v$ being the rates of reactions $a$ and $b$, respectively, and let $y$ be the effective rate constant of this module. Suppose that mutation $A$ perturbs only reaction $a$ by $\delta^A u$, and mutation $B$ perturbs only reaction $b$ by $\delta^B v$, such that $|\delta^A u| \ll 1$, $|\delta^B v| \ll 1$. If reactions $a$ and $b$ are strictly parallel in $(\mu, a, b)$, then $\varepsilon y \le 0$.

## Proof

According to Proposition 6, the topology of module $(\mu, a, b)$ can be produced by removing reactions from some strictly parallel generating topology $(R_g, a, b)$. Therefore, the function $f_2$ that maps $u$ and $v$ in this module onto its effective rate constant $y$ belongs to the parametric family that corresponds to $(R_g, a, b)$. According to **Equation 55**,

$$\varepsilon y = \frac{H_{uv}}{2\,C_u\,C_v} + o(1), \tag{68}$$

where

$$C_u = \frac{u}{y}\frac{\partial f_2}{\partial u}, \tag{69}$$

$$C_v = \frac{v}{y}\frac{\partial f_2}{\partial v}, \tag{70}$$

$$H_{uv} = \frac{u\,v}{y}\frac{\partial^2 f_2}{\partial u\,\partial v}. \tag{71}$$

According to Theorem 1, $0 \le C_u \le 1$ and $0 \le C_v \le 1$. And since all $y>0$, $u>0$, $v>0$, to prove Proposition 7, it is sufficient to show that $\frac{\partial^2 f_2}{\partial u\,\partial v} \le 0$ for any member of any parametric family that corresponds to generating topologies listed in **Table 2**.

Generating topologies $\mathrm{b, io, IO, F}$ and $\mathrm{b, i, \emptyset, F}$ (**Figure 8**). According to **Equation 34** and **Equation 35**, $y = f_2(u, v) = u + \phi(v)$, which implies that $\varepsilon y = 0$.

Generating topology $\mathrm{io, io, IO, F}$ (**Figure 8**). According to **Equation 37**,

$$y = f_2(u, v) = w_{12} + \frac{A\,u\,v + B\,u + B\,v}{D},$$

where $D = E\,u\,v + F\,u + G\,v + B$, $A = w_{42}/K_{13} + w_{32}/K_{14}$, $B = w_{32}\,w_{42} + w_{32}\,w_{43} + w_{34}\,w_{42}$, $E = 1/(K_{13}\,K_{14})$, $F = (w_{42} + w_{43})/K_{13}$, $G = (w_{32} + w_{34})/K_{14}$. Therefore,

$$\frac{\partial^2 f_2}{\partial u\,\partial v} = -2\frac{w_{34}}{K_{14}}\frac{(w_{32}\,v/K_{14} + B)\,(w_{42}\,u/K_{13} + B)}{D^3} \le 0.$$

Generating topology $\mathrm{io, io, \emptyset, P}$ (**Figure 10**). According to **Equation 38**, $y = f_2(u, v) = w_{12} + \phi(u) + \psi(v)$, which implies $\varepsilon y = 0$.

Generating topology $\mathrm{io, i, I, P}$ (**Figure 11**). Notice that metabolite 4 together with reactions $1 \leftrightarrow 4$, $a$ and $b$ form a double-marked module $(\mu', a, b)$ whose I/O metabolites are 1 and 3 and which is minimal in the topological calss $\mathcal{M}^{\mathrm{b, io, IO}}$. Denote the effective reaction rate of module $\mu'$ by $y'$. Therefore, $\varepsilon y' = 0$, as shown above. Since module $\mu'$ is contained in $\mu$, by Theorem 1, $\varepsilon y \le 0$.

Generating topology $\mathrm{io, i, \emptyset, P_1}$ (**Figure 12**). According to Property 1 of the CGP (**Box 1**), module $\mu$ can be coarse-grained by first eliminating metabolite 3. In the resulting module $\mu'$, mutation $A$ perturbs only the rate constant $u'$ of the effective reaction $a' \equiv 1 \leftrightarrow 2$ (by Properties 2 and 4 of the CGP). Then, according to **Equation 50** and Theorem 1, $|\delta^A u'| \ll 1$. The double-marked module $(\mu', a', b)$ is minimal in the topological class $\mathcal{M}^{\mathrm{b, i, \emptyset}}$ which implies that $\varepsilon y = 0$, as shown above.

Generating topology $\mathrm{io, i, \emptyset, P_2}$ (**Figure 12**). Module $\mu$ can be coarse-grained by first eliminating metabolite 5, which will result in a double-marked module $(\mu', a, b')$ that is minimal in the topological class $\mathcal{M}^{\mathrm{io, io, IO}}$. The rest of the proof for this topology is analogous to that for the topology $\mathrm{io, i, \emptyset, P_1}$.

Generating topology $\mathrm{io,i,\emptyset,P_3}$ (*Figure 12*). Notice that metabolites 4 and 5 together with reactions $a$, $b$, $1 \leftrightarrow 4$, $1 \leftrightarrow 5$, $3 \leftrightarrow 4$ and $3 \leftrightarrow 5$ form a double-marked module $(\mu', a, b)$ whose I/O metabolites are 1 and 3 and which is minimal in the topological calss $\mathcal{M}^{\mathrm{b,i,\emptyset}}$. The rest of the proof for this topology is analogous to that for the topology $\mathrm{io,i,I,P}$.

Generating topology $\mathrm{i,i,I,P_1}$ (*Figure 13*). Notice that metabolites 4 and 5 together with reactions $a$, $b$, $1 \leftrightarrow 3$, $1 \leftrightarrow 4$, $1 \leftrightarrow 5$ and $4 \leftrightarrow 5$ form a double-marked module $(\mu', a, b)$ whose I/O metabolites are 1 and 3 and which is minimal in the topological calss $\mathcal{M}^{\mathrm{io,io,IO}}$. The rest of the proof for this topology is analogous to that for the topology $\mathrm{io,i,I,P}$.

Generating topology $\mathrm{i,i,I,P_2}$ (*Figure 13*). Notice that metabolite 5 together with reactions $a$, $b$, and $4 \leftrightarrow 5$ form a double-marked module $(\mu', a, b)$ whose I/O metabolites are 3 and 4 and which is minimal in the topological calss $\mathcal{M}^{\mathrm{b,io,IO}}$. The rest of the proof for this topology is analogous to that for the topology $\mathrm{io,i,I,P}$.

Generating topology $\mathrm{i,i,\emptyset,P_1}$ (*Figure 14*). According to *Equation 35*, $y = f_2(u,v) = x_{12} + \phi(u) + \psi(v)$, which implies $\varepsilon y = 0$.

Generating topology $\mathrm{i,i,\emptyset,P_2}$ (*Figure 14*). Notice that metabolites 5 and 6 together with reactions $a$, $b$, $3 \leftrightarrow 5$, $3 \leftrightarrow 6$, $4 \leftrightarrow 5$ and $5 \leftrightarrow 6$ form a double-marked module $(\mu', a, b)$ whose I/O metabolites are 3 and 4 and which is minimal in the topological calss $\mathcal{M}^{\mathrm{b,i,\emptyset}}$. The rest of the proof for this topology is analogous to that for the topology $\mathrm{io,i,I,P}$.

Generating topology $\mathrm{i,i,\emptyset,P_3}$ (*Figure 14*). Module μ can be coarse-grained by first eliminating metabolites 4 and 6, which will result in a double-marked module $(\mu', a', b')$ that is minimal in the topological class $\mathcal{M}^{\mathrm{io,io,IO}}$. The rest of the proof for this topology is analogous to that for the topology $\mathrm{io,i,\emptyset,P_1}$.

Generating topology $\mathrm{i,i,\emptyset,P_4}$ (*Figure 14*). Module μ can be coarse-grained by first eliminating metabolite 4, which will result in a double-marked module $(\mu', a', b)$ that is minimal in the topological class $\mathcal{M}^{\mathrm{io,i,\emptyset}}$ with a strictly parallel generating topology $\mathrm{io,i,\emptyset,P_3}$. The rest of the proof for this topology is analogous to that for the topology $\mathrm{io,i,\emptyset,P_1}$.

Generating topology $\mathrm{i,i,\emptyset,P_5}$ (*Figure 14*). Module μ can be coarse-grained by first eliminating metabolite 6, which will result in a double-marked module $(\mu', a, b')$ that is minimal in the topological

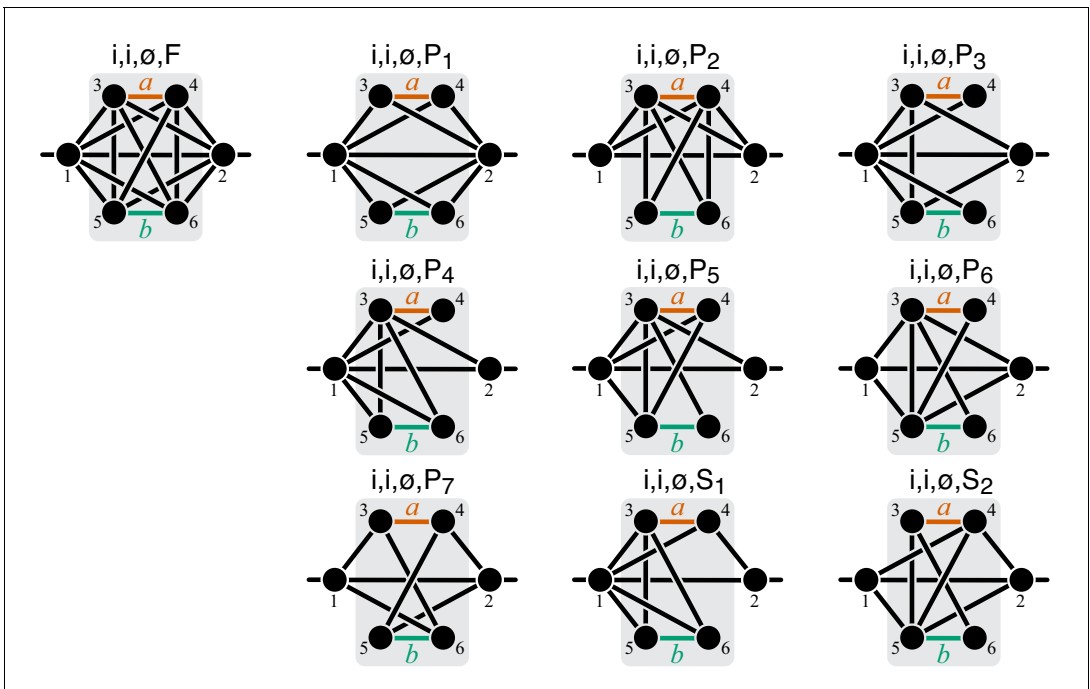

**Figure 14.** Graphical representation of strictly serial and strictly parallel generating topologies in class $\mathcal{M}^{\mathrm{i,i,\emptyset}}$. Fully connected topology $\mathrm{i,i,\emptyset,F}$ is shown for reference (same as in *Figure 8*).

class $\mathcal{M}^{\mathrm{i,i,I}}$ with a strictly parallel generating topology $\mathrm{i, i, I, P_1}$. The rest of the proof for this topology is analogous to that for the topology $\mathrm{io, i, \emptyset, P_1}$.

Generating topology $\mathrm{i, i, \emptyset, P_6}$ (**Figure 14**). Notice that metabolites 4 and 6 together with reactions $a$, $b$, $3 \leftrightarrow 5$, $3 \leftrightarrow 6$, $4 \leftrightarrow 5$ form a double-marked module $(\mu', a, b)$ whose I/O metabolites are 3 and 5 and which is minimal in the topological calss $\mathcal{M}^{\mathrm{io,io,\emptyset}}$. The rest of the proof for this topology is analogous to that for the topology $\mathrm{io, i, I, P}$.

Generating topology $\mathrm{i, i, \emptyset, P_7}$ (**Figure 14**). Using **Equation 42**, I show in Appendix 4 that

$$\frac{\partial^2 f_2}{\partial u \, \partial v} = \frac{2\beta}{K_{31}} \frac{(A_u v + B_u)(A_v u + B_v)}{(E_u v + F_u)^3}, \tag{72}$$

where $A_v$, $B_v$, $E_u$, $F_u$ are all non-negative constants and $\beta \leq 0$.

## Proposition 8

Let $(\mu, a, b)$ be a minimal double-marked module in the topological class $\mathcal{M}$, with $u$ and $v$ being the rates of reactions $a$ and $b$, respectively, and let $y$ be the effective rate constant of this module. Suppose that mutation A perturbs only reaction $a$ by $\delta^A u$, and mutation B perturbs only reaction $b$ by $\delta^B v$, such that $|\delta^A u| \ll 1$, $|\delta^B v| \ll 1$. If reactions $a$ and $b$ are strictly serial in $(\mu, a, b)$, then $\varepsilon y \geq 1$.

### Proof

The logic of the proof is the same as for Proposition 7, that is, it is sufficient to show that $\varepsilon y \geq 1$ for any double-marked module $(\mu, a, b)$ with any strictly serial generating topology listed in **Table 3**.

Generating topology $\mathrm{io, io, I, F}$ (**Figure 8**). According to **Equation 36**,

$$y = f_2(u, v) = w_{12} + \frac{uv}{D} \tag{73}$$

where $D = u/K_{13} + v$. Therefore,

$$\begin{aligned}
C_u &= \left(\frac{v}{D}\right)^2 \frac{u}{y}, \\
C_v &= \frac{1}{K_{12}}\left(\frac{u}{D}\right)^2 \frac{v}{y}, \\
H_{uv} &= \frac{2}{K_{12}} \frac{1}{yD}\left(\frac{uv}{D}\right)^2.
\end{aligned}$$

Substituting these expressions into **Equation 68**, I obtain

$$\varepsilon y = \frac{y}{uv/D} \geq 1$$

because $y \geq uv/D$ according to **Equation 73**.

Generating topology $\mathrm{io, io, \emptyset, S}$ (**Figure 10**). According to Property 1 of the CGP (**Box 1**), module $\mu$ can be coarse-grained by first eliminating metabolite 3. In the resulting module $\mu'$, mutation A perturbs only the rate constant $u'$ of the effective reaction $a' \equiv 1 \leftrightarrow 4$ (by Properties 2 and 4 of the CGP). Then, according to **Equation 50** and Theorem 1, $|\delta^A u'| \ll 1$. The double-marked module $(\mu', a', b)$ is minimal in the topological class $\mathcal{M}^{\mathrm{io,io,I}}$ which implies that $\varepsilon y \geq 1$, as shown above.

Generating topology $\mathrm{io, i, I, S_1}$ (**Figure 11**). Notice that metabolite 3 together with reactions $a$, $b$, and $1 \leftrightarrow 4$ form a double-marked module $(\mu', a, b)$ whose I/O metabolites are 1 and 4 and which is minimal in the topological calss $\mathcal{M}^{\mathrm{io,io,I}}$. Therefore, if the effective reaction rate of module $\mu'$ is $y'$, $\varepsilon y' \geq 1$, as shown above. According to **Equation 50**, **Equation 51** and Theorem 1, $|\delta^A y'| \ll 1$, $|\delta^B y'| \ll 1$. Since module $\mu'$ is contained in $\mu$, by Theorem 1, $\varepsilon y \geq 1$.

Generating topology $\mathrm{io, i, I, S_2}$ (**Figure 11**). Module $\mu$ can be coarse-grained by first eliminating metabolite 4, which results in a double-marked module $(\mu', a, b')$ that is minimal in the topological class $\mathcal{M}^{\mathrm{io,io,I}}$. The rest of the proof for this topology is analogous to that for the topology $\mathrm{io, io, \emptyset, S}$.

Generating topology $\mathrm{io, i, \emptyset, S_1}$ (**Figure 12**). Module $\mu$ can be coarse-grained by first eliminating metabolites 4 and 5, which results in a double-marked module $(\mu', a, b')$ that is minimal in the

topological class $\mathcal{M}^{\mathrm{io,io,I}}$. The rest of the proof for this topology is analogous to that for the topology $\mathrm{io, io}, \emptyset, \mathrm{S}$.

Generating topology $\mathrm{io, i}, \emptyset, \mathrm{S}_2$ (**Figure 12**). Notice that metabolites 3 and 4 together with reactions $a$, $b$, $1 \leftrightarrow 4$, $1 \leftrightarrow 5$ and $3 \leftrightarrow 4$ form a double-marked module $(\mu', a, b)$ whose I/O metabolites are 1 and 5 and which is minimal in the topological calss $\mathcal{M}^{\mathrm{io,io},\emptyset}$ with the strictly serial generating topology $\mathrm{io, io}, \emptyset, \mathrm{S}$. The rest of the proof for this topology is analogous to that for the topology $\mathrm{io, i, I}, \mathrm{S}_1$.

Generating topology $\mathrm{io, i}, \emptyset, \mathrm{S}_3$ (**Figure 12**). Notice that metabolites 3 and 5 together with reactions $a$, $b$, $1 \leftrightarrow 4$, $3 \leftrightarrow 4$, and $3 \leftrightarrow 5$ form a double-marked module $(\mu', a, b)$ whose I/O metabolites are 1 and 4 and which is minimal in the topological calss $\mathcal{M}^{\mathrm{io,io},\emptyset}$ with the strictly serial generating topology $\mathrm{io, io}, \emptyset, \mathrm{S}$. The rest of the proof for this topology is analogous to that for the topology $\mathrm{io, i, I}, \mathrm{S}_1$.

Generating topology $\mathrm{i, i, I}, \mathrm{S}_1$ (**Figure 13**). Notice that metabolite 3 together with reactions $a$, $b$, and $4 \leftrightarrow 5$ form a double-marked module $(\mu', a, b)$ whose I/O metabolites are 4 and 5 and which is minimal in the topological calss $\mathcal{M}^{\mathrm{io,io,I}}$. The rest of the proof for this topology is analogous to that for the topology $\mathrm{io, i, I}, \mathrm{S}_1$.

Generating topology $\mathrm{i, i, I}, \mathrm{S}_2$ (**Figure 13**). Notice that metabolites 3 and 5 together with reactions $a$, $b$, $1 \leftrightarrow 3$, $1 \leftrightarrow 4$, and $1 \leftrightarrow 5$ form a double-marked module $(\mu', a, b)$ whose I/O metabolites are 1 and 4 and which is minimal in the topological calss $\mathcal{M}^{\mathrm{io,i,I}}$ with the strictly serial generating topology $\mathrm{io, i, I}, \mathrm{S}_2$. The rest of the proof for this topology is analogous to that for the topology $\mathrm{io, i, I}, \mathrm{S}_1$.

Generating topology $\mathrm{i, i}, \emptyset, \mathrm{S}_1$ (**Figure 14**). Module $\mu$ can be coarse-grained by first eliminating metabolites 5 and 6, which results in a double-marked module $(\mu', a, b')$ that is minimal in the topological class $\mathcal{M}^{\mathrm{io,i,I}}$ with the strictly serial generating topology $\mathrm{io, i, I}, \mathrm{S}_1$. The rest of the proof for this topology is analogous to that for the topology $\mathrm{io, io}, \emptyset, \mathrm{S}$.

Generating topology $\mathrm{i, i}, \emptyset, \mathrm{S}_2$ (**Figure 14**). Module $\mu$ can be coarse-grained by first eliminating metabolite 6, which results in a double-marked module $(\mu', a, b')$ that is minimal in the topological class $\mathcal{M}^{\mathrm{i,i,I}}$ with the strictly serial generating topology $\mathrm{i, i, I}, \mathrm{S}_1$. The rest of the proof for this topology is analogous to that for the topology $\mathrm{io, io}, \emptyset, \mathrm{S}$.

## Proof of Theorem 2

According to Proposition 3, the coarse-graining procedure $\mathrm{CG}^{\mu \backslash \{a,b\}}$ maps the double-marked module $(\mu, a, b)$ onto a double-marked module $(\mu', a, b)$ that is minimal in the same topological class as $(\mu, a, b)$, and the rates $u$, $v$ of reactions $a$, $b$ in $\mu'$ are given by linear relations in **Equation 32** and **Equation 33**. Clearly, $|\delta^A u| \ll 1$ and $|\delta^B v| \ll 1$. Furthermore, none of the other reaction rates $w_{ij}$ in $\mu'$ depend on $\xi$ or $\eta$, so that $\delta^A w_{ij} = 0$ and $\delta^B w_{ij} = 0$ for all $w_{ij}$ other than $u$ and $v$, and $\varepsilon w_{ij} = 0$ for all $w_{ij}$ including $u$ and $v$. It then follows from Proposition 3 that $\varepsilon y_\mu = \varepsilon y_{\mu'}$.

Now, according to Corollary 4, if reactions $a$ and $b$ are strictly parallel in $(\mu, a, b)$, they are also strictly parallel in $(\mu', a, b)$. Therefore, by Proposition 7, $\varepsilon y_{\mu'} \le 0$. Analogously, if reactions $a$ and $b$ are strictly serial in $(\mu, a, b)$, they are also strictly serial in $(\mu', a, b)$. Therefore, by Proposition 8, $\varepsilon y_{\mu'} \ge 1$.

## Sensitivity of Theorem 1 and Theorem 2 with respect to the magnitude of mutational effects

According to Proposition 2, function $f_1$ for any module belongs to one of three parametric families, which correspond to the three minimal fully connected modules shown in **Figure 7**. As mentioned in the Results, for modules in the class $\mathcal{M}^{\mathrm{b}}$, function $f_1$ is linear, so that the claims of Theorem 1 continue to hold for mutations with finite effects. To evaluate the sensitivity of Theorem 1 with respect to the effect sizes of mutations for the topological classes $\mathcal{M}^{\mathrm{io}}$ and $\mathcal{M}^{\mathrm{i}}$, I generated 1000 minimal single-marked modules $\nu$ from each of these topological classes with random parameters. Evaluating only minimal modules is sufficient because for any module from a given topological class there exists a minimal module from the same class, such that both of them map the lower level phenotype $y_\mu$ onto the higher-level phenotype $y_\nu$ via the same function $f_1$ (see Proposition 2).

To this end, I drew each $x_{ij}$ ($i<j$) from a mixture of a point measure at 0 (with weight 0.25) and an exponential distribution with mean 1 (with weight 0.75). The point measure at 0 ensures that minimal modules that are not fully connected are represented in the sample. I drew each $K_{ij}$ ($i<j$) as a ratio of

two random numbers from an exponential distribution with mean 1. As a result, the distribution of non-zero $x_{ij}$ values had the interdecile range of $(5.7 \times 10^{-2}, 3.91)$ with median 0.65.

I denote the effective rate constant of the reaction that represents the lower-level module μ by $\xi \equiv y_\mu$. In modules from the topological class $\mathcal{M}^{io}$, it is reaction $1 \leftrightarrow 3$ and in modules from the topological class $\mathcal{M}^i$, it is the reaction $3 \leftrightarrow 4$. I perturbed $\xi$ by two mutations $A$ and $B$ with relative effects $\delta^A\xi$ and $\delta^B\xi$ and epistasis $\varepsilon\xi$. I chose nine different pairs of mutational effects $(\delta^A\xi, \delta^B\xi)$: $(-0.01, -0.01)$, $(-0.1, -0.1)$, $(-0.5, -0.5)$, $(0.01, 0.01)$, $(0.1, 0.1)$, $(0.5, 0.5)$, $(-0.01, 0.01)$, $(-0.1, 0.1)$, $(-0.5, 0.5)$, and 16 different values of $\varepsilon\xi$ ranging from $-1$ to 2 with an increment of 0.2. Since the rate constant $\xi^{AB}$ of the double mutant cannot be negative, I skipped those combinations of perturbations and epistasis values for which $\delta^A\xi + \delta^B\xi + 2(\varepsilon\xi)(\delta^A\xi)(\delta^B\xi) < -1$. I then computed the resulting values $\delta^A y_\nu$, $\delta^B y_\nu$ and $\varepsilon y_\nu$ at the level of the effective rate constant $y_\nu$ of the higher-level module ν.

Using these data, I inferred the function $\phi$ that maps lower-level epistasis $\varepsilon\xi$ onto higher-level epistasis $\varepsilon y_\nu$, as follows. For any minimal single-marked module from the topological classes $\mathcal{M}^{io}$ or $\mathcal{M}^i$, the effective rate constant $y_\nu$ can be written as

$$y_\nu = x_{12} + \frac{\tilde{A}\xi + \tilde{B}}{D},$$

where $D = \tilde{C}\xi + \tilde{D}$ and $\tilde{A} = x_{32}$, $\tilde{B} = 0$, $\tilde{C} = 1/K_{13}$, $\tilde{D} = x_{32}$ for modules from the topological class $\mathcal{M}^{io}$ (see **Equation 30**), and $\tilde{A} = (x_{13} + x_{14})(x_{42} + x_{32}/K_{34})$, $\tilde{B} = (x_{31} + x_{32})x_{14}x_{42} + (x_{41} + x_{42})x_{13}w_{32}$, $\tilde{C} = (x_{31} + x_{32})/K_{34} + (x_{41} + x_{42})$, $\tilde{D} = (x_{31} + x_{32})(x_{41} + x_{42})$ for modules from the topological class $\mathcal{M}^i$ (see **Equation 31**). Therefore, for any perturbation $\delta\xi$, we have

$$\delta y_\nu = \frac{\tilde{A}\tilde{D} - \tilde{B}\tilde{C}}{D^2} \frac{\xi}{y_\nu} \frac{\delta\xi}{1 + (\tilde{C}\xi/D)\delta\xi}.$$

Since $\delta^{AB}\xi$ is a linear function of $\varepsilon\xi$, $\delta^{AB}y_\nu$ is a hyperbolic function of $\varepsilon\xi$. Therefore, $\varepsilon y_\nu$ is also a hyperbolic function of $\varepsilon\xi$,

$$\varepsilon y_\nu = \phi(\varepsilon\xi) = a - \frac{b}{\varepsilon\xi + c}, \tag{74}$$

where constants $a$, $b$ and $c$ depend on the parameters of module ν and on the mutational effect sizes $\delta^A\xi$ and $\delta^B\xi$. I numerically calculated these parameters for each sampled module and each pair of mutational effects.

The main results of Theorem 1 are that, when the effects of mutations are infinitesimal, the map $\phi$ has a fixed point $\bar{\varepsilon}$, this fixed point is located between 0 and 1, and it is unstable. I use equation **Equation 74** to test whether these statements also hold when the effects of mutations are finite. Specifically, it is easy to see that the map $\phi$ has a fixed point $\bar{\varepsilon}$ if the discriminant $d = (a - c)^2 - 4(b - ac)$ is positive. In this case, I designate $\bar{\varepsilon}$ as the one of two roots $1/2(a - c \pm \sqrt{d})$ that is closer to zero. I then check whether this fixed point is located between 0 and 1. I check whether it is unstable by comparing the derivative of $\phi$ at $\bar{\varepsilon}$ with 1.

According to Proposition 6, function $f_2$ for any module where the reactions affected by mutations are strictly parallel belongs to one of 17 parameteric families, which correspond to the strictly parallel generating topologies listed in **Table 2**. And similarly, function $f_2$ for any module where the reactions affected by mutations are strictly serial belongs to one of 11 parameteric families, which correspond to the strictly serial generating topologies listed in **Table 3**. Therefore, to evaluate the sensitivity of Theorem 2 with respect to the effect sizes of mutations I generated $10^4$ double-marked modules $(\mu, a, b)$ with each of the strictly serial and strictly parallel topologies with random parameters. I drew $x_{ij}$ and $K_{ij}$ as described above. I chose the same nine pairs of mutational effects $(\delta^A\xi, \delta^B\eta)$ as above, where $\xi$ and $\eta$ are the rate constants of reactions affected by mutations $A$ and $B$: $(-0.01, -0.01)$, $(-0.1, -0.1)$, $(-0.5, -0.5)$, $(0.01, 0.01)$, $(0.1, 0.1)$, $(0.5, 0.5)$, $(-0.01, 0.01)$, $(-0.1, 0.1)$, $(-0.5, 0.5)$.

I found that, for some modules, individual mutational perturbations $\delta^A y_\mu$ and/or $\delta^B y_\mu$ at the level of the whole module were too small, which resulted in numerical instabilities. To avoid them, I

calculated epistasis $\varepsilon y_\mu$ only for cases where the effects of both mutations $\delta^A y_\mu$ and $\delta^B y_\mu$ exceeded the precision threshold of $10^{-5}$. As a result, I evaluated epistasis in less than $10^4$ modules per generating topology and pair of mutational effects, but this number never fell below 1000. When comparing the values of epistasis with 0 and 1, I used the same precision threshold of $10^{-5}$ to avoid numerical problems. In addition, I found that for mutations affecting strictly serial reactions there is a substantial fraction of modules where $\varepsilon y_\mu$ falls between 0.99 and 1 (see *Figure 4—figure supplement 3*). This is not a numerical artifact, but probably reflects real clustering of epistasis coefficients around 1, which is expected for the linear pathway irrespective of its parameters (see above).

The Matlab code for this analysis is available at https://github.com/skryazhi/epistasis_theory.

## Kinetic model of glycolysis

I downloaded the kinetic metabolic model of *E. coli* glycolysis by *Chassagnole et al., 2002* from the BioModels database (*Malik-Sheriff et al., 2019*) on September 15, 2015 (model ID BIOMD0000000051). I used the Matlab SimBiology toolbox to interpret the model. To validate the model, I simulated it for 40 s and reproduced Figures 4 and 5 from *Chassagnole et al., 2002*. The Matlab code is available at https://github.com/skryazhi/epistasis_theory.

### Modifications to the original model

I simplified and modified the model by (a) fixing the concentrations of ATP, ADP, AMP, NADPH, NADP, NADH, NAD at their steady-state values given in Table V of *Chassagnole et al., 2002* and (b) removing dilution by growth. I then created four models of sub-modules of glycolysis by retaining the subsets of metabolites and enzymes shown in *Figure 5—figure supplement 1* and *Table 4* and removing other metabolites and enzymes. Each sub-module has one input and one output metabolite. Note that, since some reactions are irreversible, it is important to distinguish the input metabolite from the output metabolite. The concentrations of the input and the output metabolites in each model are held constant at their steady-state values given in *Table 4*. I defined the flux through the sub-module as the flux toward the output metabolite contributed by the sub-module (*Table 4*). This flux is the equivalent of the quantitative phenotype $y_\mu$ of a module in the analytical model. In addition, I made the following modifications specific to individual sub-modules.

1. In the FULL model, the stoichiometry of the PTS reaction was changed to

$$[Extglu] + [pep] \leftrightarrow [g6p] + [pyr]$$

and the value of the constant $K_{\text{PTS,a1}}$ was set to 0.02 mM, based on the values found in the literature (*Stock et al., 1982*; *Natarajan and Srienc, 1999*).

2. In all models other than FULL, the extracellular compartment was deleted.
3. In all models, the concentrations of the I/O metabolites were set to values shown in *Table 4*, which are the steady-state concentrations achieved in the FULL model with the concentration of extracellular glucose being 2 μM and pyruvate concentration being 10 μM.

**Table 4.** Definition of modules in the glycolysis network shown in *Figure 5—figure supplement 1*.
Enzyme abbreviations are listed in *Table 6*. Metabolite abbreviations are listed in *Table 5*.

| Model | Internal metabolites | Concentrations of I/O metabolites | Reactions | Output flux |
|---|---|---|---|---|
| UGPP | 6 pg, dhap, e4p, f6p, fdp, rib5p, ribu5p, sed7p, xyl5p | [g6p]=3.82 mM, [gap] =0.44 mM | ALDO, G6PDH, PFK, PGDH, PGI, Ru5P, R5PI, TA, TIS, TKa, TKb | $J_{\text{ALDO}} + J_{\text{TIS}} + J_{\text{TKb}} + J_{\text{TKa}} - J_{\text{TA}}$ |
| LG | 2 pg, 3 pg, pgp | [gap]=0.44 mM, [pep] =0.08 mM | ENO, GAPDH, PGK, PGM | $J_{\text{ENO}}$ |
| GPP | all in UGPP and in LG, gap | [g6p]=3.82 mM, [pep] =0.08 mM | all in UGPP and in LG | $J_{\text{ENO}}$ |
| FULL | all in GPP, g6p, pep | [Ext glu]=2 μM, [pyr]=10 μM | all in GPP, PTS, PK, PEPCxyl | $J_{\text{PK}} + J_{\text{PTS}}$ |

**Table 5.** Names of metabolites used in the kinetic model of glycolysis.

| **2 pg** | **2-Phosphoglycerate** |
| --- | --- |
| 3 pg | 3-Phosphoglycerate |
| 6 pg | 6-Phosphogluconate |
| dhap | Dihydroxyacetonephosphate |
| e4p | Erythrose-4-phosphate |
| f6p | Fructose-6-phosphate |
| fdp | Fructose-1,6-bisphosphate |
| g6p | Glucose-6-phosphate |
| gap | Glyceraldehyde-3-phosphate |
| glu | Glucose |
| pep | Phosphoenolpyruvate |
| pgp | 1,3-Diphosphoglycerate |
| pyr | Pyruvate |
| rib5p | Ribose-5-phosphate |
| ribu5p | Ribulose-5-phosphate |
| sed7p | Sedoheptulose-7-phosphate |
| xyl5p | Xylulose-5-phosphate |

## Calculation of flux control coefficients and epistasis coefficients

I calculate the first- and second-order flux control coefficients (FCC) $C_i$ and $H_{ij}$ for flux $J$ with respect to reactions $i$ and $j$ as follows (see *Equation 45* and *Equation 46*). I perturb the $r_{\max,i}$ of reaction $i$ by factor between 0.75 and 1.25 (10 values in a uniformly-spaced grid), such that $\delta r_{\max,i} \in [-0.25, 0.25]$. Then, I obtain the steady-state flux $J'$ in each perturbed model and calculate the flux perturbations $\delta J = J'/J^0 - 1$, where $J^0$ is the corresponding flux in the unperturbed model. Then, to obtain $C_i$ and $H_{ii}$, I fit the linear model

**Table 6.** Names of enzymes used in the kinetic model of glycolysis.

| **ALDO** | **Aldolase** |
| --- | --- |
| ENO | Enolase |
| G6PDH | Glucose-6-phosphate dehydrogenase |
| GAPDH | Glyceraldehyde-3-phosphate dehydrogenase |
| PFK | Phosphofructokinase |
| PGDH | 6-Phosphogluconate dehydrogenase |
| PGI | Glucose-6-phosphateisomerase |
| PGK | Phosphoglycerate kinase |
| PGM | Phosphoglycerate mutase |
| PEPCxyl | PEP carboxylase |
| PK | Pyruvate kinase |
| PTS | Phosphotransferase system |
| R5PI | Ribose-phosphateisomerase |
| Ru5P | Ribulose-phosphate epimerase |
| TA | Transaldolase |
| TIS | Triosephosphate isomerase |
| TKa | Transketolase, reaction a |
| TKb | Transketolase, reaction b |

$$\delta J \sim C_i \left( \delta r_{\max,i} \right) + \frac{H_{ii}}{2} \left( \delta r_{\max,i} \right)^2$$

by least squares. If the estimated value of $C_i$ was below $10^{-4}$ for a given sub-module, I set $C_i$ to zero and exclude this reaction from further consideration in that sub-module because it does not affect flux to the degree that is accurately measurable. If the estimated value of $H_{ii}$ is below $10^{-4}$, I set $H_{ii}$ to zero.

To calculate the non-diagonal second-order control coefficients $H_{ij}$, I create a $4 \times 4$ grid of perturbations of $\delta r_{\max,i}$ and $\delta r_{\max,j}$ and calculate the resulting flux perturbations $\delta J$ (16 perturbations total). Since $C_i$, $C_j$, $H_{ii}$ and $H_{jj}$ are known, I obtain $H_{ij}$, by regressing

$$\delta J - \left( C_i \left( \delta r_{\max,i} \right) + \frac{H_{ii}}{2} \left( \delta r_{\max,i} \right)^2 \right) - \left( C_j \left( \delta r_{\max,j} \right) + \frac{H_{jj}}{2} \left( \delta r_{\max,j} \right)^2 \right)$$

against

$$\left( \delta r_{\max,i} \right) \left( \delta r_{\max,j} \right).$$

If the estimated value of $H_{ij}$ is below $10^{-4}$, I set $H_{ij}$ to zero. I estimate the epistasis coefficient $\varepsilon J$ between mutations affecting reactions $i$ and $j$ as

$$\varepsilon J = \frac{H_{ij}}{2 C_i C_j}.$$

## Establishing the topological relationships between pairs of reactions

To establish the topological relationship (strictly serial, strictly parallel, or serial-parallel) between two reactions, I consider the smallest module (LG, UGPP, GPPP, or FULL) which contains both reactions. I then manually identify whether there exists a simple path connecting the input metabolite with the output metabolite for that module that passes through both reactions. (Note that, since some reactions are irreversible in this model, it is important to distinguish the input metabolite from the output metabolite). If such path does not exist, I classify the topological relationship between the two reactions as strictly parallel. If such path exists, I check if there are two paths connecting the input to the output metabolites such that each path contains only one of the two focal reactions. If such paths do not exist, I classify the topological relationship between the two reactions as strictly serial. Otherwise, I classify it as serial-parallel.

## Acknowledgements

I thank Chris Marx for discussions that stimulated this work, David McCandlish for making me aware of the Kron reduction, Daniel P Rice, the Kryazhimskiy lab, and the reviewers for feedback on the manuscript. This work was supported by the BWF Career Award at Scientific Interface (Grant 1010719.01), the Alfred P Sloan Foundation (Grant FG-2017–9227), the Hellman Foundation and NIH (Grant 1R01GM137112).

## Additional information

### Funding

| Funder | Grant reference number | Author |
|---|---|---|
| Burroughs Wellcome Fund | Career Award at Scientific Interface (1010719.01) | Sergey Kryazhimskiy |
| Alfred P. Sloan Foundation | FG-2017-9227 | Sergey Kryazhimskiy |
| Hellman Foundation | Hellman Fellowship | Sergey Kryazhimskiy |
| National Institutes of Health | 1R01GM137112 | Sergey Kryazhimskiy |

The funders had no role in study design, data collection and interpretation, or the decision to submit the work for publication.

## Author contributions

Sergey Kryazhimskiy, Conceptualization, Software, Formal analysis, Funding acquisition, Investigation, Visualization, Methodology, Writing - original draft, Project administration, Writing - review and editing

## Author ORCIDs

Sergey Kryazhimskiy  https://orcid.org/0000-0001-9128-8705

## Decision letter and Author response

Decision letter https://doi.org/10.7554/eLife.60200.sa1
Author response https://doi.org/10.7554/eLife.60200.sa2

# Additional files

## Supplementary files

• Supplementary file 1. Mathematica notebook 'Case i,i,emptyset,P7.nb' for evaluating epistasis for the generating topology $i, i, \emptyset, P_7$.

• Supplementary file 2. PDF version of the Mathematica notebook 'Case i,i,emptyset,P7.nb'.

• Transparent reporting form

## Data availability

All data generated or analyzed during this study are included in the manuscript and supporting files. Code is available on GitHub.

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

## Appendix 1

### Proof of *Equation 24*

First, the terms in *Equation 24* can be re-arranged as follows

$$
\begin{aligned}
x_{ij}^E &= x_{ij} + \frac{x_{ik_1} x_{k_1 j}}{D_{k_1}^{E \setminus \{k_1\}}} + \frac{x_{ik_2} x_{k_2 j}}{D_{k_2}^{E \setminus \{k_2\}}} + \cdots \\
&\quad + \frac{x_{ik_1} x_{k_1 k_2} x_{k_2 j}}{D_{k_1}^{E \setminus \{k_1\}} D_{k_2}^{E \setminus \{k_1, k_2\}}} + \cdots + \frac{x_{ik_2} x_{k_2 k_1} x_{k_1 j}}{D_{k_2}^{E \setminus \{k_2\}} D_{k_1}^{E \setminus \{k_1, k_2\}}} + \cdots \\
&= x_{ij} + \frac{x_{ik_1}}{D_{k_1}^{E \setminus \{k_1\}}} \underbrace{\left( x_{k_1 j} + \frac{x_{k_1 k_2} x_{k_2 j}}{D_{k_2}^{E \setminus \{k_1, k_2\}}} + \cdots \right)}_{= x_{k_1 j}^{E \setminus \{k_1\}}} \\
&\quad + \frac{x_{ik_2}}{D_{k_2}^{E \setminus \{k_2\}}} \underbrace{\left( x_{k_2 j} + \frac{x_{k_2 k_1} x_{k_1 j}}{D_{k_1}^{E \setminus \{k_1, k_2\}}} + \cdots \right)}_{= x_{k_2 j}^{E \setminus \{k_2\}}} + \cdots \\
&= x_{ij} + \sum_{\ell \in E} \frac{x_{i\ell} x_{\ell j}^{E \setminus \{\ell\}}}{D_{\ell}^{E \setminus \{\ell\}}}.
\end{aligned}
\tag{75}
$$

Next, I will demonstrate the validity of *Equation 75* by induction. It is clear that for $E = \{k\} \subset A_\mu$, *Equation 24* reduces to *Equation 12*. Now suppose that $n_E > 1$ and that there exists a metabolite $k \in E$, such that *Equation 75* holds for the subset $E' = E \setminus \{k\}$, that is,

$$
x_{ij}^{E'} = x_{ij} + \sum_{\ell \in E'} \frac{x_{i\ell} x_{\ell j}^{E' \setminus \{\ell\}}}{D_{\ell}^{E' \setminus \{\ell\}}}.
\tag{76}
$$

I will now show that then *Equation 75* also holds for $E$. To do so, I use the definition of $x_{ij}^E$ (*Equation 19*) and apply *Equation 76* to expand terms $x_{ij}^{E'}$ and $x_{ik}^{E'}$ as follows.

$$
\begin{aligned}
x_{ij}^E &= x_{ij}^{E'} + \frac{x_{ik}^{E'} x_{kj}^{E'}}{D_k^{E'}} \\
&= x_{ij} + \sum_{\ell \in E'} \frac{x_{i\ell} x_{\ell j}^{E' \setminus \{\ell\}}}{D_{\ell}^{E' \setminus \{\ell\}}} + \frac{x_{ik} x_{kj}^{E'}}{D_k^{E'}} + \sum_{\ell \in E'} \frac{x_{i\ell} x_{\ell k}^{E' \setminus \{\ell\}} x_{kj}^{E'}}{D_{\ell}^{E' \setminus \{\ell\}} D_k^{E'}} \\
&= x_{ij} + \sum_{\ell \in E'} \frac{x_{i\ell}}{D_{\ell}^{E' \setminus \{\ell\}}} \left( x_{\ell j}^{E' \setminus \{\ell\}} + \frac{x_{\ell k}^{E' \setminus \{\ell\}} x_{kj}^{E'}}{D_k^{E'}} \right) + \frac{x_{ik} x_{kj}^{E'}}{D_k^{E'}}.
\end{aligned}
\tag{77}
$$

Recalling that $E = E' \setminus \{k\}$, I re-write *Equation 75* as

$$
x_{ij}^E = x_{ij} + \sum_{\ell \in E'} \frac{x_{i\ell} x_{\ell j}^{E \setminus \{\ell\}}}{D_{\ell}^{E \setminus \{\ell\}}} + \frac{x_{ik} x_{kj}^{E'}}{D_k^{E'}}.
\tag{78}
$$

Since the first and third terms of *Equation 77* and *Equation 78* are identical, to complete the proof, it is sufficient to show that for any $\ell \in E'$,

$$
\frac{1}{D_{\ell}^{E' \setminus \{\ell\}}} \left( x_{\ell j}^{E' \setminus \{\ell\}} + \frac{x_{\ell k}^{E' \setminus \{\ell\}} x_{kj}^{E'}}{D_k^{E'}} \right) \equiv \frac{1}{D_{\ell}^{E \setminus \{k, \ell\}}} \left( x_{\ell j}^{E \setminus \{k, \ell\}} + \frac{x_{\ell k}^{E \setminus \{k, \ell\}} x_{kj}^{E \setminus \{k\}}}{D_k^{E \setminus \{k\}}} \right) = \frac{x_{\ell j}^{E \setminus \{\ell\}}}{D_{\ell}^{E \setminus \{\ell\}}}.
\tag{79}
$$

To show that *Equation 79* holds, I first use *Equation 19* and *Equation 21* to express $x_{kj}^{E \setminus \{k\}}$ and $D_k^{E \setminus \{k\}}$ in terms of effective reaction rates after the elimination of the metabolite set $E \setminus \{k, \ell\}$,

$$x_{kj}^{E\setminus\{k\}} = x_{kj}^{E\setminus\{k,\ell\}} + \frac{x_{k\ell}^{E\setminus\{k,\ell\}} x_{\ell j}^{E\setminus\{k,\ell\}}}{D_{\ell}^{E\setminus\{k,\ell\}}}$$

$$D_{k}^{E\setminus\{k\}} = D_{k}^{E\setminus\{k,\ell\}} - \frac{x_{k\ell}^{E\setminus\{k,\ell\}} x_{\ell k}^{E\setminus\{k,\ell\}}}{D_{\ell}^{E\setminus\{k,\ell\}}},$$

(80)

which imply that

$$\frac{1}{D_{\ell}^{E\setminus\{k,\ell\}}} \left( x_{\ell j}^{E\setminus\{k,\ell\}} + \frac{x_{\ell k}^{E\setminus\{k,\ell\}} x_{kj}^{E\setminus\{k\}}}{D_{k}^{E\setminus\{k\}}} \right) = \frac{D_{k}^{E\setminus\{k,\ell\}}}{D_{\ell}^{E\setminus\{k,\ell\}} D_{k}^{E\setminus\{k\}}} \left( x_{\ell j}^{E\setminus\{k,\ell\}} + \frac{x_{\ell k}^{E\setminus\{k,\ell\}} x_{kj}^{E\setminus\{k,\ell\}}}{D_{k}^{E\setminus\{k,\ell\}}} \right)$$

$$= \frac{D_{k}^{E\setminus\{k,\ell\}} x_{\ell j}^{E\setminus\{\ell\}}}{D_{\ell}^{E\setminus\{k,\ell\}} D_{k}^{E\setminus\{k\}}}.$$

Finally, it follows from *Equation 80* that $D_{\ell}^{E\setminus\{k,\ell\}} D_{k}^{E\setminus\{k\}} = D_{k}^{E\setminus\{k,\ell\}} D_{\ell}^{E\setminus\{\ell\}}$, which completes the proof of *Equation 79*. Thus, *Equation 75* and equivalently *Equation 24* hold for any metabolite subset $E \subseteq A_{\mu}$.

## Appendix 2

### Existence of a simple path that contains a given reaction
### Lemma 1

Let μ be a module with the reaction set $R_\mu$. Then, for any reaction $a \in R_\mu$, there exists a simple path $p_{12}(a)$ within μ that connects the I/O metabolites and contains reaction $a$, that is, $\mathcal{P}^\mu_{12}(a) \neq \emptyset$.

### Proof

Reaction $a$ is either a bypass, i/o, or internal reaction for module μ. If $a$ is a bypass reaction, then the statement is trivially true. If $a$ is an i/o reaction, then, without loss of generality, let $a = 1 \leftrightarrow j$. Since μ is a module, there exists a simple path $j \leftrightarrow j_1 \leftrightarrow \cdots \leftrightarrow 2$ that connects the internal metabolite $j$ to the I/O metabolite 2. Therefore, the path $1 \leftrightarrow j \leftrightarrow j_1 \leftrightarrow \cdots \leftrightarrow 2$ connects the I/O metabolites and contains reaction $a$.

Suppose that $a = i \leftrightarrow j$ is an internal reaction. To prove the statement, it is sufficient to show that there exists a pair of non-intersecting paths $p'_{1i}$ and $p'_{2i}$, such that one of them contains $a$ and the other does not. Since μ is a module, there exists a pair of non-intersecting paths $p_{1i}$ and $p_{2i}$ and a pair of non-intersecting paths $p_{1j}$ and $p_{2j}$ within module μ (I omitted super-index μ to simplify notations). There are two mutually exclusive possibilities. (i) Metabolite $j$ is contained in either of the paths $p_{1i}$ or $p_{2i}$ and/or metabolite $i$ is contained in either of the paths $p_{1j}$ or $p_{2j}$. (ii) Metabolite $j$ is not contained in either of the paths $p_{1i}$ or $p_{2i}$ and metabolite $i$ is not contained in either of the paths $p_{1j}$ or $p_{2j}$, that is, $j \notin p_{ui}$ and $i \notin p_{uj}$, $u = 1, 2$. It is trivial to construct the neccessary paths $p'_{1i}$ and $p'_{2i}$ in case (i).

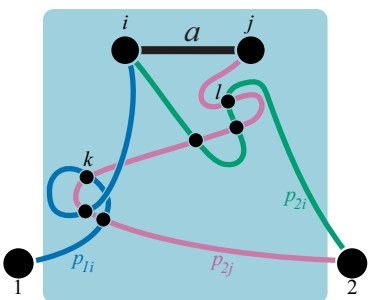

**Appendix 2—figure 1.** Illustration for the proof of Lemma 1.

Consider case (ii). If paths $p_{2j}$ and $p_{1i}$ do not intersect, then let $p'_{1i} = p_{1i}$ and

$$p'_{2i} = \underbrace{i \leftrightarrow j}_{=a} \overbrace{\leftrightarrow \cdots \leftrightarrow 2}^{=p_{2j}},$$

and the statement is true. Suppose paths $p_{2j}$ and $p_{1i}$ intersect. Then, among all metabolites that belong to both $p_{1i}$ and $p_{2j}$, let metabolite $k$ be the one closest to $j$ along the path $p_{2j}$ (see Figure). Then the segment $p_{kj}$ of path $p_{2j}$ and the path $p_{1i}$ do not intersect. Let

$$p''_{1i} = \overbrace{1 \leftrightarrow \cdots \leftrightarrow k}^{\text{along } p_{1i}} \underbrace{\leftrightarrow \cdots \leftrightarrow}_{\text{along } p_{2j}} \overbrace{j \leftrightarrow i}^{=a},$$

If $p''_{1i}$ and $p_{2i}$ do not intersect, then the lemma is true. If $p''_{1i}$ and $p_{2i}$ do intersect, this intersection can only occur within the segment $p_{kj}$ of path $p''_{1i}$, excluding metabolites $k$ and $j$ (see Figure). This is because the remaining segment $p_{1k}$ of path $p''_{1i}$ is also a segment of $p_{1i}$, which, by assumption, does not intersect $p_{2i}$. Suppose that among all metabolites that belong to both the segment $p_{kj}$ of path $p''_{1i}$ and the path $p_{2i}$ metabolite $\ell$ is the one closest to $j$ along the path $p''_{1i}$. Then let

$$p'_{1i} = p_{1i},$$

$$p'_{2i} = \overbrace{2 \leftrightarrow \cdots \leftrightarrow}^{\text{along } p_{2i}} \underbrace{\ell \leftrightarrow \cdots \leftrightarrow}_{\text{along } p_{2j}} \overbrace{j \leftrightarrow i}^{=a}.$$

The path $p'_{2i}$ does not intersect the path $p_{1i}$ because its first segment $p_{2\ell}$ belongs to path $p_{2i}$ and its second segment $p_{\ell j}$ belongs to the segment $p_{kj}$ of path $p''_{1i}$ (and, as mentioned above, segment $p_{kj}$ does not intersect $p_{1i}$). Thus, the statement holds for case (ii) as well.

## Appendix 3

### An algorithm for discovering all strictly serial and strictly parallel generating topologies

Suppose that $(R, a, b) \in \mathcal{G}^{\text{ser}}_{\mathcal{M}}$, that is, $(R, a, b)$ is a strictly serial generating topology in the topological class $\mathcal{M}$. Since $R \subseteq R_{\mathcal{M}}$ where $R_{\mathcal{M}}$ is the complete reaction set $R_{\mathcal{M}}$ for class $\mathcal{M}$, $R$ can be discovered by sequentially removing reactions from $R_{\mathcal{M}}$. The same logic holds for strictly parallel generating topologies. The following algorithm implements this idea.

1. Define function generate_topology_list. This function takes a topology $(R, a, b) \in \mathcal{R}^{\text{sp}}_{\mathcal{M}}$ as input and returns a new list of topologies $L$ as output, which is produced as follows. Initialize $L = \emptyset$. For every reaction $c_i \in R \setminus \{a, b\}$, construct the reaction subset $R_i = R \setminus \{c_i\}$ and use Definition 1 to test whether $R_i$ corresponds to a valid module. If $R_i$ corresponds to a module, add $(R_i, a, b)$ to list $L$; otherwiese, discard. It can be proven that, as long as $(R, a, b) \in \mathcal{R}^{\text{sp}}_{\mathcal{M}}$, there exists at least one $c_i \in R$, such that $R_i$ corresponds to a module, that is, $L \neq \emptyset$. Return list $L$.

2. Initialization.
   a. Pick a topological class $\mathcal{M}$.
   b. Test whether $(R_{\mathcal{M}}, a, b) \in \mathcal{R}^{\text{ser}}_{\mathcal{M}}$. If so, $\mathcal{G}^{\text{ser}}_{\mathcal{M}} = \{(R_{\mathcal{M}}, a, b)\}$ and $\mathcal{G}^{\text{par}}_{\mathcal{M}} = \emptyset$. Return $\mathcal{G}^{\text{ser}}_{\mathcal{M}}$, $\mathcal{G}^{\text{par}}_{\mathcal{M}}$.
   c. Test whether $(R_{\mathcal{M}}, a, b) \in \mathcal{R}^{\text{par}}_{\mathcal{M}}$. If so, $\mathcal{G}^{\text{par}}_{\mathcal{M}} = \{(R_{\mathcal{M}}, a, b)\}$ and $\mathcal{G}^{\text{ser}}_{\mathcal{M}} = \emptyset$. Return $\mathcal{G}^{\text{ser}}_{\mathcal{M}}$, $\mathcal{G}^{\text{par}}_{\mathcal{M}}$.
   d. Set $\mathcal{G}^{\text{ser}}_{\mathcal{M}} = \emptyset$, $\mathcal{G}^{\text{par}}_{\mathcal{M}} = \emptyset$. Use function generate_topology_list with $(R_{\mathcal{M}}, a, b)$ as input and obtain the list of reaction sets $L$. Proceed to Step 3 with list $L$.

3. Take list $L = ((R_1, a, b), (R_2, a, b), \ldots, (R_k, a, b))$ as input. Set $L' = \emptyset$. Proceed to Step a with $i = 1$.
   a. Test whether $(R_i, a, b)$ belongs to $\mathcal{R}^{\text{par}}_{\mathcal{M}}$, $\mathcal{R}^{\text{ser}}_{\mathcal{M}}$ or $\mathcal{R}^{\text{sp}}_{\mathcal{M}}$. Choose one of the alternatives b, c, or d.
   b. $(R_i, a, b) \in \mathcal{R}^{\text{par}}_{\mathcal{M}}$. If $(R_i, a, b)$ is not in the set $\mathcal{G}^{\text{par}}_{\mathcal{M}}$ and if $(R_i \cup \{c\}, a, b) \in \mathcal{R}^{\text{sp}}_{\mathcal{M}}$ for all $c \in R_{\mathcal{M}} \setminus R_i$, then add $(R_i, a, b)$ to $\mathcal{G}^{\text{par}}_{\mathcal{M}}$. Proceed to Step e.
   c. $(R_i, a, b) \in \mathcal{R}^{\text{ser}}_{\mathcal{M}}$. If $(R_i, a, b)$ is not in the set $\mathcal{G}^{\text{ser}}_{\mathcal{M}}$ and if $(R_i \cup \{c\}, a, b) \in \mathcal{R}^{\text{sp}}_{\mathcal{M}}$ for all $c \in R_{\mathcal{M}} \setminus R_i$, then add $(R_i, a, b)$ to $\mathcal{G}^{\text{ser}}_{\mathcal{M}}$. Proceed to Step e.
   d. $(R_i, a, b) \in \mathcal{R}^{\text{sp}}_{\mathcal{M}}$. Use function generate_topology_list with $(R_i, a, b)$ as input and obtain the list of reaction sets $L_i$. Replace $L'$ with $L' \cup L_i$. Proceed to Step e.
   e. If $i = k$, proceed to Step f. Otherwise, proceed to Step a with $i + 1$.
   f. If $L' \neq \emptyset$, then proceed to Step 3 with $L'$ as input. Otherwise, return $\mathcal{G}^{\text{ser}}_{\mathcal{M}}$, $\mathcal{G}^{\text{par}}_{\mathcal{M}}$.

## Appendix 4

### Derivation of *Equation 72*

In this case, *Equation (42)* simplify to

$$
\begin{aligned}
W_{12}(v) &= w_{12} + \frac{w_{16}\,w_{52}\,v/K_{56}}{D_{56}(v)}, \\
W_{13}(v) &= w_{13} + \frac{w_{16}\,w_{63}\,D_5(v)}{D_{56}(v)}, \\
W_{14}(v) &= \frac{w_{16}\,w_{54}\,v/K_{56}}{D_{56}(v)}, \\
W_{23}(v) &= \frac{w_{25}\,w_{63}\,v}{D_{56}(v)}, \\
W_{24}(v) &= w_{24} + \frac{w_{25}\,w_{54}\,D_6(v)}{D_{56}(v)}, \\
W_{34}(u,v)u\ &+ \frac{w_{36}\,w_{54}\,v/K_{56}}{D_{56}(v)},
\end{aligned}
$$

where

$$
\begin{aligned}
D_{56}(v) &= D_5(v)\,D_6(v) - \frac{v^2}{K_{56}}, \\
D_5(v) &= w_{52} + w_{54} + v, D_6(v) \\
D_6(v) &= w_{61} + w_{63} + v/K_{56}.
\end{aligned}
$$

Notice that the only effective rate constant that depends on $u$ is $W_{34}$, and $\frac{\partial W_{34}}{\partial u} = 1$. Thus, it is easy to differentiate $y$, given by *Equation 35*, with respect to $u$, if we isolate the term $W_{34}$ in both the numerator and the denomintor,

$$
y = W_{12} + \frac{A_{56,34}\,W_{34} + B_{56,34}}{D_{56,34}}, \tag{81}
$$

where

$$
\begin{aligned}
D_{56,34} &= E_{56,34}\,W_{34} + F_{56,34}, \\
A_{56,34} &= W_{14}\,W_{42} + \frac{W_{13}\,W_{32}}{K_{34}} + W_{13}\,W_{42} + \frac{W_{14}\,W_{32}}{K_{34}}(W_{13} + W_{14})\left(\frac{W_{32}}{K_{34}} + W_{42}\right), \\
B_{56,34} &= W_{14}\,W_{42}\,(W_{31} + W_{32}) + W_{13}\,W_{32}\,(W_{41} + W_{42}), \\
E_{56,34} &= \frac{W_{31} + W_{32}}{K_{34}} + (W_{41} + W_{42}), \\
F_{56,34} &= (W_{31} + W_{32})(W_{41} + W_{42}).
\end{aligned}
$$

It is also useful to obtain another expression for $y$, which is easier to differentiate with respect to $v$. To do that, we can first eliminate metabolites 3 and 4 to obtain effective reaction rates

$$
\begin{aligned}
V_{12}(u) &= w_{12} + \frac{w_{13}\,w_{42}\,u}{D_{34}(u)}, \\
V_{15}(u) &= \frac{w_{13}\,w_{45}\,u}{D_{34}(u)}, \\
V_{16}(u) &= w_{16} + \frac{w_{13}\,w_{36}\,D_4(u)}{D_{34}(u)}, \\
V_{25}(u) &= w_{25} + \frac{w_{24}\,w_{45}\,D_3(u)}{D_{34}(u)}, \\
V_{26}(u) &= \frac{w_{24}\,w_{36}\,u/K_{34}}{D_{34}(u)}, \\
V_{56}(u,v) &= v + \frac{w_{36}\,w_{54}\,u/K_{34}}{D_{34}(u)},
\end{aligned}
$$

where

$$
\begin{aligned}
D_{34}(u) &= D_3(u)\,D_4(u) - \frac{u^2}{K_{34}}, \\
D_3(u) &= w_{31} + u + w_{36}, \\
D_4(u) &= w_{42} + u/K_{34} + w_{45}.
\end{aligned}
$$

The only effective activity that depends on $v$ is $V_{56}$, and $\frac{\partial V_{56}}{\partial v} = 1$. Thus, isolating the term $V_{56}$ (and recalling that $V_{65} = V_{56}/K_{56}$), we obtain the following expression for $y$, which is easy to differentiate with respect to $v$.

$$
y = V_{12} + \frac{A_{34,56}\,V_{65} + B_{34,56}}{D_{34,56}}, \tag{82}
$$

where

$$
\begin{aligned}
D_{34,56} &= E_{34,56}\,V_{65} + F_{34,56}, \\
A_{34,56} &= (V_{16} + V_{15})\left(\frac{V_{62}}{K_{65}} + V_{52}\right), \\
B_{34,56} &= V_{15}\,V_{52}\,(V_{61} + V_{62}) + V_{16}\,V_{62}\,(V_{51} + V_{52}), \\
E_{34,56} &= \frac{V_{61} + V_{62}}{K_{65}} + (V_{51} + V_{52}), \\
F_{34,56} &= (V_{61} + V_{62})(V_{51} + V_{52}).
\end{aligned}
$$

Using symbolic computation it is possible to show that (see Mathematica notebook [**Supplementary file 1**] and Mathematica notebook pdf [**Supplementary file 2**], p. 2)

$$
D_{56,34}\,D_{56} = D_{34,56}\,D_{34}. \tag{83}
$$

Also notice that, any module with the reaction set $i, i, \emptyset, P_7$ is symmetric with respect to swapping metabolite labels 1 with 2, 3 with 5, and 4 with 6. It is easy to check that **Equations (81), (82)** respect this symmetry.

Differentiating **Equation 81** with respect to $u$, after some algebra I obtain

$$
\frac{\partial y}{\partial u} = \frac{\partial y}{\partial W_{34}} = \frac{1}{K_{31}}\left(\frac{W_{31}\,W_{42} - W_{32}\,W_{41}}{D_{56,34}}\right)^2. \tag{84}
$$

Analogously, differentiating **Equation 82** with respect to $v$, I obtain

$$
\frac{\partial y}{\partial v} = \frac{1}{K_{56}}\frac{\partial y}{\partial V_{65}} = \frac{1}{K_{51}}\left(\frac{V_{52}\,V_{61} - V_{51}\,V_{62}}{D_{34,56}}\right)^2. \tag{85}
$$

Notice that **Equation 85** can also be obtained from **Equation 84** by symmetry with respect to the aforementioned metabolite relabeling.

Next, using symbolic computation (see Mathematica notebook (**Supplementary file 1**) and Mathematica notebook pdf (**Supplementary file 2**), p. 3), it is possible to show that

$$
\frac{(W_{31}\,W_{42} - W_{32}\,W_{41})\,D_{56}}{D_{56,34}\,D_{56}} = \frac{A_u\,v + B_u}{E_u\,v + F_u}, \tag{86}
$$

where all coefficients

$$
\begin{aligned}
A_u &= \frac{w_{31}}{K_{56}}\psi + w_{42}\,\phi, \\
B_u &= \psi\phi, \\
E_u &= u\left[\frac{w_{31}\,w_{52} + w_{31}\,w_{54} + w_{36}\,w_{52}}{K_{34}\,K_{56}} + (w_{42}\,w_{61} + w_{42}\,w_{63} + w_{45}\,w_{61})\right] + D_4(u)\,\phi + \frac{D_3(u)}{K_{56}}\,\psi, \\
F_u &= u\left(\frac{w_{52} + w_{54}}{K_{34}}\,\phi + (w_{61} + w_{63})\,\psi\right) + \phi\,\psi.
\end{aligned}
$$

and

$$
\begin{aligned}
\phi &= w_{31}\,w_{61} + w_{31}\,w_{63} + w_{36}\,w_{61}, \\
\psi &= w_{42}\,w_{52} + w_{42}\,w_{54} + w_{45}\,w_{52}
\end{aligned}
$$

are independent of $u$ and $v$ and are non-negative. Similarly (see Mathematica notebook (*Supplementary file 1*) and Mathematica notebook pdf (*Supplementary file 2*), p. 4),

$$\frac{(V_{52} V_{61} - V_{51} V_{62}) D_{34}}{D_{34,56} D_{34}} = \frac{A_v u + B_v}{E_v u + F_v},\tag{87}$$

where

$$\begin{aligned}
A_v &= w_{61} \psi + \frac{w_{52}}{K_{34}} \phi, \\
B_v &= \psi \phi, \\
E_v &= v \left[ (w_{42} w_{61} + w_{42} w_{63} + w_{45} w_{61}) + \frac{w_{31} w_{52} + w_{31} w_{54} + w_{36} w_{52}}{K_{34} K_{56}} \right] + \frac{D_5(v)}{K_{34}} \phi + D_6(v) \psi, \\
F_v &= v \left( (w_{42} + w_{45}) \phi + \frac{w_{31} + w_{36}}{K_{56}} \psi \right) + \phi \psi.
\end{aligned}$$

We can now obtain the second derivative $\frac{\partial^2 y}{\partial u \partial v}$, taking into account *Equation 86*. Alternatively, we can obtain $\frac{\partial^2 y}{\partial u \partial v}$ by differentiating $\frac{\partial y}{\partial v}$ with respect to $u$, taking into account *Equation 87*. The denominators in both expressions would be identical due to *Equation 83*. Therefore the expression for the second derivative must have the form given by *Equation 72*, that is,

$$\frac{\partial^2 y}{\partial u \partial v} = \frac{2\beta}{K_{31}} \frac{(A_u v + B_u)(A_v u + B_v)}{(E_u v + F_u)^3},$$

where β is independent of $u$ and $v$. Thus, according to *Equation 84*, *Equation 86*,

$$\beta = \frac{A_u F_u - B_u E_u}{A_v u + B_v} = -(w_{36} \psi / K_{56} + w_{45} \phi),$$

which is verified in Mathematica notebook (*Supplementary file 1*) and Mathematica notebook pdf (*Supplementary file 2*), p. 4. Similarly, according to *Equation 85*, *Equation 87*,

$$\frac{\beta}{K_{31}} = \frac{1}{K_{51}} \frac{A_v F_v - B_v E_v}{A_u v + B_u} = -\frac{w_{63} \psi + w_{54}/K_{34} \phi}{K_{51}}$$

which is verified in Mathematica notebook (*Supplementary file 1*) and Mathematica notebook pdf (*Supplementary file 2*), p. 5.

## Appendix 5

### Relationship to the flux balance analysis

Here, I discuss the relationship of the model presented in this paper to the flux balance analysis (FBA), a widely used approach to modeling whole-cell metabolism (*Orth et al., 2010*). FBA and my model are designed to address different questions. My model was designed to explore how the flux depends on the kinetic parameters of a module. FBA was designed to avoid this dependence. Nevertheless, the two models are conceptually and mathematically related and in fact are in some sense equivalent, as discussed below. The most important similarity is that both models rely on flux balance—the assumption that internal metabolites are at steady state—as a key simplification.

The major difference is that my model assumes that all reaction kinetics are first order, which allows for analytical tractability, while FBA is agnostic with respect to reaction kinetics and takes into account only their stoichiometries and possibly some additional 'capacity' constraints on the reaction rates (*Orth et al., 2010*). This difference deserves some additional discussion. In real cells, the concentrations of internal metabolites and the resulting fluxes (which are functions of these concentrations) are determined by the stoichiometries of reactions, the activities of enzymes and the concentrations of external nutrients. A general approach to modeling such systems, adopted by the metabolic control analysis and related theories, is to explicitly specify the dynamic equations for the metabolites, such as *Equation 10*, and solve them to find steady-state concentrations of internal metabolites (see e.g. *Savageau, 1976*). The steady-state fluxes are then determined automatically. In my model, the internal steady state exists and is unique for any module (see Corollary 1). In contrast, there are no dynamic equations within FBA. Instead, the steady-state fluxes are subject only to the mass-balance equations and the capacity conditions, which typically form a severely underdetermined system. Therefore, arriving at a unique flux distribution in FBA requires additional assumptions. The typical approach is to first fix at least some nutrient uptake rates and then maximize an objective function, such as the growth rate or biomass yield (*Feist and Palsson, 2010*). In other words, FBA trades-off the ability to handle reactions with arbitrary kinetics and unknown kinetic parameters against the necessity to impose auxiliary conditions to find the right solution among many plausible ones.

As a consequence of these model design choices, mutations that add or remove reactions can be naturally studied within the FBA framework, but there is no natural way to incorporate mutations that perturb the kinetic parameters of reactions (*He et al., 2010*; *Alzoubi et al., 2019*). In contrast, mutations of small or large effect, including reaction additions and deletions, can be in principle naturally studied within my model.

Due to differences in the assumptions, my model and FBA describe different sets of biological systems. However, there are special cases which can be described by both FBA and my model, and in these special cases the two models are mathematically equivalent.

From the perspective of FBA, modules with two I/O metabolites and first-order kinetics described by my model are a special case. So, let us apply FBA to a metabolic module $\mu = (A_\mu, \vec{x}_\mu)$ with two I/O metabolites 1 and 2. For the purposes of FBA, let the I/O metabolite 1 be the 'external nutrient' and let the I/O metabolite 2 be the 'biomass'. Suppose that metabolites in the set $A_{\text{in}} \subset A_\mu$ are adjacent to the external nutrient and metabolites in the set $A_{\text{out}} \subset A_\mu$ are adjacent to the biomass. Then reactions $1 \leftrightarrow i$ for $i \in A_{\text{in}}$, that is, those that convert the nutrient into intermediate metabolites, are the 'uptake reactions'. And the reactions $i \leftrightarrow 2$ for $i \in A_{\text{out}}$, that is, those that convert intermediate metabolites into biomass, are the 'biomass reactions'. Denote the rate of reaction consuming metabolite $i$ and producing metabolite $j$ by $v_{ij}$ and let $J_{\text{in}} = \sum_{i \in A_{\text{in}}} v_{1i}$ and $J_{\text{out}} = \sum_{i \in A_{\text{out}}} v_{i2}$ be the input and output fluxes, respectively. To study epistasis within the FBA framework, we would need to obtain $J_{\text{out}}$.

The assumption of my model that all reaction kinetics are first-order translates into the FBA formalism as the fact that the elements of the stoichiometry matrix $\vec{S}$ take values $-1$, $+1$ or $0$ and there are no capacity constraints on the rates of reactions. Then, the mass-balance equation $\vec{S}\vec{v} = \vec{0}$ leads to the equality $J_{\text{in}} = J_{\text{out}}$, so that the flux through the module does not depend on module's internal structure. Instead, it must be given as an auxiliary condition.

In my model, the steady-state flux $J$ through the module depends in general on the internal structure of the module (through the effective rate constant $y_\mu$) and is given by $J = y_\mu(S_1 - S_2/K_{12})$ for any

concentrations $S_1$ and $S_2$ of the external nutrient and the biomass. However, in the degenerate case where all uptake reactions are irreversible (i.e., $K_{1i} = \infty$ for all $i \in A_{\text{in}}$), $J = J_{\text{in}} = \sum_{i \in A_{\text{in}}} x_{1i} S_1$, such that it is independent of the internal structure of the module. In this sense, this special case of my model is equivalent to FBA. However, my model and FBA are not equivalent in terms of the distribution of internal fluxes. My model still produces a unique steady sate and the corresponding flux distribution, whereas FBA does not specify a unique flux distribution in this case without additional auxiliary conditions.

## Appendix 6

### Connection between metabolism and growth

Here I describe a simple 'bioreactor' model for connecting a hierarchical metabolic network described in this paper with cellular growth. Suppose that module μ with I/O metabolites 1 and $n$ is at the top level of the metabolic hierarchy and describes the whole cell. The I/O metabolite 1 can be thought of as nutrients, and the I/O metabolite $n$ can be thought of as proteins, so that cellular metabolism converts nutrients into biomass. Denote the concentrations of metabolites and proteins within cells by $S_i$ and $p$. Suppose that cells have a fixed volume $v$ and the number of cells is $N$. Then, the absolute abundance of proteins and metabolites across all $N$ cells is $P = pvN$ and $A_i = S_i vN$, $i = 1, \ldots, n-1$.

The proteins produced by the cell are the enzymes that catalyze all the reactions inside module μ. Let the relative expression level of the enzyme catalyzing reaction $i \leftrightarrow j$, $i<j$ be $\phi_{ij}$ (with $\sum_{i=1}^{n-1} \sum_{j>i} \phi_{ij} = 1$), so that its concentration inside cells is $\phi_{ij} p$. If the specific forward and reverse activities of this enzyme are $a_{ij}$ and $a_{ji}$ (so that $a_{ij}$ obey the Haldane equalities and $a_{ii} = 0$), then the total forward and reverse activities are $x_{ij} = a_{ij} \phi_{ij} p$ and $x_{ji} = a_{ji} \phi_{ij} p$.

Finally, I assume that all reactions converting internal metabolites into biomass are irreversible, and I assume that the cell density in the bioreactor is small enough that the nutrient concentration $S_1$ stays constant. In other words, I model early exponential growth. Then the dynamics of metabolite and protein abundances are governed by equations

$$\dot{A}_i = \sum_{i=1}^{n-1} x_{ji} A_j - D_i A_i, \quad i = 2, \ldots n-1. \tag{88}$$

$$\dot{P} = \sum_{i=2}^{n-1} x_{in} A_i, \tag{89}$$

where $D_i = \sum_{i=1}^{n} x_{ij}$.

I assume that all cells are identical and at steady state, such that protein and metabolite concetrations inside cells are constants and the production of proteins and metabolites manifests itself in the multiplication of cells. Then *Equation 88* and *Equation 89* become

$$\lambda S_i = \sum_{i=1}^{n-1} x_{ji} S_j - D_i S_i, \quad i = 2, \ldots n-1. \tag{90}$$

$$\dot{N} = \lambda N, \tag{91}$$

where $\lambda = \sum_{i=2}^{n-1} x_{in} S_i$ is the steady-state flux into biomass which defines the exponential growth rate of the system.

Finally, assuming that the rates of consumption and production of all metabolites are much greater than their dilution by growth, I set $\lambda S_i \approx 0$. Then *Equation 90* defines the same steady state for module $\mu$ as described in the section Network coarse-graining. Therefore, module $\mu$ can be replaced with the effective activity $x_\mu$ between nutrients and biomass, yielding $\lambda = x_\mu$.

