## [Decision Letter]

**Acceptance summary:**

In our opinion, the work presented here will have a broad influence in the field, by presenting an analytical model linking epistasis to mechanistic processes. This is important as such models are scarce. The paper is conceptually innovative, by studying epistasis in the context of a chemical reaction network, and showing how the underlying biochemical reactions constrain epistasis. This adds to ongoing efforts to more precisely understand, at the mechanistic level, how epistasis arises from microscopic processes, and as such will be of interest to a wide range of researchers in evolutionary cell biology, evolutionary genetics, and biophysics.

**Decision letter after peer review:**

Thank you for submitting your article "Emergence and Propagation of Epistasis in Metabolic Networks" for consideration by *eLife*. Your article has been reviewed by three peer reviewers, and the evaluation has been overseen by a Reviewing Editor and Patricia Wittkopp as the Senior Editor. The following individuals involved in review of your submission have agreed to reveal their identity: Benjamin H Good (Reviewer #1); Arvind Murugan (Reviewer #3).

The reviewers have discussed the reviews with one another and the Reviewing Editor has drafted this decision to help you prepare a revised submission.

Summary:

This manuscript investigates the emergence and propagation of epistasis in metabolic networks. To that end, the author provides a mathematical analysis of the relationship between epistasis for steady-state flux and network toplogy in linear metabolic networks. The approach is a mathematical study of the combined effect of pairwise perturbations of parameters in the set of linear ODEs that describes the system. The author considers these perturbations of microscopic reaction rates as "mutations", and the interaction between the effects is interpreted as epistasis between the mutations.

Reviewers were positive overall about the paper, but noted three important areas where it must be improved prior to being acceptable for publication. I proceed to summarize these areas where revisions are needed.

With the goal of making these criticisms as useful and constructive as possible, I also include potential solutions to the issues raised by the reviewers. Note that these are offered as suggestions.

1) Clarity: The reviewers found that the paper was hard to read and that it would benefit from a clearer presentation. Below I summarize some of the main concerns regarding this issue and provide potential solutions:

Revision #1: The manuscript uses a complicated notation and terminology to derive results that are in the end (mathematically) rather simple. This makes the manuscript hard to read, and it will limit the readership. Just remembering the notation and the exact meaning of the terms is a challenge, and this level of complexity is not necessary to derive these results.

Possible solutions: Attempt to simplify notation. State the two theorems in the main text rather than in the appendix. Simplify the notation in the main text. Consider moving Proposition 6 to the main text.

Revision #2: The Materials and methods were hard to navigate even for specialists who are familiar with the math. A better organization should help. For instance, what the reviewers looked for in the Materials and methods was – (a) how to compute the expansion in Equation 3 for a simple network and in general, (b) details to understand the results on series and parallel pathways. These ideas are not easily found in Materials and methods.

Possible solutions: Re-organize the Materials and methods section in a more modular way, with clear headings and an introduction describing what is going to be accomplished in each subsection, linking each section with the main results in the text (e.g. "Derivation of Equation X"). Organizing by such themes, working out a specific example, and following it up with the general proofs would make it more easily readable. Clearly separating proofs/propositions that get at zero-th order ideas from mathematical technicalities would help.

Revision #3: The results in Figure 1D and e.g., why negative epistasis is maintained across coarse-graining. (see text below Equation 4) were not immediately intuitive.

Possible solutions: Present explicit expressions for H, F, epsilon in Equations 3,4 for a particular simple network and then state that these results in fact hold for any topology as shown in Materials and methods.

2) Generalizability and scope: All results are derived for linear ODEs near equilibrium. The linear steady-state assumption needs to be contextualized. The scope of the study and its practical applications could be understood more clearly if the validity of this approximation were more explicitly discussed (e.g. its validity for mutations of small effect, or as a null model)

Revision #4: The linear steady-state assumptions would be appropriate for mutations of small effect. This issue should be addressed, i.e. whether aspects of network toplogy can be inferred from system-level epistasis may simply depend on the effect size of the "mutations".

Possible solutions: In addition to explicitly address the linear quasi-steady state assumption head on in the Abstract, Introduction and Discussion, the author should also evaluate the conditions where this assumption would be valid, e.g. how small do the changes need to be for the system to behave according to the theory?

3) Context: All reviewers found that the connection to previous work could be strengthened significantly, including references to key previous work.

Revision #5: The revised manuscript should include a more structured discussion of the relevant literature on gene interactions in metabolic networks and in gene-regulatory networks where the assumption of linear reactions is common. A brief discussion of Fisher's geometric model would be useful (e.g. the study by Martin (2014) Genetics should be referenced). The paper should also include a discussion of other systems where a similar “emergent” epistasis has been reported. For protein epistasis, Otwinoswki et al., 2018 made a phenomenological observation of emergence – e.g., epistasis between mutations in β-lactamase can be explained as a global non-linearity applied to non-epistatic linear trait. Sailer et al., 2017, Husain et al., 2020 give mechanistic explanations for how simple global epistasis can emerge from complex underlying interactions, closely tied to the discussion in “Inter-gene epistasis is generic”. Similarly, epistasis due to steady states, central to this paper, is similar to the arguments of Bender, Case and Gilpin, Ecology, 1984 (and later reviews by Case, Billick) in the context of (mistakenly) inferring ecological interactions from correlated variations of species abundances at steady state.

Possible fix: See the references suggested above, as well as:

Clark, A. G. 1991. Mutation-Selection Balance and Metabolic Control theory. Genetics 129: 909-923.

Fievet, J. B., C. Dillmann, and D. de Vienne. 2010. Systemic properties of metabolic networks lead to an epistasis-based model for heterosis. Theor. Appl. Genet. 120: 463-473.

Gjuvsland, A. B., B. J. Hayes, S. W. Omholt, and O. Carlborg. 2007. Statistical epistasis is a generic feature of gene regulatory networks. Genetics 175: 411-420.

Hansen, T. F., and G. P. Wagner. 2001. Modeling genetic architecture: A multilinear model of gene interaction. Theor. Pop. Biol. 59: 61-86.

Keightley, P. D. 1989. Models of quantitative variation of flux in metabolic pathways. Genetics 121: 869-876.

Keightley, P. D. 1996. Metabolic models in selection response. J. theor. biol. 182: 311-316.

Omholt, S., E. Plahte, L. Øyehaug, and K. F. Xiang. 2000. Gene regulatory networks generating the phenomena of additivity, dominance and epistasis. Genetics 155, 969-980.

Peccoud, J., K. Vander Velden, D. Podlich, C. Winkler, L. Arthur and M. Cooper. 2004. The selective values of alleles in a molecular network model are context dependent. Genetics 166: 1715-1725.

Plathe, E., A. B. Gjuvsland, and S. W. Omholt. 2013. Propagation of genetic variation in gene regulatory networks. PhysicaD 256: 7-20.

Wagner, G. P., M. D. Laubichler, and H. Bagheri-Chaichian. 1998. Genetic measurement theory of epistatic effects. Genetica 102/103: 569-580.

Jayawardhana et al. (Handling Biological complexity using Kron reduction, Mathematical Control Theory I, Lecture Notes in Control and Information Sciences, 2015)

Martin (2014), Fisher's Geometrical Model Emerges as a Property of Complex Integrated Phenotypic Networks, Genetics 197(1): 237-255.

Revision#6: Finally, reviewers also identified an issue that needs clarification. The main results appear to be derived for closed chemical reaction networks at steady state, rather than the driven metabolic networks (e.g. those with fixed input flux) that are more commonly considered in other works. This may complicate the interpretation of the results, since we do not know which findings are specific to the equilibrium assumption.

Possible solutions: The author may either (i) explicitly extend the results to allow for networks with a fixed input flux or (ii) simply rephrase key parts of the Abstract, Introduction, and Discussion to make this distinction more explicit. The author should discuss how commonly used approaches such as Flux Balance Analysis would relate to this work, and in particular whether the assumptions of FBA are covered by his model. Can the author comment on how the results would change if one used dynamic flux balance where the environment changes over time as cells grow on it?

---

## [Author Response]

Reviewers were positive overall about the paper, but noted three important areas where it must be improved prior to being acceptable for publication. I proceed to summarize these areas where revisions are needed.With the goal of making these criticisms as useful and constructive as possible, I also include potential solutions to the issues raised by the reviewers. Note that these are offered as suggestions.1) Clarity: The reviewers found that the paper was hard to read and that it would benefit from a clearer presentation. Below I summarize some of the main concerns regarding this issue and provide potential solutions:Revision #1: The manuscript uses a complicated notation and terminology to derive results that are in the end (mathematically) rather simple. This makes the manuscript hard to read, and it will limit the readership. Just remembering the notation and the exact meaning of the terms is a challenge, and this level of complexity is not necessary to derive these results.Possible solutions: Attempt to simplify notation. State the two theorems in the main text rather than in the appendix. Simplify the notation in the main text. Consider moving Proposition 6 to the main text.

I simplified notations wherever possible, reorganized and streamlined the Model and Results sections. I also created the new Figure 1 which illustrates the most important definitions and concepts of the model. I moved Theorems 1 and 2 into the main text and described the key ideas underlying their proofs more clearly. I opted against moving Proposition 5 (former Proposition 6) into the main text because it is rather technical and its purpose would not be clear without providing more details, which would clutter the exposition.

Revision #2: The Materials and methods were hard to navigate even for specialists who are familiar with the math. A better organization should help. For instance, what the reviewers looked for in the Materials and methods was – (a) how to compute the expansion in Equation 3 for a simple network and in general, (b) details to understand the results on series and parallel pathways. These ideas are not easily found in Materials and methods.Possible solutions: Re-organize the Materials and methods section in a more modular way, with clear headings and an introduction describing what is going to be accomplished in each subsection, linking each section with the main results in the text (e.g. "Derivation of Equation X"). Organizing by such themes, working out a specific example, and following it up with the general proofs would make it more easily readable. Clearly separating proofs/propositions that get at zero-th order ideas from mathematical technicalities would help.

I agree that the Materials and methods section in the original version was overly technical and poorly organized. I have now made several substantive changes that hopefully address this problem.

1) The names of the sections now clearly state their purpose.

2) I added a new section “Key ideas and logic” where I now describe the overall zero-th order ideas of the proofs of Theorems 1 and 2 in more detail than in the main text.

3) At the beginning of each section, I now provide additional even more detailed but still informal explanations of what the section is supposed to accomplish.

4) I removed Proposition 4 and Corollary 4 because their statements are quite obvious. Similarly, I removed excessive details from some proofs, which can be easily filled in by the reader, e.g., in Proposition 1 and in Proposition 5 (former Proposition 6).

5) I added sections “Computation of effective rate constants for simple modules” and “Calculation of epistasis in simple modules” which should help the reader understand how to compute function F(**x**) in Equation 1 and epistasis expansions Equation 6 and 9 in practice.

6) I reorganized the appendices.

a) I moved the material from former Appendix 2 (“Properties of the coarse-graining procedure”) into Box 1 in the Materials and methods.

b) I moved the content of the former Appendix 3 (“Functions mapping the rates of two reactions onto module’s effective reaction rate”) into the body of Corollary 3.

c) I moved the content of former Appendix 4 (“Epistasis in a toy metabolic network”) into the new sections mentioned in #5.

d) I added the new Appendix 5 (“Relationship to the Flux Balance Analysis”) to address an issue raised by reviewers (see below).

e) I added the new Appendix 6 (“Connection between metabolism and growth”) to demonstrate explicitly how epistasis that arises in metabolism can translate to epistasis for growth rate. This issues was raised in a separate conversation with one of the reviewers (Ben Good).

Revision #3: The results in Figure 1D and e.g., why negative epistasis is maintained across coarse-graining. (see text below Equation 4) were not immediately intuitive.Possible solutions: Present explicit expressions for H, F, epsilon in Equations 3,4 for a particular simple network and then state that these results in fact hold for any topology as shown in Materials and methods.

Following this suggestion, I added a paragraph immediately before Theorem 1 where I describe the intuition behind this result using the linear pathway example.

2) Generalizability and scope: All results are derived for linear ODEs near equilibrium. The linear steady-state assumption needs to be contextualized. The scope of the study and its practical applications could be understood more clearly if the validity of this approximation were more explicitly discussed (e.g. its validity for mutations of small effect, or as a null model)Revision #4: The linear steady-state assumptions would be appropriate for mutations of small effect. This issue should be addressed, i.e. whether aspects of network toplogy can be inferred from system-level epistasis may simply depend on the effect size of the "mutations".Possible solutions: In addition to explicitly address the linear quasi-steady state assumption head on in the Abstract, Introduction and Discussion, the author should also evaluate the conditions where this assumption would be valid, e.g. how small do the changes need to be for the system to behave according to the theory?

To address this important issue, I carried out additional numerical analyses asking whether the results of Theorems 1 and 2 hold when the effects of mutations are finite. The results of these new analyses are now presented in a new section “Sensitivity of results with respect to the magnitude of mutational effects” and in new Figure 4. Briefly, I tested the sensitivity of Theorem 1 with respect to the magnitude of mutations by numerically calculating the function that maps lower-level epistasis onto higher-level epistasis for modules with randomly sampled kinetic parameters and for mutations with various magnitudes and signs. Overall, the results of Theorem 1 appear to be fairly robust to the magnitude of mutational effects. Predictions of Theorem 1 begin to fail when both mutations have 10-50% positive effects on the rate constant of the lower-level module (see Figure 4A). The results of Theorem 2 are also quite robust with respect to the effect size of mutations. By carrying out similar numerical calculations, I found that mutations in parallel reactions never produced positive epistasis. Mutations in serial reactions never produced negative epistasis, but they did produce values of epistasis between 0 and 1 in a fraction of tested cases (Figure 4B).

I updated the Abstract, Introduction and Discussion accordingly to make the assumptions underlying analytical work clear.

3) Context: All reviewers found that the connection to previous work could be strengthened significantly, including references to key previous work.Revision #5: The revised manuscript should include a more structured discussion of the relevant literature on gene interactions in metabolic networks and in gene-regulatory networks where the assumption of linear reactions is common. A brief discussion of Fisher's geometric model would be useful (e.g. the study by Martin (2014) Genetics should be referenced). The paper should also include a discussion of other systems where a similar “emergent” epistasis has been reported. For protein epistasis, Otwinoswki et al., 2018 made a phenomenological observation of emergence – e.g., epistasis between mutations in β-lactamase can be explained as a global non-linearity applied to non-epistatic linear trait. Sailer et al., 2017, Husain et al., 2020 give mechanistic explanations for how simple global epistasis can emerge from complex underlying interactions, closely tied to the discussion in “Inter-gene epistasis is generic”. Similarly, epistasis due to steady states, central to this paper, is similar to the arguments of Bender, Case and Gilpin, Ecology, 1984 (and later reviews by Case, Billick) in the context of (mistakenly) inferring ecological interactions from correlated variations of species abundances at steady state.Possible fix: See the references suggested above, as well as:Clark, A. G. 1991. Mutation-Selection Balance and Metabolic Control theory. Genetics 129: 909-923.Fievet, J. B., C. Dillmann, and D. de Vienne. 2010. Systemic properties of metabolic networks lead to an epistasis-based model for heterosis. Theor. Appl. Genet. 120: 463-473.Gjuvsland, A. B., B. J. Hayes, S. W. Omholt, and O. Carlborg. 2007. Statistical epistasis is a generic feature of gene regulatory networks. Genetics 175: 411-420.Hansen, T. F., and G. P. Wagner. 2001. Modeling genetic architecture: A multilinear model of gene interaction. Theor. Pop. Biol. 59: 61-86.Keightley, P. D. 1989. Models of quantitative variation of flux in metabolic pathways. Genetics 121: 869-876.Keightley, P. D. 1996. Metabolic models in selection response. J. theor. biol. 182: 311-316.Omholt, S., E. Plahte, L. Øyehaug, and K. F. Xiang. 2000. Gene regulatory networks generating the phenomena of additivity, dominance and epistasis. Genetics 155, 969-980.Peccoud, J., K. Vander Velden, D. Podlich, C. Winkler, L. Arthur and M. Cooper. 2004. The selective values of alleles in a molecular network model are context dependent. Genetics 166: 1715-1725.Plathe, E., A. B. Gjuvsland, and S. W. Omholt. 2013. Propagation of genetic variation in gene regulatory networks. PhysicaD 256: 7-20.Wagner, G. P., M. D. Laubichler, and H. Bagheri-Chaichian. 1998. Genetic measurement theory of epistatic effects. Genetica 102/103: 569-580.Jayawardhana et al. (Handling Biological complexity using Kron reduction, Mathematical Control Theory I, Lecture Notes in Control and Information Sciences, 2015)Martin (2014), Fisher's Geometrical Model Emerges as a Property of Complex Integrated Phenotypic Networks, Genetics 197(1): 237-255.

I very much appreciate this comment and the list references. Even though I was aware of most of the papers mentioned by the reviewers, I was missing many important connections, especially with the evolutionary literature. As a side note, the evolutionary angle was present much more prominently in earlier drafts of the manuscript but inadvertently got edited out in later versions. The new version of the paper has an expanded and structured discussion of the relevant previous literature in the Introduction, and I added several new paragraphs in the Discussion expounding some of these connections. I now reference almost all of the papers suggested in the list above. I decided not to cite Bender et al., 1984 and Plathe et al., 2013 because they only tangentially relevant. Instead of Jayawardhana et al., 2015, I now cite Rao et al., 2014, which is on the same topic but more relevant.

Revision#6: Finally, reviewers also identified an issue that needs clarification. The main results appear to be derived for closed chemical reaction networks at steady state, rather than the driven metabolic networks (e.g. those with fixed input flux) that are more commonly considered in other works. This may complicate the interpretation of the results, since we do not know which findings are specific to the equilibrium assumption.Possible solutions: The author may either (i) explicitly extend the results to allow for networks with a fixed input flux or (ii) simply rephrase key parts of the Abstract, Introduction, and Discussion to make this distinction more explicit. The author should discuss how commonly used approaches such as Flux Balance Analysis would relate to this work, and in particular whether the assumptions of FBA are covered by his model. Can the author comment on how the results would change if one used dynamic flux balance where the environment changes over time as cells grow on it?

A few points in response to this comment. First, a technical correction. A module—the main object of this work for which all the results are derived—is an open system, in the sense that there is constant metabolic flux through it. However, the steady-state flux through a module is determined by both the concentrations of its I/O metabolites (which, importantly, are not part of the module) and by module’s internal kinetic parameters. So, in that sense, modules are indeed not “driven”, as in FBA.

Second, the assumption that a module achieves a steady state, conditional on the concentrations of its I/O metabolites, is not only a more realistic assumption for modeling cells than the assumption of a fixed input flux, it is also a standard assumption in the metabolic control analysis literature. I now discuss both of these assumptions in more detail in the new Appendix 5 (“Relationship to the flux balance analysis”).

Third, modules in my model can be analyzed in the fixed input flux regime simply by making all reactions that connect the input metabolite to the internal metabolites irreversible. This regime (and the formal connection to FBA more generally) is now also discussed in the new Appendix 5. However, this regime is not very interesting from the perspective of epistasis because the flux through the module stops depending on the internal structure of the module. But technically, all my results still hold. Therefore, I did not modify the text.

Finally, the question how the changes in the environment affect epistasis is of course very important, but it goes beyond the scope of this paper. The only statement that I can make at this point is that, as long as the concentrations of the I/O metabolites change sufficiently slowly for the module to remain at steady state (as far as I understand, dynamic FBA makes the same assumption), all results remain unchanged.